# A Bayesian Approach to Contextual Dynamic Pricing using the Proportional Hazards Model with Discrete Price Data

**Dongguen Kim**[1], **Young-Geun Choi**[2], **Minwoo Chae**[1][*]
[1]Department of Industrial and Management Engineering,
Pohang University of Science and Technology
[2]Department of Mathematics Education, Sungkyunkwan University
{dgkim, mchae}@postech.ac.kr; ygchoi@skku.edu

## Abstract

Dynamic pricing algorithms typically assume continuous price variables, which may not reflect real-world scenarios where prices are often discrete. This paper demonstrates that leveraging discrete price information within a semi-parametric model can substantially improve performance, depending on the size of the support set of the price variable relative to the time horizon. Specifically, we propose a novel semi-parametric contextual dynamic pricing algorithm, namely BayesCoxCP, based on a Bayesian approach to the Cox proportional hazards model. Our theoretical analysis establishes high-probability regret bounds that adapt to the sparsity level $\gamma$, proving that our algorithm achieves a regret upper bound of $\widetilde{O}(T^{(1+\gamma)/2} + \sqrt{dT})$ for $\gamma < 1/3$ and $\widetilde{O}(T^{2/3} + \sqrt{dT})$ for $\gamma \geq 1/3$, where $\gamma$ represents the sparsity of the price grid relative to the time horizon $T$. Through numerical experiments, we demonstrate that our proposed algorithm significantly outperforms an existing method, particularly in scenarios with sparse discrete price points.

## 1 Introduction

Contextual dynamic pricing involves updating product prices over time based on contextual information such as customer features, product attributes, and market conditions. Given its importance and practical applications in revenue management, this topic has been extensively explored across statistics, machine learning, and operations research [9, 43, 34]. The primary objective of contextual dynamic pricing is to maximize the seller's revenue through determining optimal prices that account for both covariates and demand uncertainty. A key challenge in dynamic pricing is balancing exploration, which focuses on learning the underlying demand, with exploitation, which leverages current knowledge to set optimal prices. Striking this balance is essential for developing effective dynamic pricing strategies.

A commonly studied framework in contextual dynamic pricing is the binary choice model, where the seller receives binary purchase feedback based on the posted prices [2, 24, 37, 44, 3, 30, 7, 10, 31]. Specifically, at each time $t = 1, \ldots, T$, the seller observes a covariate $X_t \in \mathbb{R}^d$ that captures customer and product features. Based on the observed covariate and historical sales data, the seller determines a price $P_t$ for the product. The customer's valuation of the product, denoted as a random variable $V_t \in \mathbb{R}_{\geq 0}$, is unknown to the seller. Following the posted price, the seller receives binary feedback $Y_t \in \{0, 1\}$, indicating whether a purchase occurred. The customer purchases the product if and only if their valuation $V_t$ exceeds the offered price $P_t$, which can be expressed as $Y_t = \mathbb{1}\{V_t > P_t\}$. Notably, $V_t$ is not directly observed, as it is censored by $P_t$. In the statistical literature, such data

---

[*]Correspondence to: mchae@postech.ac.kr

39th Conference on Neural Information Processing Systems (NeurIPS 2025).

structures are called case 1 interval-censored data, also known as current status data [18]. Case 1 interval-censored data has been extensively studied in survival analysis [11, 13, 28, 38, 25, 20, 19].

We consider a contextual pricing problem under the binary choice model. Let $F(v \mid X_t) = \mathbb{P}(V_t \leq v \mid X_t)$ and $S(v \mid X_t) = 1 - F(v \mid X_t)$ be the cumulative distribution function (c.d.f.) and complementary c.d.f. (or survival function) of $V_t$ given $X_t$, respectively. The expected revenue from a posted price $p$ given the covariate $X_t$ is given as $\mathbb{E}(p \cdot Y_t \mid X_t) = p\mathbb{P}(V_t > p \mid X_t) = pS(p \mid X_t)$. Then an optimal price $P_t^*$ at time $t$ is defined as a price that maximizes the expected revenue:

$$P_t^* \in \underset{p}{\operatorname{argmax}} \, pS(p \mid X_t). \tag{1}$$

The regret at time $t$ is the difference between the expected revenue generated by the optimal price $P_t^*$ and that from the posted price $P_t$, given by $r(t) = P_t^* S(P_t^* \mid X_t) - P_t S(P_t \mid X_t)$. An important objective is to design a pricing policy that minimizes the cumulative regret over a given time horizon $T$, defined as $R(T) = \sum_{t=1}^{T} r(t)$.

As shown in (1), designing an effective pricing policy necessitates accurately estimating the complementary c.d.f. $S(\cdot \mid X_t)$. Thus, a wide range of contextual dynamic pricing algorithms have been developed using various models for the conditional distribution of $V_t$ given $X_t$. Linear models [2, 36, 23, 17, 24, 3, 44, 6, 30, 33, 10, 31], where $F(v \mid X_t) = F_0(v - X_t^\top \beta)$, and log-linear models [37], where $F(v \mid X_t) = F_0(v \cdot \exp(-X_t^\top \beta))$, serve as key examples. Here, $F_0(v) = \mathbb{P}(V_t \leq v \mid X_t = 0)$ represents the baseline c.d.f., and $\beta \in \mathbb{R}^d$ captures the contextual effect. More recently, [7] proposed using the Cox proportional hazards (PH) model, in which the complementary c.d.f. is modeled as $S(v \mid X_t) = S_0(v)^{\exp(X_t^\top \beta)}$. Here, $S_0(v) = 1 - F_0(v)$ represents the baseline complementary c.d.f. In particular, semi-parametric models, which assume that both the nonparametric baseline function $F_0$ (or $S_0$) and the parametric coefficient $\beta$ are unknown, have gained considerable attention recently due to their flexibility and interpretability [37, 30, 7, 10, 31].

In real-world applications, it is crucial to note that offered prices are often observed only on a discrete set. For instance, retailers commonly restrict prices to convenient values for ease of communication and consumer familiarity, and businesses often adhere to predefined discount levels or promotional price points [39, 22]. The significance of discrete price sets in revenue management has been widely recognized [12, 4, 32]. Much of the existing dynamic pricing literature, however, focuses on continuous price spaces [2, 36, 23, 17, 24, 37, 3, 44, 33, 6, 30, 7, 10, 31].

From a theoretical perspective, discrete price sets provide significant advantages for estimating model parameters. In a simple survival analysis setup with i.i.d. case 1 interval-censored observations, [41] demonstrated that the inferential performance of the underlying survival function can be improved by leveraging the fact that the monitoring time (the offered price in the dynamic pricing setting) is discretely supported. To be specific, if the monitoring time is continuous, the optimal convergence rate for estimating the unknown survival function with case 1 interval-censored data is known as $n^{-1/3}$, where $n$ is the sample size [21]. On the other hand, in [41], the monitoring times are assumed to be supported on an equally spaced grid set. Then, they proved that the nonparametric maximum likelihood estimator (NPMLE) achieves a convergence rate of $n^{-(1-\gamma)/2}$ for $\gamma < 1/3$ and $n^{-1/3}$ for $\gamma \geq 1/3$, where $\gamma \in (0, 1]$ represents the sparsity level of the grid relative to the sample size $n$ (a rigorous definition is provided in Section 2). In other words, one can achieve much faster convergence rates if the grid is sparse ($\gamma < 1/3$). Moreover, they developed an inferential procedure, such as the construction of confidence intervals, that does not depend on the unknown quantity $\gamma$, often referred to as an adaptive procedure. While the adaptive procedure in [41] is quite complicated, [5] demonstrated that a much simpler and more practical Bayes procedure is also adaptive, and that the corresponding Bayes estimator achieves the same convergence rate. Although these aforementioned theoretical results are based on a non-contextual setup and i.i.d. data, they suggest that incorporating the discrete support of the price may lead to a pricing policy with smaller cumulative regret compared to one that ignores this information.

Motivated by these insights, we propose a novel semi-parametric contextual dynamic pricing algorithm, BayesCoxCP, based on a Bayesian approach to the Cox PH model with case 1 interval-censored data. The algorithm is specifically designed to exploit the discreteness of the offered price, leading to improved performance. Our theoretical contributions are threefold:

- We derive the posterior convergence rate of the Bayes estimators of the semi-parametric Cox PH model under the i.i.d. setup. We assume that the offered price is supported on an equally

Table 1: Existing regret bounds for contextual dynamic pricing algorithms based on semi-parametric models. Note that the optimal rates depend on the model and assumptions, such as the smoothness of $F_0$.

| Method | Model for $V_t$ | Regret Upper Bound | Optimality in $T$ | Adaptation to discrete support |
|---|---|---|---|---|
| [10] | Linear | $\widetilde{O}((Td)^{\frac{2m+1}{4m-1}})$ | ✕ | ✕ |
| [31] | Linear | $\widetilde{O}(T^{\frac{2}{3}} + \|\widehat{\beta} - \beta^*\|_1 T)$ | ✕ | ✕ |
| [30] | Linear | $\widetilde{O}(T^{\frac{2}{3}} d^2)$ | ✕ | ✕ |
| [30] | Linear | $\widetilde{O}(T^{\frac{3}{4}} d)$ | ✕ | ✕ |
| [37] | Log-linear | $\widetilde{O}(T^{\frac{1}{2}} d^{\frac{11}{4}})$ | ✕ | ✕ |
| [7] | PH | $\widetilde{O}(T^{\frac{2}{3}} d)$ | ○ | ✕ |
| Our work | PH | $\widetilde{O}(T^{\frac{1+\gamma}{2}} + \sqrt{dT})$ $(\gamma < 1/3)$ 
 $\widetilde{O}(T^{\frac{2}{3}} + \sqrt{dT})$ $(\gamma \geq 1/3)$ | ○ | ○ |

spaced grid set and prove that the posterior distribution converges at the optimal rate, which adapts to the grid sparsity. This result generalizes the work of [5], who studied the survival model without covariates, to the PH model. It is also worth noting that our prior for the baseline cumulative hazard differs from that of [5] and can achieve computational benefits.

- We derive the regret upper bound of the proposed BayesCoxCP algorithm. Specifically, our algorithm achieves a regret upper bound of order $T^{\frac{1+\gamma}{2}} + (dT)^{1/2}$ for $\gamma < 1/3$ and $T^{2/3} + (dT)^{1/2}$ for $\gamma \geq 1/3$, up to a logarithmic factor, where $\gamma$ represents the grid sparsity relative to the time horizon $T$. Notably, the BayesCoxCP algorithm does not rely on the value of $\gamma$, i.e., our algorithm adapts to the sparsity level. A careful selection of the exploration parameter $\eta_l$ is crucial in the algorithm's design; see Section 4 for further details.

- We also establish a non-contextual minimax lower bound for the cumulative regret in the discrete pricing problem, as stated in Theorem 5.3. It turns out that our regret upper bound for the BayesCoxCP algorithm is optimal up to a logarithmic factor in terms of $T$.

Through extensive numerical experiments, we empirically demonstrate that the proposed pricing algorithm significantly outperforms the state-of-the-art method in [7] when prices are discretely supported.

The remainder of this paper is organized as follows. In the following subsections, we introduce the notations used throughout the paper and provide a brief summary of related works. Section 2 introduces the basic setup for case 1 interval-censored data on a grid and describes the Cox PH model, along with the prior distributions employed. Section 3 establishes the convergence rate of the posterior distribution under the i.i.d. setup. Section 4 introduces the BayesCoxCP algorithm and Section 5 presents its regret analysis. Finally, Section 6 presents numerical experiments to evaluate the effectiveness of our proposed algorithm.

## 1.1 Notation

For two real numbers $a$ and $b$, $a \vee b$ and $a \wedge b$ denote the maximum and minimum of $a$ and $b$, respectively. For two densities $p$ and $q$ with dominating measure $\nu$, let $\mathcal{D}_H(p, q) = (\int (p^{1/2} - q^{1/2})^2 d\nu)^{1/2}$ be the Hellinger distance and $K(p, q) = \int \log(p/q) p d\nu$ be Kullback–Leibler divergence. For a metric space $(\mathcal{F}, \mathcal{D})$, the $\epsilon$-covering and $\epsilon$-bracketing numbers of $\mathcal{F}$ with respect to distance $\mathcal{D}$ are denoted as $N(\epsilon, \mathcal{F}, \mathcal{D})$ and $N_{[\,]}(\epsilon, \mathcal{F}, \mathcal{D})$, respectively. We write $a = O(b)$ or $a \lesssim b$ if $a \leq Cb$ for some constant $C > 0$, where $C$ is an absolute constant unless otherwise specified. In addition, we write $a = \Omega(b)$ or $a \gtrsim b$ if $a \geq Cb$ for some constant $C > 0$. The notation $\widetilde{O}(\cdot)$ denotes the corresponding bound that ignore logarithmic factors.

## 1.2 Related works

The problem of contextual dynamic pricing has been extensively studied in the literature. Many recent works have focused on semi-parametric models where $F_0$ is unknown and nonparametric. For instance, [30, 31, 10] considered linear models with an unknown $F_0$ under certain smoothness

assumptions. In [10], $F_0$ is assumed to be $m(\geq 2)$th-order smooth, achieving a regret upper bound of $\widetilde{O}((Td)^{\frac{2m+1}{4m-1}})$. [31] relaxed this assumption by assuming that $F_0$ is Lipschitz continuous and second-order smooth. They obtained a regret upper bound of $\widetilde{O}(T^{2/3} + \|\widehat{\beta} - \beta^*\|_1 T)$, where $\|\widehat{\beta} - \beta^*\|_1$ represents the estimation error of $\beta^*$. Similarly, [30] considered the same setting and achieved a regret upper bound of $\widetilde{O}(T^{2/3}d^2)$ under the Lipschitz and second-order smoothness assumptions on $F_0$, while showing that under a weaker Lipschitz assumption alone, the regret upper bound increases to $\widetilde{O}(T^{3/4}d)$. On the other hand, [37] used a log-linear model with a second-order smoothness assumption on $F_0$, achieving a regret upper bound of $\widetilde{O}(T^{1/2}d^{11/4})$ but with suboptimal dependency on the dimension $d$. Similar to our approach, [7] used the Cox PH model, assuming that $F_0$ is Lipschitz continuous. They derived a regret upper bound of $\widetilde{O}(T^{2/3}d)$, which improves the dimensional dependency compared to [37], but their analysis is limited to continuous pricing settings. The overall comparison of regret bounds from these semi-parametric studies, along with our results, is summarized in Table 1. In addition to these works, earlier studies often assumed that $F_0$ is known and log-concave. For instance, [24] and [44] both considered linear models under these assumptions. [24] additionally analyzed the case where $F_0$ is unknown but belongs to a parametric log-concave family, deriving a regret upper bound of order $T^{1/2}$.

## 2 Preliminaries

### 2.1 Basic setup

In the current and next sections, we study the behavior of the posterior distribution from the PH model for analyzing case 1 interval-censored data on a grid under the i.i.d. regime.

To set the scene, suppose that $(X_t, P_t, Y_t)$, $t = 1, \ldots, n$, are i.i.d. copies of $(X, P, Y)$. In particular, we assume that $P_t$'s are supported on the grid set $\mathcal{G} = \{g_1, \ldots, g_K\}$ within the (fixed) interval $[p_{\min}, p_{\max}]$, whose cardinality may depend on the sample size $n$. The grid points are assumed to be uniform in the sense that $g_{k+1} = g_k + \delta$ for every $k \geq 0$, where $g_0 = p_{\min}$, and $K$ is the largest integer such that $g_K \leq p_{\max}$, that is, $K = \lfloor (p_{\max} - p_{\min})/\delta \rfloor$. We further assume that the grid resolution $\delta$ is controlled by two constants $\gamma \in (0, 1]$ and $\kappa > 0$, according to the relation $\delta = \kappa n^{-\gamma}$. Note that generalizations to nonuniform grids are discussed in Appendix D.

Let $\mathbb{Q}(\cdot \mid X)$ denote the conditional distribution of $P$ given $X$, with $q(\cdot \mid X)$ denoting the corresponding probability mass function. In addition, the marginal distribution of the price $P$ is denoted by $\mathbb{Q}(\cdot)$, with its probability mass function given by $q(\cdot)$. Let $\mathbb{P}_X$ and $p_X$ be the marginal distribution and the corresponding density of $X$.

### 2.2 Proportional hazards model for $V_t$

We consider the Cox PH model for the conditional distribution of $V_t$ given $X_t$. Formally, the complementary c.d.f. $v \mapsto S(v \mid X_t)$ of $V_t$ is modeled as

$$S(v \mid X_t) = S_0(v)^{\exp(X_t^\top \beta)},$$

where $S_0(\cdot)$ is a baseline complementary c.d.f., and $\beta \in \mathbb{R}^d$ is a regression coefficient. We assume that $V_t$ is continuous. Let $F_0 = 1 - S_0$, $\lambda_0 = F_0'/S_0$ and $\Lambda_0(v) = \int_0^v \lambda_0(u)du$ be the c.d.f., hazard and cumulative hazard functions, respectively, corresponding to $S_0$, where $F_0'$ denotes the derivative of $F_0$.

We remark that the joint distribution of $(X_t, P_t, Y_t)$ depends on the unknown parameters $(S_0, \beta, p_X, q(\cdot \mid \cdot))$. Among these, While $(S_0, \beta)$ are the parameters of interest, while $(p_X, q(\cdot \mid \cdot))$ are treated as nuisance parameters (at least in the current and next sections). Since $P_t$ is supported on $\mathcal{G}$, the joint distribution of $(X_t, P_t, Y_t)$ depends on $S_0$ only through the vector $\mathbf{S}_0 = (S_{0,1}, \ldots, S_{0,K}) \in [0,1]^K$, where $S_{0,k} = S_0(g_k)$. Here, $\mathcal{X}$ represents the support of the covariate $X$. The parameter space is defined as $\Theta = \{\theta = (\mathbf{S}_0, \beta) \in \mathcal{S}_0 \times \mathbb{R}^d\}$, where $\mathcal{S}_0 = \{\mathbf{S}_0 = (S_{0,1}, \ldots, S_{0,K}) : 1 > S_{0,1} \geq \cdots \geq S_{0,K} > 0\}$.

## 2.3 Prior

We note that $V_t$ is continuous while $P_t$ is discrete with support $\mathcal{G}$. To reflect this structure, we model the baseline hazard function $\lambda_0(\cdot)$ as a left-continuous step function, where the jump points are located at grid points. Let $\boldsymbol{\lambda}_0 = (\lambda_{0,1}, \dots, \lambda_{0,K})$, $\boldsymbol{\Lambda}_0 = (\Lambda_{0,1}, \dots, \Lambda_{0,K})$ and

$$\lambda_0(p) = \sum_{k=1}^{K} \lambda_{0,k} \mathbb{1}\{p \in (g_{k-1}, g_k]\}, \tag{2}$$

where $\Lambda_{0,k} = -\log S_{0,k}$. Since there is a one-to-one correspondence between $\mathbf{S}_0$ and $\boldsymbol{\lambda}_0$, one can impose a prior on $\mathbf{S}_0$ through $\boldsymbol{\lambda}_0$. We consider an independent prior for the unknown parameters $\boldsymbol{\lambda}_0$ and $\beta$, specified as $\Pi = \Pi_\beta \times \Pi_{\boldsymbol{\lambda}_0}$. Here, $\Pi_{\boldsymbol{\lambda}_0}$ consists of independent gammas:

$$\lambda_{0,k} \sim \text{Gamma}(\alpha_k, \rho), \quad k = 1, \dots, K, \tag{3}$$

where $\text{Gamma}(\alpha_k, \rho)$ denotes the gamma distribution with mean $\alpha_k/\rho$ and variance $\alpha_k/\rho^2$. Gamma priors are commonly employed for $\lambda_0$ in Bayesian analyses of the PH model, as seen in [27, 47, 35, 29]. We further impose the following conditions on the prior:

(P1) $\Pi_\beta$ has a continuous and positive Lebesgue density on $\mathbb{R}^d$.

(P2) There exist positive constants $\underline{\alpha} < \overline{\alpha}$, such that $\underline{\alpha} \le \alpha_k \le \overline{\alpha}$ for $k = 1, \dots, K$.

## 3 Posterior convergence rate under i.i.d. setup

To clarify notation, we use the superscript $*$ to denote the true parameter, e.g., $\Lambda_0^*$, $\beta^*$, and $\lambda_0^*$. Suppose that there exists a true parameter $\theta^* = (\mathbf{S}_0^*, \beta^*)$ generating the data $\mathbf{D}_n = \{(X_t, P_t, Y_t)\}_{t=1}^n$. (We may regard $p_X$ and $q(\cdot \mid \cdot)$ as known parameters if we are only interested in inferring $\theta$.) Given the data $\mathbf{D}_n$, let $\Pi(\cdot \mid \mathbf{D}_n)$ be the joint posterior distribution of $\boldsymbol{\lambda}_0$ and $\beta$.

**Assumptions** We will prove that $\Pi(\cdot \mid \mathbf{D}_n)$ concentrates around $\theta^*$ under the following assumptions:

(A1) $\|\beta^*\|_2 \le B$ for some constant $B > 0$.

(A2) $\mathbb{P}_X(\mathcal{X}) = 1$ and $p_X$ is bounded away from zero on $\mathcal{X}$, where $\mathcal{X} = \{x \in \mathbb{R}^d : \|x\|_2 \le L\}$.

(A3) $\mathbb{P}(X^\top \beta_1 \ne X^\top \beta_2) > 0$ for $\beta_1 \ne \beta_2$.

(A4) For $x \in \mathcal{X}$ and $1 \le k \le K$, suppose $\mathbb{Q}(\mathcal{G} \mid X = x) = 1$, and $q(g_k \mid x) \gtrsim n^{-\frac{1+\gamma}{2}}(\log n)^{\frac{1}{2}}$ if $\gamma < 1/3$, or $q(g_k \mid x) \gtrsim n^{-\gamma - \frac{1}{3}}(\log n)^{\frac{1}{2}}$ otherwise.

(A5) The support of $F_0^*$ is $[v_{\min}, v_{\max}]$, $v_{\min} < p_{\min} < p_{\max} < v_{\max}$, and $S_0^*$ has a continuous and strictly negative derivative on $[v_{\min}, v_{\max}]$.

Assumptions (A1) and (A2) are commonly adopted in the stochastic contextual dynamic pricing literature. Assumption (A3) ensures the identifiability of the regression coefficient. Assumption (A4) requires that the conditional distribution $\mathbb{Q}(\cdot \mid x)$ maintains a certain level of uniformity over the grid set $\mathcal{G}$. For instance, when $\mathbb{Q}(\cdot \mid x)$ follows a uniform distribution over $\mathcal{G}$, (A4) is satisfied for any $\gamma \in [0, 1]$. In the contextual dynamic pricing problem, the function $q(\cdot \mid x)$ is parameterized by the pricing policy. Therefore, constructing a policy that satisfies (A4) is crucial. In Section 4, we explicitly design a policy that fulfills (A4). Assumption (A5) implies that $S_0^*$ is $L_0$-Lipschitz on $[p_{\min}, p_{\max}]$ for some constant $L_0 > 0$, and that $S_{0,1}$ and $S_{0,K}$ are bounded away from 0 and 1.

We define the distance $\mathcal{D}_{\mathbb{Q}}$ on the parameter space $\Theta$ as

$$\mathcal{D}_{\mathbb{Q}}(\theta_1, \theta_2) = \|\mathbf{S}_{0,1} - \mathbf{S}_{0,2}\|_{2,\mathbb{Q}} + \|\beta_1 - \beta_2\|_2,$$

for any $\theta_1 = (\mathbf{S}_{0,1}, \beta_1), \theta_2 = (\mathbf{S}_{0,2}, \beta_2) \in \Theta$, where $\|\cdot\|_{2,\mathbb{Q}}$ denotes the $L_2(\mathbb{Q})$ norm with respect to a probability measure $\mathbb{Q}$, that is, $\|\mathbf{S}_0\|_{2,\mathbb{Q}} = \left(\sum_{k=1}^K (S_{0,k})^2 q(g_k)\right)^{1/2}$. For a given parameter $\theta$, let $\mathbb{P}_\theta^n$ denote the law of $\mathbf{D}_n$ under $\theta$, and let $\mathbb{E}_\theta^n$ be the corresponding expectation. With these definitions in place, we now state two theorems that establish the convergence rates of the posterior distribution in two distinct cases: $\gamma < 1/3$ and $\gamma \ge 1/3$.

**Theorem 3.1** (Case $\gamma < 1/3$). *Suppose that $\gamma < 1/3$ and assumptions (A1)-(A5) hold. Let $\epsilon_n = n^{-\frac{1-\gamma}{2}}\sqrt{\log n} + \sqrt{\frac{d}{n}}\sqrt{\log(d \vee n)}$. Then, there exist positive constants $C_1, \ldots, C_4$, depending only on $L, B, p_{\min}, p_{\max}, \kappa, \underline{\alpha}, \overline{\alpha}, \rho$, such that for $n \geq C_4$,*

$$\Pi\left(\mathcal{D}_{\mathbb{Q}}(\theta, \theta^*) \geq C_1 \epsilon_n \mid \mathbf{D}_n\right) \leq C_2 \exp(-C_3 n \epsilon_n^2),$$

*with $\mathbb{P}_{\theta^*}^n$-probability at least $1 - \left(\exp(-C_3 n \epsilon_n^2) + 1/n\epsilon_n^2\right)$.*

**Theorem 3.2** (Case $\gamma \geq 1/3$). *Suppose that $\gamma \geq 1/3$ and assumptions (A1)-(A5) hold. Let $\epsilon_n = \left(\frac{\log n}{n}\right)^{\frac{1}{3}} + \sqrt{\frac{d}{n}}\sqrt{\log(d \vee n)}$. Then, there exist positive constants $C_1, \ldots, C_4$, depending only on $L, B, p_{\min}, p_{\max}, \kappa, \underline{\alpha}, \overline{\alpha}, \rho$, such that*

$$\Pi\left(\mathcal{D}_{\mathbb{Q}}(\theta, \theta^*) \geq C_1 \epsilon_n \mid \mathbf{D}_n\right) \leq C_2 \xi_n, \quad n \geq C_4$$

*with $\mathbb{P}_{\theta^*}^n$-probability at least $1 - \left(\xi_n + 1/n\epsilon_n^2\right)$, where $\xi_n = \begin{cases} \exp(-C_3 n \epsilon_n^2) & \text{if } \gamma < \frac{2}{3}, \\ \exp\left(-C_3 n^{\frac{1}{3}}\right) & \text{if } \gamma \geq \frac{2}{3}. \end{cases}$*

Theorems 3.1 and 3.2 show that the convergence rate of the posterior distribution adapts to the sparsity level $\gamma$. Importantly, when $\gamma \geq 1/3$, the posterior achieves the convergence rate of $n^{-1/3}$, as in the continuous observation setting. In contrast, for $\gamma < 1/3$, the posterior attains a faster rate of $n^{-\frac{1-\gamma}{2}}$, highlighting the advantage of discrete observations in sparse grids. This result generalizes the work of [5], which focused on the non-contextual case 1 interval-censored data, to the PH model.

## 4 Proposed BayesCoxCP algorithm

We now propose the contextual discrete pricing algorithm, namely BayesCoxCP, based on a Bayesian approach to the semi-parametric Cox PH model. Consider the discrete pricing setting introduced earlier. Assume that the support of $P_t$ is $\mathcal{G} = \{g_k : k = 1, \ldots, K\}$ for every $t = 1, \ldots, T$, where $T$ denotes the time horizon. Here, $g_k = p_{\min} + k\delta$ for $k = 1, \ldots, K$, $K = \lfloor(p_{\max} - p_{\min})/\delta\rfloor$, and $\delta = \kappa T^{-\gamma}$ for two constants $\gamma \in (0, 1]$ and $\kappa > 0$. Under the PH assumption with the true pair $(S_0^*, \beta^*)$, the optimal price $P_t^*$ at time $t$ can be defined as $P_t^* \in \mathrm{argmax}_{p \in \mathcal{G}}\left\{p \cdot S_0^*(p)^{\exp(X_t^\top \beta^*)}\right\}$. Let $\mathbb{Q}^*$ denote the marginal distribution of $P_t^*$, with its associated probability mass function denoted by $q^*$.

We employ an epoch-based design that divides the given horizon $T$ into multiple epochs and executes identical pricing policies on a per-epoch basis. Such a design was widely adopted in the literature [24, 44, 7]. Epochs are indexed by $l$, and the length of the epoch $l$ is denoted by $n_l$. The length increases geometrically with $l$, given by $n_l = n_1 2^{l-1}$ for $l \geq 1$. The set of time indices for epoch $l$ is given by $\mathcal{E}_l = \{\sum_{s=0}^{l-1} n_s + 1, \ldots, \sum_{s=0}^{l} n_s\}$, with $n_0 = 0$, ensuring a sequential partitioning of the entire horizon.

**Posterior-based estimation** Let $\mathbf{D}_l = \{(X_t, P_t, Y_t)\}_{t \in \mathcal{E}_l}$ denote the data collected during epoch $l \geq 1$. We employ a consistent prior across all epochs, denoted as $\Pi = \Pi_\beta \times \Pi_{\boldsymbol{\lambda}_0}$, where $\Pi_{\boldsymbol{\lambda}_0}$ consists of independent gamma distributions:

$$\lambda_{0,k} \sim \text{Gamma}(\alpha_k, \rho), \quad k = 1, \ldots, K, \tag{4}$$

with $\alpha_k = \alpha$ for $k = 1, \ldots, K$ and a fixed constant $\alpha > 0$. The prior $\Pi_\beta$ on $\beta$ has a density with respect to the Lebesgue measure on $\mathbb{R}^d$, bounded away from zero in a neighborhood of $\beta^*$. Common choices for $\Pi_\beta$ include multivariate distributions such as the normal distribution. For each epoch $l$, let $\Pi(\cdot \mid \mathbf{D}_{l-1})$ denote the joint posterior distribution of $\boldsymbol{\lambda}_0$ and $\beta$ based on the data from the previous epoch, $\mathbf{D}_{l-1}$. We denote the point estimator for the true parameter $\theta^*$ as $\widehat{\theta}^{l-1} = (\widehat{S}_0^{l-1}, \widehat{\beta}^{l-1})$, derived from the observations $\mathbf{D}_{l-1}$ in the previous epoch. Specifically, the estimator $\widehat{\theta}^{l-1}$ is obtained as the mean of truncated posterior distribution $\widetilde{\Pi}(\cdot \mid \mathbf{D}_{l-1}) = \Pi(\cdot \mid \mathbf{D}_{l-1})/\Pi(\widetilde{\Theta} \mid \mathbf{D}_{l-1})$, where the truncated parameter space is defined by $\widetilde{\Theta} = \mathcal{S}_0 \times [a, b]^d$ for fixed constants $a$ and $b$.

---

**Algorithm 1** Bayes Cox Contextual Pricing Algorithm (BayesCoxCP)

---

    **Input:** $n_1$: the length of the first epoch; $\eta_1, \eta_2$: degree of exploration; $\Pi_{\boldsymbol{\lambda}_0}, \Pi_\beta$: prior; $a, b$: truncated range

1: For $t = 1, \ldots, n_1$, uniformly choose $P_t$ from $\mathcal{G}$, and get reward $Y_t$;
2: **for** epoch $l = 2, 3, \ldots$ **do**
3:     Obtain the estimator $\widehat{\theta}^{l-1} = (\widehat{\mathbf{S}}_0^{l-1}, \widehat{\beta}^{l-1})$ from $\Pi(\cdot \mid \mathbf{D}_{l-1})$.
4:     **for** time $t \in \mathcal{E}_l$ **do**
5:         Observe $X_t$ and draw a binary number $R$ from Bernoulli$(1 - \eta_l)$;
6:         **if** $R = 1$ **then** $P_t \in \mathrm{argmax}_{p \in \mathcal{G}} \left\{ p \cdot \widehat{S}_0^{l-1}(p)^{\exp(X_t^\top \widehat{\beta}^{l-1})} \right\}$
7:         **else** Uniformly choose $P_t$ from $\mathcal{G}$
8:         **end if**
9:         Get reward $Y_t$.
10:     **end for**
11: **end for**

---

**Pricing policy** We denote the pricing policy for epoch $l$ as $\pi_l : \mathcal{X} \to \mathcal{P}(\mathcal{G})$, where $\mathcal{P}(\mathcal{G})$ denotes the set of all probability distributions over the grid $\mathcal{G}$. Specifically, given covariates $X_t$ for $t \in \mathcal{E}_l$, the distribution $\pi_l(X_t)$ is defined as a mixture distribution given by

$$\pi_l(X_t)(A) = (1 - \eta_l) \cdot \delta_{\widehat{P}_t^{l-1}}(A) + \eta_l \cdot \mathbb{U}_\mathcal{G}(A) \tag{5}$$

for any $A \subset \mathcal{G}$, where $\widehat{P}_t^{l-1}$ is the myopic policy determined by the estimate $\widehat{\theta}^{l-1} = (\widehat{\mathbf{S}}_0^{l-1}, \widehat{\beta}^{l-1})$ as $\widehat{P}_t^{l-1} \in \mathrm{argmax}_{p \in \mathcal{G}} \left\{ p \cdot \widehat{S}_0^{l-1}(p)^{\exp(X_t^\top \widehat{\beta}^{l-1})} \right\}$. Here, $\delta_P$ denotes the Dirac measure centered at $P$, $\mathbb{U}_\mathcal{G}$ represents the discrete uniform distribution over $\mathcal{G}$, and $\eta_l$ is an epoch-specific exploration parameter, defined as

$$\eta_l = \min \left\{ \eta_1 \left( \eta_2 \sqrt{|\mathcal{G}|/2^{l-1}} \wedge 2^{-\frac{l-1}{3}} \right) \sqrt{\log 2^{l-1}}, 1 \right\}, \tag{6}$$

where $\eta_1$ and $\eta_2$ are global constants. The design in (6) reduces the need for uniform exploration when the grid is sparse, while increasing it as the grid becomes denser, effectively balancing exploration and exploitation across different epochs. The choice of $\eta_l$ directly ensures that assumption (A4) is satisfied, since $\eta_l$ controls the degree of uniform exploration over the grid, which is reflected in $q(\cdot \mid x)$. This connection is rigorously established in Lemma C.4. In our numerical experiments, $\eta_1$ and $\eta_2$ are tuned to optimize the degree of exploration.

in

The pseudo-code for the proposed policy is presented in Algorithm 1. In this algorithm, for each time $t \in \mathcal{E}_l$, the offered price $P_t$ solely relies on the observed covariate $X_t$ and the data from previous epochs, $\mathbf{D}_1, \ldots, \mathbf{D}_{l-1}$, while the distribution of $V_t$ only depends on $X_t$. Thus, Algorithm 1 ensures conditional independence between $V_t$ and $P_t$ given $X_t$, i.e., $V_t \perp P_t | X_t$ for each $t \in \mathcal{E}_l$. Moreover, given the data from previous epochs $1, \ldots, l-1$, $\{(X_t, P_t, Y_t)\}_{t \in \mathcal{E}_l}$ are independent and identically distributed observations, which facilitates separate estimation of $\theta^*$ for each epoch.

For the computation of $\widehat{\theta}^{l-1}$ for each epoch $l$, we employ the variational Bayesian (VB) method for the PH model with case 1 interval-censored data. The VB approach has recently emerged as a computationally efficient alternative while maintaining estimation accuracy; see [29]. Alternatively, one may employ Markov chain Monte Carlo (MCMC) methods [27, 47, 35], which facilitate inference, such as constructing credible intervals for $\theta^*$.

## 5 Regret analysis

In this section, we analyze the regret upper bound for the BayesCoxCP algorithm. Furthermore, we prove the regret lower bound for the discrete pricing problem.

### 5.1 Regret upper bound

We first introduce several technical assumptions and a key lemma that establishes the estimation error of the estimator $\widehat{\theta}^{l-1}$ for each epoch.

We begin by assuming the following additional conditions:

- (B1) For any $x \in \mathcal{X}$, there exists a unique maximizer of the map $p \mapsto pS_0^*(p)^{\exp(x^\top \beta^*)}$ : $[p_{\min}, p_{\max}] \to \mathbb{R}$.

- (B2) The density of the unique maximizer of the map $p \mapsto pS_0^*(p)^{\exp(X^\top \beta^*)}$ : $[p_{\min}, p_{\max}] \to \mathbb{R}$ is bounded away from zero on $[p_{\min}, p_{\max}]$.

The uniqueness condition in assumption (B1) is commonly adopted in the contextual dynamic pricing literature [24, 44, 10]. Aditionally, assumption (B2) ensures that $q^*(p) \asymp \delta$ for all $p \in \mathcal{G}$.

We remark that the grid $\mathcal{G}$ remains unchanged across epochs in our setup, so the grid sparsity relative to the sample size $n_l$ differs for each epoch. For each $l = 1, 2, \ldots$, define $\gamma_l$ as the sparsity level in epoch $l$, such that $K = \lfloor (p_{\max} - p_{\min})/(\kappa n_l^{-\gamma_l}) \rfloor$. Therefore, for all $l = 1, 2, \ldots$, we have

$$\left\lfloor \frac{p_{\max} - p_{\min}}{\kappa} n_l^{\gamma_l} \right\rfloor = \left\lfloor \frac{p_{\max} - p_{\min}}{\kappa} T^{\gamma} \right\rfloor. \tag{7}$$

Let $\mathbb{Q}_l(\cdot \mid X)$ and $q_l(\cdot \mid X)$ denote the conditional distribution and corresponding probability mass function of $P_t$ given $X$ (and $\mathbf{D}_1, \ldots, \mathbf{D}_{l-1}$) during epoch $l$, respectively. The marginal distribution of $P_t$ during epoch $l$ is denoted by $\mathbb{Q}_l$, with $q_l$ as its probability mass function. Then, $q_l(\cdot \mid x) = \pi_l(x)(\cdot)$ for $x \in \mathcal{X}$.

The following lemma provides an upper bound on the estimation error of the point estimator $\widehat{\theta}^{l-1}$ at epoch $l$.

**Lemma 5.1.** *Let the prior $\Pi$ and policy $\pi_l$ be as described above (see (4) and (5)). Suppose that assumptions (A1)-(A3) and (A5) hold. Then, there exist positive constants $C_1, C_2, C_3$ and $C_4$ depending on $L, B, p_{\min}, p_{\max}, \kappa, \alpha, \rho, a, b$ and $n_1$, such that for $l \geq C_4$,*

$$\mathcal{D}_{\mathbb{Q}_{l-1}}(\widehat{\theta}^{l-1}, \theta^*) \leq C_1 \epsilon_{l-1}$$

*with probability at least $1 - \zeta_{l-1} - 1/(n_{l-1}\epsilon_{l-1}^2)$, where*

$$\epsilon_l = \begin{cases} n_l^{-\frac{1-\gamma_l}{2}} \sqrt{\log n_l} + \sqrt{\frac{d}{n_l}} \sqrt{\log(d \vee n_l)} & \text{if } \gamma_l < \frac{1}{3} \\ \left(\frac{\log n_l}{n_l}\right)^{\frac{1}{3}} + \sqrt{\frac{d}{n_l}} \sqrt{\log(d \vee n_l)} & \text{if } \gamma_l \geq \frac{1}{3} \end{cases} \quad \text{and} \quad \zeta_l = \begin{cases} \exp(-C_2 n_l \epsilon_l^2) & \text{if } \gamma_l < \frac{1}{3}, \\ \exp(-C_3 n_l^{\frac{1}{3}}) & \text{if } \gamma_l \geq \frac{1}{3}. \end{cases}$$

Lemma 5.1 implies that $\widehat{\theta}^{l-1}$ achieves an error bound that is adaptive to the grid sparsity level $\gamma_l$ in epoch $l$. By leveraging the consistency of $\widehat{\theta}^{l-1}$ and Hoeffding's inequality, the regret during epoch $l$ can be upper bounded by

$$\sum_{t \in \mathcal{E}_l} r(t) \leq C_1 n_l \mathcal{D}_{\mathbb{Q}_{l-1}}(\widehat{\theta}^{l-1}, \theta^*) + C_2 n_l \eta_l$$

with high probability, where $C_1$ and $C_2$ are positive constants which do not scale with $n_l$ and $d$ (see Lemma B.1 for details). This inequality shows how the estimation error of $\widehat{\theta}^{l-1}$ and the exploration parameter $\eta_l$ affect the regret upper bound. Combining Lemma 5.1 and (6), we now state the main results regarding the regret upper bound for the BayesCoxCP algorithm.

**Theorem 5.2.** *Under the same conditions as in Lemma 5.1, along with assumptions (B1) and (B2), there exist positive constants $C_1, \ldots, C_7$ depending on $L, B, p_{\min}, p_{\max}, \kappa, \alpha, \rho, a, b, \eta_1, \eta_2, \gamma$ and $n_1$ such that for $T \geq C_1$,*

$$R(T) \leq \begin{cases} C_2\sqrt{dT \log(d \vee T)} + C_3 T^{\frac{\gamma+1}{2}} \sqrt{\log T} & \text{if } \gamma < \frac{1}{3}, \\ C_4\sqrt{dT \log(d \vee T)} + C_5 T^{\frac{2}{3}} \sqrt{\log T} & \text{if } \gamma \geq \frac{1}{3}, \end{cases}$$

*with probability at least $1 - \zeta$, where $\zeta = \begin{cases} C_6 \log(T/n_1 + 1)/T^{\gamma} & \text{if } \gamma < \frac{1}{3}, \\ C_7 \log(T/n_1 + 1)/T^{2/9} & \text{if } \gamma \geq \frac{1}{3}. \end{cases}$*

Theorem 5.2 shows that the BayesCoxCP algorithm achieves a regret upper bound that adapts to the unknown sparsity level $\gamma$, ensuring efficient performance without prior knowledge of $\gamma$.

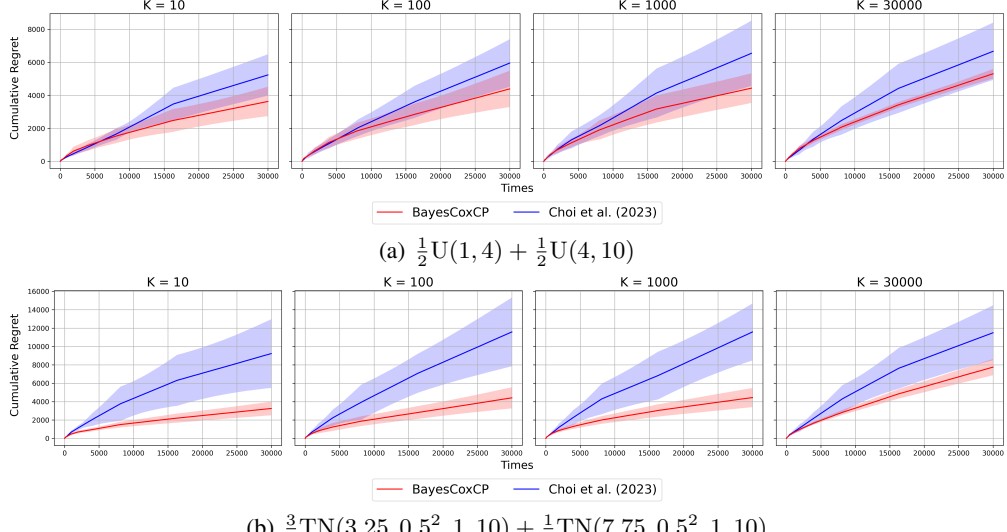

(a) $\frac{1}{2}\mathrm{U}(1,4) + \frac{1}{2}\mathrm{U}(4,10)$

(b) $\frac{3}{4}\mathrm{TN}(3.25, 0.5^2, 1, 10) + \frac{1}{4}\mathrm{TN}(7.75, 0.5^2, 1, 10)$

Figure 1: The cumulative regret curves compare the proposed algorithm (BayesCoxCP) with other method. Solid lines indicate averages, and bands show standard errors over replications.

Compared to existing work in continuous pricing settings, our results underscore the theoretical advantage of utilizing the information that the price is discretely supported. For instance, [7] derived a regret upper bound of $\widetilde{O}(T^{2/3}d)$ under similar assumptions. While this bound is comparable to our result for $\gamma \geq 1/3$, our algorithm achieves a strictly faster regret rate of $O(T^{\frac{1+\gamma}{2}} + (dT)^{1/2})$ for $\gamma < 1/3$, which is a distinct advantage of the grid-based setting. For additional discussion on the possibility of replacing the Bayes estimator with the NPMLE and its effect on exploration parameter choice, please refer to Appendix G.

## 5.2   Regret lower bound

In this subsection, we establish a regret lower bound for the non-contextual pricing problem in the discrete pricing setting. The proof carefully incorporates ideas from [26] and [14], widely used for regret lower bounds in dynamic pricing and multi-armed bandit problems, with a focus on the discrete price setting. Specifically, for dense grids where $\gamma \geq 1/3$, we partition the grid set $\mathcal{G}$ into $T^{1/3}$ segments to derive the lower bound. Further details of the proof are provided in Appendix B.3.

**Theorem 5.3.** *(Lower bound for non-contextual pricing) Consider a non-contextual pricing problem where the valuations are sampled independently and identically from a fixed unknown distribution satisfying the c.d.f. $F(v)$ is bounded away from 0 and 1 for $v \in [p_{\min}, p_{\max}]$ and at least one maximizer of the revenue curve $v \cdot (1 - F(v))$ lies over $\mathcal{G}$. Then, for any $\eta > 0$, no pricing policy (algorithm) can achieve expected regret $O(T^{\frac{1+\gamma}{2} - \eta})$ if $\gamma < 1/3$, and $O(T^{\frac{2}{3} - \eta})$ if $\gamma \geq 1/3$.*

As in Theorem 5.3, the regret lower bound depends on the grid sparsity level $\gamma$ as well. Specifically, for $\gamma < 1/3$, the regret lower bound scales as $\Omega(T^{\frac{1+\gamma}{2}})$, while for $\gamma \geq 1/3$, it scales as $\Omega(T^{2/3})$. Comparing these results with Theorems 3.1 and 3.2, the regret upper bounds achieved by BayesCoxCP algorithm match the lower bounds in terms of $T$, up to a logarithmic factor. Note that the dependency on the dimension $d$ is not addressed in this work, leaving it as a direction for future research.

## 6   Numerical experiments

In this section, we conduct numerical experiments to evaluate the performance of the BayesCoxCP algorithm. Since our objective is to highlight the benefits of leveraging discrete support information, we focus on a comparison with the algorithm proposed by [7]. For comparisons of the PH model-based algorithm with other approaches, such as linear and log-linear model-based algorithms, we refer to [7].

We consider the following experimental setup. The covariate $X_t$ is drawn from a $d$-dimensional ball with a radius of $1/2$ under a uniform distribution, where $d = 5$. The true regression coefficient $\beta^*$ is

set as $\beta^* = \frac{4}{\sqrt{d}}\mathbf{1}_d$, where $\mathbf{1}_d$ denotes a $d$-dimensional vector of ones. For the true baseline distribution, we consider two different mixture distributions. The first is a uniform mixture distribution given by $\frac{1}{2}\mathrm{U}(1,4) + \frac{1}{2}\mathrm{U}(4,10)$, where $\mathrm{U}(a,b)$ denotes the uniform distribution over $[a,b]$. The second is a truncated normal mixture distribution given by $\frac{3}{4}\mathrm{TN}(3.25, 0.5^2, 1, 10) + \frac{1}{4}\mathrm{TN}(7.75, 0.5^2, 1, 10)$, where $\mathrm{TN}(\mu, \sigma^2, a, b)$ represents the truncated normal distribution with mean $\mu$, variance $\sigma^2$, and support $[a,b]$. The grid set $\mathcal{G} = \{g_1, \ldots, g_K\}$ is chosen from $[1, 10]$ with four different values of $K \in \{10, 100, 1000, 30000\}$. The total time horizon is set to $T = 30000$ for all experiments. To conserve space, the detailed hyperparameter settings for all algorithms used in the experiments are provided in Appendix H.

The cumulative regret results for different grid sizes, averaged over 20 replications, are shown in Figure 1. BayesCoxCP consistently achieves lower cumulative regret compared to the method proposed by [7], with the difference being particularly significant when $K$ is small. Notably, the performance gap gradually diminishes as $K$ increases. These findings empirically demonstrate that BayesCoxCP adapts effectively to varying grid resolutions, providing strong empirical support for its theoretical guarantees.

## Acknowledgements

This work was supported by the National Research Foundation of Korea (NRF) grant funded by the Korea government (MSIT) (No. RS-2023-00240861, RS-2023-00252026), a Korea Institute for Advancement of Technology (KIAT) grant funded by the Korea Government (MOTIE) (RS-2024-00409092, 2024 HRD Program for Industrial Innovation), and Institute of Information & Communications Technology Planning & Evaluation(IITP)-Global Data-X Leader HRD program grant funded by the Korea government (MSIT) (IITP-2024-RS-2024-00441244).

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

## Appendix

We begin this appendix with a proof roadmap that outlines how the main lemmas and theorems are logically connected. The roadmap provides a high-level overview of the argument structure, highlighting the key intermediate steps and how they contribute to the main convergence and regret results. This overview is intended to enhance clarity and to guide the reader through the subsequent detailed proofs.

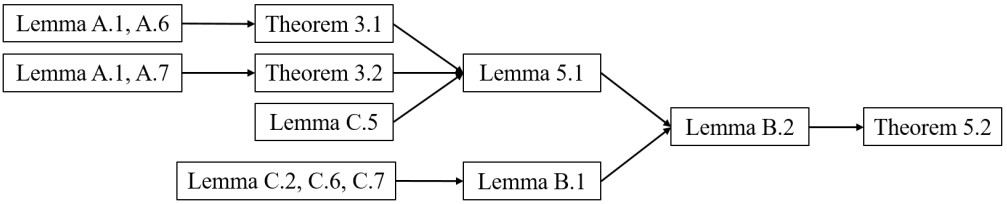

Figure 2: Proof roadmap summarizing the logical connections among lemmas and theorems leading to the main results.

## A  Proofs for Section 3

In this section, we first establish the posterior consistency of the Cox PH model, which serves as a foundation for proving the main theorems in Section 3.

**Lemma A.1.** *(Posterior consistency) Suppose that the grid resolution satisfies $\delta = \kappa n^{-\gamma}$ for $\kappa > 0$ and $\gamma \in (0,1]$, and assumptions (A1)-(A5) hold. If $\gamma < 2/3$, then, for every $\epsilon > 0$, there exist positive constants $C_1$, $C_2$ and $C_3$ depending on ($L$, $B$, $p_{\min}$, $p_{\max}$, $\kappa$, $\underline{\alpha}$, $\overline{\alpha}$, $\rho$, $\epsilon$) such that*

$$\Pi(U^c \mid \mathbf{D}_n) < C_2 \exp(-C_3 n), \quad n \geq C_1, \tag{8}$$

*where*

$$U = \left\{ \theta \in \Theta : \|\mathbf{S}_0 - \mathbf{S}_0^*\|_\infty \vee \|\beta - \beta^*\|_2 < \epsilon \right\}$$

*with $\mathbb{P}_{\theta^*}^n$-probability at least $1 - \exp(-C_3 n)$.*

*If $\gamma \geq 2/3$, then, for every $\epsilon > 0$, there exist positive constants $C_4$, $C_5$ and $C_6$ depending on ($L$, $B$, $p_{\min}$, $p_{\max}$, $\kappa$, $\underline{\alpha}$, $\overline{\alpha}$, $\rho$, $\epsilon$) such that*

$$\Pi(U^c \mid \mathbf{D}_n) < C_5 \exp\left(-C_6 n^{\frac{1}{3}}\right), \quad n \geq C_4,$$

*with $\mathbb{P}_{\theta^*}^n$-probability at least $1 - \exp(-C_6 n^{1/3})$.*

*Remark* A.2. As discussed in Section 4, the conditional distribution of posted prices $\mathbb{Q}(\cdot \mid X)$ is parameterized by the policy. By allowing uniform sampling at a rate of $\eta_l$, defined in (6), the policy constructed in (5) satisfies (A4). Strengthening assumption (A4) to the more restrictive condition $q(g \mid x) \gtrsim n^{-1}(\log n)^{1/2}$ for $g \in \mathcal{G}$ and $x \in \mathcal{X}$ when $\gamma \geq 1/3$ yields the same results as in (8) for all $\gamma \in (0,1]$. In such a case, $\eta_l$ can be adjusted accordingly to satisfy this restrictive condition. However, increasing $\eta_l$ leads to a higher regret due to increased exploration. Therefore, imposing a weak condition, as in (A4), is essential for achieving a tight regret upper bound. For further details, see the proof in Section B.2.

To begin with, for a given parameter $\theta = (\mathbf{S}_0, \beta)$, the joint density $p_\theta$ of $(X_t, P_t, Y_t)$ is expressed as:

$$p_\theta(x, p, y) = \{S_\theta(p|x)\}^y \{1 - S_\theta(p|x)\}^{1-y} q(p|x) p_X(x),$$

for $x \in \mathcal{X}, p \in \mathcal{G}$ and $y \in \{0, 1\}$, where $S_\theta(p \mid x) = S_0(p)^{\exp(x^\top \beta)}$, and $\mathcal{X}$ denotes the support of the covariate $X$. Here, we suppress the dependency of $p_\theta$ on the nuisance parameters, as they do not affect the inference of $\theta$ once an independent prior is used. The log-likelihood function corresponding to $\theta \in \Theta$ for the data $\mathbf{D}_n = \{(X_t, P_t, Y_t)\}_{t=1}^n$ is given by:

$$\ell_n(\theta) = \sum_{t=1}^n \log p_\theta(X_t, P_t, Y_t).$$

## A.1 Proof of Lemma A.1

**Lemma A.3.** *Under the conditions of Lemma A.1, there is an exponentially consistent sequence of tests for*

$$H_0 : \theta = (\mathbf{S}_0^*, \beta^*),$$
$$H_1 : \theta \in \{(\mathbf{S}_0, \beta) \in \mathcal{S}_0 \times \mathbb{R}^d : \|\beta - \beta^*\|_2 \geq \eta\}$$

*for any $\eta > 0$.*

*Proof.* Suppose that $\gamma < 1/3$. Let $\mathcal{T}_n$ denote the set of every disjoint pair of index sets $I_1$ and $I_2$ such that $I_1 \cup I_2 = [K]$. Given an index set $I \subseteq [K]$, we denote the subset of $\mathcal{G}$ corresponding to $I$ by $\mathcal{G}(I) = \{g_k \in \mathcal{G} : k \in I\}$. For each $(I_1, I_2) \in \mathcal{T}_n$, define $\mathcal{S}_0(I_1, I_2) := \{\mathbf{S}_0 = (S_{0,1}, \ldots, S_{0,K}) \in \mathcal{S}_0 : S_{0,i} \geq S_{0,i}^*, S_{0,j} < S_{0,j}^*$ for $i \in I_1$ and $j \in I_2\}$. We define the quadrant $Q_{\mathbf{e}} = \{\mathbf{z} \in \mathbb{R}^d : z_j e_j > 0, \forall j = 1, \ldots, d\}$ for each $\mathbf{e} = (e_1, \ldots, e_d) \in \{-1, 1\}^d$. For $j = 1, \ldots, d$, let $\mathbf{e}^{j,+}, \mathbf{e}^{j,-} \in \{-1, 1\}^d$ denote the vectors where $j$-th element is positive and negative, respectively. Consider the following two groups of hypotheses for each $(I_1, I_2) \in \mathcal{T}_n$ and $\mathbf{e}^{j,+}, \mathbf{e}^{j,-} \in \{-1, 1\}^d$ with $j = 1, \ldots, d$,

$$H_0 : \theta = (\mathbf{S}_0^*, \beta^*), \quad H_1 : \theta \in \Theta_{\mathbf{e}^{j,-}, I_1, I_2} \tag{9}$$
$$H_0 : \theta = (\mathbf{S}_0^*, \beta^*), \quad H_1 : \theta \in \Theta_{\mathbf{e}^{j,+}, I_1, I_2} \tag{10}$$

where $\Theta_{\mathbf{e}^{j,-}, I_1, I_2} = \mathcal{S}_0(I_1, I_2) \times \{\beta \in \mathbb{R}^d : \beta_j^* \geq \beta_j + \xi, \beta - \beta^* \in Q_{\mathbf{e}^{j,-}}\}$, $\Theta_{\mathbf{e}^{j,+}, I_1, I_2} = \mathcal{S}_0(I_1, I_2) \times \{\beta \in \mathbb{R}^d : \beta_j \geq \beta_j^* + \xi, \beta - \beta^* \in Q_{\mathbf{e}^{j,+}}\}$ and $\xi = \eta/\sqrt{d}$.

Fix an arbitrary $j = 1, \ldots, d$, $\mathbf{e}^{j,-}, \mathbf{e}^{j,+} \in \{-1, 1\}^d$ and $(I_1, I_2) \in \mathcal{T}_n$. By Lemma C.4, take a constant $\epsilon > 0$ such that $\mathbb{P}(|X_{t,j}| > \epsilon) > 0$ for all $j = 1, \ldots, d$, given data $D_t = (X_t, P_t, Y_t)$. For the first group of hypotheses (9), define a function $\phi_1 = \max\{\phi_{1,1}, \phi_{1,2}\}$, where $\phi_{1,1}(D_t) = \mathbb{1}\{X_t \in Q_{-\mathbf{e}^{j,-}}, |X_{t,j}| > \epsilon, P_t \in \mathcal{G}(I_1), Y_t = 1\}$ and $\phi_{1,2}(D_t) = \mathbb{1}\{X_t \in Q_{\mathbf{e}^{j,-}}, |X_{t,j}| > \epsilon, P_t \in \mathcal{G}(I_2), Y_t = 0\}$. Under the event $\Omega_1 = \{X_t \in Q_{-\mathbf{e}^{j,-}}, |X_{t,j}| > \epsilon, P_t \in \mathcal{G}(I_1)\}$, for any $\theta = (\mathbf{S}_0, \beta) \in \Theta_{\mathbf{e}^{j,-}, I_1, I_2}$, we have $X_t^\top \beta < X_t^\top \beta^* - \epsilon\xi$. This implies $\exp(X_t^\top \beta) < \exp(X_t^\top \beta^*) \exp(-\epsilon\xi)$. Then, on the event $\Omega_1$, we have

$$S_0(P_t)^{\exp(X_t^\top \beta)} \geq S_0^*(P_t)^{\exp(X_t^\top \beta)} > S_0^*(P_t)^{\exp(X_t^\top \beta^*)\exp(-\epsilon\xi)} > S_0^*(P_t)^{\exp(X_t^\top \beta^*)} + \Delta_1,$$

where the last inequality holds because of the mean value theorem and assumptions (A1), (A2), and (A5), with a positive constant $\Delta_1$ depending on $M_1, M_2, L, B, \epsilon$ and $\xi$. Let $q_n = n^{-(1+\gamma)/2}(\log n)^{1/2}$. By assumption (A4), we have $q(p \mid x) \gtrsim q_n$ for all $x \in \mathcal{X}$ and $p \in \mathcal{G}$ when $\gamma < 1/3$. Then, for any $\theta = (\mathbf{S}_0, \beta) \in \Theta_{\mathbf{e}^{j,-}, I_1, I_2}$, we have

$$\begin{aligned}
\mathbb{E}_\theta [\phi_{1,1}(D_t)] &= \mathbb{E}_{X_t, P_t} \left[ S_0(P_t)^{\exp(X_t^\top \beta)} \mathbb{1}\{\Omega_1\} \right] \\
&> \mathbb{E}_{X_t, P_t} \left[ S_0^*(P_t)^{\exp(X_t^\top \beta^*)} \mathbb{1}\{\Omega_1\} \right] + \mathbb{E}_{X_t, P_t} [\Delta_1 \mathbb{1}\{\Omega_1\}] \\
&\geq \mathbb{E}_{\theta^*} [\phi_{1,1}(D_t)] + C_1 \Delta_1 |I_1| q_n,
\end{aligned}$$

where $C_1$ be a positive constant depending on $\mathbb{P}_X$ and $\epsilon$. Similarly, under the event $\Omega_2 = \{X_t \in Q_{\mathbf{e}^{j,-}}, |X_{t,j}| > \epsilon, P_t \in \mathcal{G}(I_2)\}$, for any $\theta = (\mathbf{S}_0, \beta) \in \Theta_{\mathbf{e}^{j,-}, I_1, I_2}$, we have $\exp(X_t^\top \beta) > \exp(X_t^\top \beta^*) \exp(\epsilon\xi)$. Then, on the event $\Omega_2$, we have $S_0(P_t)^{\exp(X_t^\top \beta)} < S_0^*(P_t)^{\exp(X_t^\top \beta)} < S_0^*(P_t)^{\exp(X_t^\top \beta^*)\exp(\epsilon\xi)} < S_0^*(P_t)^{\exp(X_t^\top \beta^*)} - \Delta_2$, where the last inequality holds because of the mean value theorem and assumptions (A1), (A2), and (A5), with a positive constant $\Delta_2$ depending on $M_1, M_2, L, B, \epsilon$ and $\xi$. Then, for any $\theta = (\mathbf{S}_0, \beta) \in \Theta_{\mathbf{e}^{j,-}, I_1, I_2}$, we have

$$\begin{aligned}
\mathbb{E}_\theta [\phi_{1,2}(D_t)] &= \mathbb{E}_{X_t, P_t} \left[ \left(1 - S_0(P_t)^{\exp(X_t^\top \beta)}\right) \mathbb{1}\{\Omega_2\} \right] \\
&> \mathbb{E}_{X_t, P_t} \left[ \left(1 - S_0^*(P_t)^{\exp(X_t^\top \beta^*)}\right) \mathbb{1}\{\Omega_2\} \right] + \mathbb{E}_{X_t, P_t} [\Delta_2 \mathbb{1}\{\Omega_2\}] \\
&\geq \mathbb{E}_{\theta^*} [\phi_{1,2}(D_t)] + C_2 \Delta_2 |I_2| q_n,
\end{aligned}$$

where $C_2$ be a positive constant depending on $\mathbb{P}_X$ and $\epsilon$. Combining the last two displays, we have

$$
\begin{aligned}
\mathbb{E}_\theta\left[\phi_1(D_t)\right] &= \mathbb{E}_\theta\left[\phi_{1,1}(D_t)\right] + \mathbb{E}_\theta\left[\phi_{1,1}(D_t)\right] \\
&> \mathbb{E}_{\theta^*}\left[\phi_{1,1}(D_t)\right] + \mathbb{E}_{\theta^*}\left[\phi_{1,2}(D_t)\right] + \min\{C_1\Delta_1, C_2\Delta_2\}(|I_1| + |I_2|)q_n \quad (11) \\
&> \mathbb{E}_{\theta^*}\left[\phi_1(D_t)\right] + C_3 n^\gamma q_n,
\end{aligned}
$$

where $C_3$ be a positive constant depending on $C_1, C_2, \Delta_1, \Delta_2, p_{\min}, p_{\max}$ and $\kappa$. Define tests as follows:

$$
\Phi_{\mathbf{e}^{j,-}, I_1, I_2}(\mathbf{D}_n) := \mathbb{1}\left\{\sum_{t=1}^n \phi_1(D_t) > \sum_{t=1}^n (\mathbb{E}_{\theta^*}\left[\phi_1(D_t)\right] + \mathbb{E}_\theta\left[\phi_1(D_t)\right])/2\right\}.
$$

Then, we have

$$
\begin{aligned}
\mathbb{E}_{\theta^*}^n[\Phi_{\mathbf{e}^{j,-}, I_1, I_2}(\mathbf{D}_n)] &= \mathbb{P}_{\theta^*}^n\left(\sum_{t=1}^n(\phi_1(D_t) - \mathbb{E}_{\theta^*}\left[\phi_1(D_t)\right]) > \sum_{t=1}^n(\mathbb{E}_\theta\left[\phi_1(D_t)\right] - \mathbb{E}_{\theta^*}\left[\phi_1(D_t)\right])/2\right) \\
&\leq \mathbb{P}_{\theta^*}^n\left(\sum_{t=1}^n(\phi_1(D_t) - \mathbb{E}_{\theta^*}\left[\phi_1(D_t)\right]) > n(C_3 n^\gamma q_n)/2\right) \\
&\leq \exp\left(-\frac{C_3^2 n^{1+2\gamma} q_n^2}{2}\right),
\end{aligned}
$$

where the first inequality holds by (11) and the last inequality holds by Hoeffding's inequality. On the other hand, applying Hoeffding's inequality to $1 - \phi_1(D_t)$,

$$
\begin{aligned}
&\sup_{\theta \in \Theta_{\mathbf{e}^{j,-}, I_1, I_2}} \mathbb{E}_\theta^n[1 - \Phi_{\mathbf{e}^{j,-}, I_1, I_2}(\mathbf{D}_n)] \\
&= \sup_{\theta \in \Theta_{\mathbf{e}^{j,-}, I_1, I_2}} \mathbb{P}_\theta^n\left(\sum_{t=1}^n((1 - \phi_1(D_t)) - (1 - \mathbb{E}_\theta\left[\phi_1(D_t)\right])) \geq \sum_{t=1}^n(\mathbb{E}_\theta\left[\phi_1(D_t)\right] - \mathbb{E}_{\theta^*}\left[\phi_1(D_t)\right])/2\right) \\
&\leq \sup_{\theta \in \Theta_{\mathbf{e}^{j,-}, I_1, I_2}} \mathbb{P}_\theta^n\left(\sum_{t=1}^n((1 - \phi_1(D_t)) - (1 - \mathbb{E}_\theta\left[\phi_1(D_t)\right])) \geq n(C_3 n^\gamma q_n)/2\right) \\
&\leq \exp\left(-\frac{C_3^2 n^{1+2\gamma} q_n^2}{2}\right),
\end{aligned}
$$

where the first inequality holds by (11).

The construction of tests for the second group of hypotheses (10) is similar. Define the tests as follows:

$$
\Phi_{\mathbf{e}^{j,+}, I_1, I_2}(\mathbf{D}_n) := \mathbb{1}\left\{\sum_{t=1}^n \phi_2(D_t) > \sum_{t=1}^n (\mathbb{E}_{\theta^*}\left[\phi_2(D_t)\right] + \mathbb{E}_\theta\left[\phi_2(D_t)\right])/2\right\},
$$

where $\phi_2 = \max\{\phi_{2,1}, \phi_{2,2}\}$ is a function with $\phi_{2,1}(D_t) = \mathbb{1}\{X_t \in Q_{-\mathbf{e}^{j,+}}, |X_{t,j}| > \epsilon, P_t \in \mathcal{G}(I_1), Y_t = 1\}$ and $\phi_{2,2}(D_t) = \mathbb{1}\{X_t \in Q_{\mathbf{e}^{j,+}}, |X_{t,j}| > \epsilon, P_t \in \mathcal{G}(I_2), Y_t = 0\}$. Similarly, we see that

$$
\mathbb{E}_{\theta^*}^n[\Phi_{\mathbf{e}^{j,+}, I_1, I_2}(\mathbf{D}_n)] \leq \exp\left(-\frac{C_4^2 n^{1+2\gamma} q_n^2}{2}\right),
$$

$$
\sup_{\theta \in \Theta_{\mathbf{e}^{j,-}, I_1, I_2}} \mathbb{E}_\theta^n[1 - \Phi_{\mathbf{e}^{j,-}, I_1, I_2}(\mathbf{D}_n)] \leq \exp\left(-\frac{C_4^2 n^{1+2\gamma} q_n^2}{2}\right),
$$

where $C_4$ be a positive constant.

Note that the union of the sets in the alternative hypotheses (9) and (10) for all $(I_1, I_2) \in \mathcal{T}_n$ and $\mathbf{e}^{j,+}, \mathbf{e}^{j,-} \in \{-1, 1\}^d$ with $j = 1, \ldots, d$ contains $\Theta_\eta := \{(\mathbf{S}_0, \beta) \in \mathcal{S}_0 \times \mathbb{R}^d : \|\beta - \beta^*\|_2 \geq \eta\}$. We set $\Phi_n := \max_{(I_1, I_2) \in \mathcal{T}_n, \mathbf{e}^{j,-}, \mathbf{e}^{j,+} \in \{-1,1\}^d, j \in [d]}\{\Phi_{\mathbf{e}^{j,-}, I_1, I_2} \vee \Phi_{\mathbf{e}^{j,+}, I_1, I_2}\}$, then we have

$$
\begin{aligned}
\mathbb{E}_{\theta^*}^n[\Phi_n(\mathbf{D}_n)] &\leq d2^d 2^K \exp\left(-C_5 n^{1+2\gamma} q_n^2\right) \\
&\leq \exp\left(C_6 d \vee n^\gamma - C_5 n^{1+2\gamma} q_n^2\right), \\
\sup_{\theta \in \Theta_\eta} \mathbb{E}_\theta^n[1 - \Phi_n(\mathbf{D}_n)] &\leq \exp\left(-C_5 n^{1+2\gamma} q_n^2\right),
\end{aligned}
$$

where $C_5 = \min\{C_3^2/2, C_4^2/2\}$ and $C_6$ be a positive constant depending on $p_{\min}, p_{\max}$ and $\kappa$. Then, for fixed $d$, by the definition of $q_n$, we have $\mathbb{E}_{\theta*}^n[\Phi_n(\mathbf{D}_n)] \to 0$ and $\sup_{\theta \in \Theta_n} \mathbb{E}_\theta^n[1 - \Phi_n(\mathbf{D}_n)] \to 0$ as $n \to \infty$. By Lemma D.11 of [16], there exist tests $\Psi_n$ and a constant $C_7 > 0$ such that $\mathbb{E}_{\theta*}^n[\Psi_n(\mathbf{D}_n)] \leq \exp(-C_7 n)$ and $\sup_{\theta \in \Theta_n} \mathbb{E}_\theta^n[1 - \Psi_n(\mathbf{D}_n)] \leq \exp(-C_7 n)$.

Suppose that $\gamma \geq 1/3$. Recall the grid support $\mathcal{G} = \{g_k : k = 1, \ldots, K\}$, where each grid point $g_k$ is defined as $g_k = p_{\min} + k\delta$ with $\delta = \kappa n^{-\gamma}$ for some constant $\kappa > 0$. Let $\epsilon_n = n^{-1/3}$ and $J = \lceil (p_{\max} - p_{\min})/(\kappa \epsilon_n) \rceil$. Define $(k_1, \ldots, k_J)$ as a subsequence of $[K]$ such that $p_{\min} + (j-1)\kappa\epsilon_n < g_{k_j} \leq p_{\min} + (j-1)\kappa\epsilon_n + \delta$ for $j = 1, \ldots, J-1$, and set $k_J = K$.

Let $\mathcal{T}_J$ denote the set of every disjoint pair of sets $I'_1$ and $I'_2$ such that $I'_1 \cup I'_2 = [J]$. For each $(I'_1, I'_2) \in \mathcal{T}_J$, define

$$\mathcal{S}_0(I'_1, I'_2) = \{\mathbf{S}_0 = (S_{0,1}, \ldots, S_{0,K}) \in \mathcal{S}_0 : S_{0,k_i} \geq S_{0,k_i}^*, S_{0,k_j} < S_{0,k_j}^* \text{ for } i \in I'_1 \text{ and } j \in I'_2\}.$$

Consider the following two groups of hypotheses for each $(I'_1, I'_2) \in \mathcal{T}_J$ and $\mathbf{e}^{j,+}, \mathbf{e}^{j,-} \in \{-1, 1\}^d$ with $j = 1, \ldots, d$,

$$H_0 : \theta = (\mathbf{S}_0^*, \beta^*), \quad H_1 : \theta \in \Theta_{\mathbf{e}^{j,-}, I'_1, I'_2} \tag{12}$$

$$H_0 : \theta = (\mathbf{S}_0^*, \beta^*), \quad H_1 : \theta \in \Theta_{\mathbf{e}^{j,+}, I'_1, I'_2}, \tag{13}$$

where $\Theta_{\mathbf{e}^{j,-}, I'_1, I'_2} = \mathcal{S}_0(I'_1, I'_2) \times \{\beta \in \mathbb{R}^d : \beta_j^* \geq \beta_j + \xi, \beta - \beta^* \in Q_{\mathbf{e}^{j,-}}\}$, $\Theta_{\mathbf{e}^{j,+}, I'_1, I'_2} = \mathcal{S}_0(I'_1, I'_2) \times \{\beta \in \mathbb{R}^d : \beta_j \geq \beta_j^* + \xi, \beta - \beta^* \in Q_{\mathbf{e}^{j,+}}\}$ and $\xi = \eta/\sqrt{d}$.

Fix $j = 1, \ldots, d$, $\mathbf{e}^{j,-}, \mathbf{e}^{j,+} \in \{-1, 1\}^d$ and $(I'_1, I'_2) \in \mathcal{T}_J$. Define the index set between $k_i$ and $k_{i+1}$ as $I_i = \{k \in [K] : k_i \leq k \leq k_{i+1}\}$ for $i = 1, \ldots, J-1$. We define partitions $\mathcal{I}_1, \mathcal{I}_2, \mathcal{I}_3$ and $\mathcal{I}_4$ of set $\{I_1, \ldots, I_{J-1}\}$ by

$$\mathcal{I}_1 = \{I_i, i = 1, \ldots, J-1 : S_{0,k_i} \geq S_{0,k_i}^*, S_{0,k_{i+1}} \geq S_{0,k_{i+1}}^*\},$$
$$\mathcal{I}_2 = \{I_i, i = 1, \ldots, J-1 : S_{0,k_i} < S_{0,k_i}^*, S_{0,k_{i+1}} < S_{0,k_{i+1}}^*\},$$
$$\mathcal{I}_3 = \{I_i, i = 1, \ldots, J-1 : S_{0,k_i} < S_{0,k_i}^*, S_{0,k_{i+1}} \geq S_{0,k_{i+1}}^*\},$$
$$\mathcal{I}_4 = \{I_i, i = 1, \ldots, J-1 : S_{0,k_i} \geq S_{0,k_i}^*, S_{0,k_{i+1}} < S_{0,k_{i+1}}^*\}.$$

Note that for any $I \in \mathcal{I}_4$, there exists a unique $k' \in I$ such that $S_{0,k'} \geq S_{0,k'}^*$ and $S_{0,k'+1} < S_{0,k'+1}^*$. Thus, given $I \in \mathcal{I}_4$, we can define $\overline{I} = \{k \in I : k \leq k'\}$ and $\underline{I} = \{k \in I : k > k'\}$. For the first group of hypotheses (12), we define a function $\phi_3 = \max\{\phi_{3,1}, \phi_{3,2}, \phi_{3,3}, \phi_{3,4}, \phi_{3,5}\}$, where

$$\phi_{3,1}(D_t) = \mathbb{1}\{X_t \in Q_{-\mathbf{e}^{j,-}}, |X_{t,j}| > \epsilon, P_t \in \bigcup_{I \in \mathcal{I}_1} \mathcal{G}(I), Y_t = 1\},$$

$$\phi_{3,2}(D_t) = \mathbb{1}\{X_t \in Q_{\mathbf{e}^{j,-}}, |X_{t,j}| > \epsilon, P_t \in \bigcup_{I \in \mathcal{I}_2} \mathcal{G}(I), Y_t = 0\},$$

$$\phi_{3,3}(D_t) = \mathbb{1}\{X_t \in Q_{-\mathbf{e}^{j,-}}, |X_{t,j}| > \epsilon, P_t \in \bigcup_{I \in \mathcal{I}_3} \mathcal{G}(I), Y_t = 1\},$$

$$\phi_{3,4}(D_t) = \mathbb{1}\{X_t \in Q_{-\mathbf{e}^{j,-}}, |X_{t,j}| > \epsilon, P_t \in \bigcup_{I \in \mathcal{I}_4} \mathcal{G}(\overline{I}), Y_t = 1\},$$

$$\phi_{3,5}(D_t) = \mathbb{1}\{X_t \in Q_{\mathbf{e}^{j,-}}, |X_{t,j}| > \epsilon, P_t \in \bigcup_{I \in \mathcal{I}_4} \mathcal{G}(\underline{I}), Y_t = 0\}.$$

Note that under the event $\Omega_{3,1} := \{X_t \in Q_{-\mathbf{e}^{j,-}}, |X_{t,j}| > \epsilon, P_t \in \bigcup_{I \in \mathcal{I}_1} \mathcal{G}(I)\}$, for $\theta = (\mathbf{S}_0, \beta) \in \Theta_{\mathbf{e}^{j,-}, I'_1, I'_2}$, we have $\exp(X_t^\top \beta) < \exp(X_t^\top \beta^*) \exp(-\epsilon\xi)$. For any $I_i \in \mathcal{I}_1$ and $k \in I_i$, we have

$$S_{0,k} - S_{0,k}^* \geq S_{0,k_{i+1}} - S_{0,k_{i+1}}^* + S_{0,k_{i+1}}^* - S_{0,k}^*$$
$$\geq -L_0(\kappa\epsilon_n + \delta)$$
$$\geq -2L_0\kappa\epsilon_n,$$

where the second inequality holds by the definition of $\mathcal{I}_1$ and $I_i$, and the last inequality holds because $\delta \leq \epsilon_n$ for $\gamma \geq 1/3$. Then, on the event $\Omega_{3,1}$, we have

$$
\begin{aligned}
S_0(P_t)^{\exp(X_t^T \beta)} &> S_0(P_t)^{\exp(X_t^T \beta^*) \exp(-\epsilon \xi)} \\
&\geq (S_0^*(P_t) - 2L_0 \kappa \epsilon_n)^{\exp(X_t^T \beta^*) \exp(-\epsilon \xi)} \\
&> S_0^*(P_t)^{\exp(X_t^T \beta^*) \exp(-\epsilon \xi)} - C_8 \epsilon_n \\
&> S_0^*(P_t)^{\exp(X_t^T \beta^*)} + \Delta_1 - C_8 \epsilon_n \\
&> S_0^*(P_t)^{\exp(X_t^T \beta^*)} + \Delta_1/2,
\end{aligned}
$$

where the first inequality holds because $\exp(X_t^\top \beta) < \exp(X_t^\top \beta^*) \exp(-\epsilon \xi)$, the second inequality holds by the preceding display, the third and fourth inequality holds because of the mean value theorem and assumptions (A1), (A2), and (A5), and the last inequality holds for sufficiently large $n$ such that $\epsilon_n < \Delta_1/(2C_8)$. Let $q_n' = n^{-\gamma - 1/3}(\log n)^{1/2}$. By assumption (A4), we have $q(p \mid x) \gtrsim q_n'$ for all $x \in \mathcal{X}$ and $p \in \mathcal{G}$ when $\gamma \geq 1/3$.

Then, for any $\theta = (\mathbf{S}_0, \beta) \in \Theta_{\mathbf{e}^{j,-}, I_1', I_2'}$, we have

$$
\begin{aligned}
\mathbb{E}_\theta[\phi_{3,1}(D_t)] &= \mathbb{E}_{X_t, P_t}\left[ S_0(P_t)^{\exp(X_t^T \beta)} \mathbb{1}\{\Omega_{3,1}\} \right] \\
&> \mathbb{E}_{X_t, P_t}\left[ S_0^*(P_t)^{\exp(X_t^T \beta^*)} \mathbb{1}\{\Omega_{3,1}\} \right] + \Delta_1/2 \cdot \mathbb{E}_{X_t, P_t}[\mathbb{1}\{\Omega_{3,1}\}] \\
&\geq \mathbb{E}_{\theta^*}[\phi_{3,1}(D_t)] + C_9 |\mathcal{I}_1| K q_n'/J,
\end{aligned}
$$

where the second inequality holds by the preceding display, and the last inequality holds with a positive constant $C_9$ because $|I_i| \geq K/J$ for all $i = 1, \ldots, J - 1$. Similarly, there exist positive constants $C_{10}$, $C_{11}$, $C_{12}$ and $C_{13}$ such that

$$
\begin{aligned}
\mathbb{E}_\theta[\phi_{3,2}(D_t)] &> \mathbb{E}_{\theta^*}[\phi_{3,2}(D_t)] + C_{10} |\mathcal{I}_2| K q_n'/J, \\
\mathbb{E}_\theta[\phi_{3,3}(D_t)] &> \mathbb{E}_{\theta^*}[\phi_{3,3}(D_t)] + C_{11} |\mathcal{I}_3| K q_n'/J, \\
\mathbb{E}_\theta[\phi_{3,4}(D_t)] &> \mathbb{E}_{\theta^*}[\phi_{3,4}(D_t)] + C_{12} \sum_{I \in \mathcal{I}_4} |\bar{I}| q_n', \\
\mathbb{E}_\theta[\phi_{3,5}(D_t)] &> \mathbb{E}_{\theta^*}[\phi_{3,5}(D_t)] + C_{13} \sum_{I \in \mathcal{I}_4} |\underline{I}| q_n'.
\end{aligned}
$$

Combining the last two displays, we have

$$
\begin{aligned}
\mathbb{E}_\theta[\phi_3(D_t)] &= \sum_{s=1}^{5} \mathbb{E}_\theta[\phi_{3,s}(D_t)] \\
&> \sum_{s=1}^{5} \mathbb{E}_{\theta^*}[\phi_{3,s}(D_t)] + C_{14}\left( (|\mathcal{I}_1| + |\mathcal{I}_2| + |\mathcal{I}_3|) K q_n'/J + \sum_{I \in \mathcal{I}_4} (|\bar{I}| + |\underline{I}|) q_n' \right) \\
&> \sum_{s=1}^{5} \mathbb{E}_{\theta^*}[\phi_{3,s}(D_t)] + C_{14}(|\mathcal{I}_1| + |\mathcal{I}_2| + |\mathcal{I}_3| + |\mathcal{I}_4|) K q_n'/J \\
&> \mathbb{E}_{\theta^*}[\phi_3(D_t)] + C_{15} n^\gamma q_n', \quad (14)
\end{aligned}
$$

where the second inequality holds because $|\bar{I}| + |\underline{I}| = |I| \geq K/J$ for all $I \in \mathcal{I}_4$, and the last inequality holds because $|\mathcal{I}_1| + |\mathcal{I}_2| + |\mathcal{I}_3| + |\mathcal{I}_4| = J$. Define tests as follows:

$$
\Phi_{\mathbf{e}^{j,-}, I_1', I_2'}(\mathbf{D}_n) := \mathbb{1}\left\{ \sum_{t=1}^{n} \phi_3(D_t) > \sum_{t=1}^{n} (\mathbb{E}_{\theta^*}[\phi_3(D_t)] + \mathbb{E}_\theta[\phi_3(D_t)])/2 \right\}.
$$

By Hoeffding's inequality and (14), we have

$$\mathbb{E}_{\theta^*}^n[\Phi_{\mathbf{e}^{j,-},I_1',I_2'}(\mathbf{D}_n)] \leq \mathbb{P}_{\theta^*}^n\left(\sum_{t=1}^n(\phi_3(D_t) - \mathbb{E}_{\theta^*}[\phi_3(D_t)]) > n(C_{15}n^\gamma q_n')/2\right)$$

$$\leq \exp\left(-\frac{C_{15}^2 n^{1+2\gamma} q_n'^2}{2}\right),$$

$$\sup_{\theta \in \Theta_{\mathbf{e}^{j,-},I_1',I_2'}} \mathbb{E}_{\theta^*}^n[1 - \Phi_{\mathbf{e}^{j,-},I_1',I_2'}(\mathbf{D}_n)] \leq \exp\left(-\frac{C_{15}^2 n^{1+2\gamma} q_n'^2}{2}\right).$$

The construction of tests for the second group of hypotheses (13) is similar. Define the tests as follows:

$$\Phi_{\mathbf{e}^{j,+},I_1',I_2'}(\mathbf{D}_n) := \mathbb{1}\left\{\sum_{t=1}^n \phi_4(D_t) > \sum_{t=1}^n(\mathbb{E}_{\theta^*}[\phi_4(D_t)] + \mathbb{E}_\theta[\phi_4(D_t)])/2\right\},$$

where $\phi_4 = \max\{\phi_{4,1}, \phi_{4,2}, \phi_{4,3}, \phi_{4,4}, \phi_{4,5}\}$ is a function with

$$\phi_{4,1}(D_t) = \mathbb{1}\{X_t \in Q_{-\mathbf{e}^{j,+}}, |X_{t,j}| > \epsilon, P_t \in \bigcup_{I \in \mathcal{I}_1} \mathcal{G}(I), Y_t = 1\},$$

$$\phi_{4,2}(D_t) = \mathbb{1}\{X_t \in Q_{\mathbf{e}^{j,+}}, |X_{t,j}| > \epsilon, P_t \in \bigcup_{I \in \mathcal{I}_2} \mathcal{G}(I), Y_t = 0\},$$

$$\phi_{4,3}(D_t) = \mathbb{1}\{X_t \in Q_{-\mathbf{e}^{j,+}}, |X_{t,j}| > \epsilon, P_t \in \bigcup_{I \in \mathcal{I}_3} \mathcal{G}(I), Y_t = 1\},$$

$$\phi_{4,4}(D_t) = \mathbb{1}\{X_t \in Q_{-\mathbf{e}^{j,+}}, |X_{t,j}| > \epsilon, P_t \in \bigcup_{I \in \mathcal{I}_4} \mathcal{G}(\overline{I}), Y_t = 1\},$$

$$\phi_{4,5}(D_t) = \mathbb{1}\{X_t \in Q_{\mathbf{e}^{j,+}}, |X_{t,j}| > \epsilon, P_t \in \bigcup_{I \in \mathcal{I}_4} \mathcal{G}(\underline{I}), Y_t = 0\}.$$

Similarly, we see that

$$\mathbb{E}_{\theta^*}^n[\Phi_{\mathbf{e}^{j,+},I_1',I_2'}(\mathbf{D}_n)] \leq \exp\left(-\frac{C_{16}^2 n^{1+2\gamma} q_n'^2}{2}\right),$$

$$\sup_{\theta \in \Theta_{\mathbf{e}^{j,+},I_1',I_2'}} \mathbb{E}_\theta^n[1 - \Phi_{\mathbf{e}^{j,+},I_1',I_2'}(\mathbf{D}_n)] \leq \exp\left(-\frac{C_{16}^2 n^{1+2\gamma} q_n'^2}{2}\right),$$

where $C_{16}$ be a positive constant.

Note that the union of the sets in the alternative hypotheses (12) and (13) for all $(I_1', I_2') \in \mathcal{T}_J$ and $\mathbf{e}^{j,+}, \mathbf{e}^{j,-} \in \{-1,1\}^d$ with $j = 1, \ldots, d$ contains $\Theta_\eta := \{(\mathbf{S}_0, \beta) \in \mathcal{S}_0 \times \mathbb{R}^d : \|\beta - \beta^*\|_2 \geq \eta\}$. We set $\Phi_n' := \max_{(I_1',I_2') \in \mathcal{T}_J, \mathbf{e}^{j,-}, \mathbf{e}^{j,+} \in \{-1,1\}^d, j \in [d]}\{\Phi_{\mathbf{e}^{j,-},I_1',I_2'} \vee \Phi_{\mathbf{e}^{j,+},I_1',I_2'}\}$, then we have

$$\mathbb{E}_{\theta^*}^n[\Phi_n'(\mathbf{D}_n)] \leq d2^d 2^J \exp\left(-C_{17} n^{1+2\gamma} q_n'^2\right)$$

$$\leq \exp\left(C_{18} d \vee n^{\frac{1}{3}} - C_{17} n^{1+2\gamma} q_n'^2\right),$$

$$\sup_{\theta \in \Theta_\eta} \mathbb{E}_\theta^n[1 - \Phi_n'(\mathbf{D}_n)] \leq \exp\left(-C_{17} n^{1+2\gamma} q_n'^2\right),$$

where $C_{17} = \min\{C_{15}^2/2, C_{16}^2/2\}$ and $C_{18}$ be a positive constant depending on $p_{\min}, p_{\max}$ and $\kappa$. Then, for fixed $d$, by the definition of $q_n'$, we have $\mathbb{E}_{\theta^*}^n[\Phi_n'(\mathbf{D}_n)] \to 0$ and $\sup_{\theta \in \Theta_\eta} \mathbb{E}_\theta^n[1 - \Phi_n'(\mathbf{D}_n)] \to 0$ as $n \to \infty$. By Lemma D.11 of [16], there exist tests $\Psi_n'$ and a constant $C_{19} > 0$ such that $\mathbb{E}_{\theta^*}^n[\Psi_n'(\mathbf{D}_n)] \leq \exp(-C_{19}n)$ and $\sup_{\theta \in \Theta_\eta} \mathbb{E}_\theta^n[1 - \Psi_n'(\mathbf{D}_n)] \leq \exp(-C_{19}n)$. The proof is then complete. □

**Lemma A.4.** *Suppose that the grid resolution satisfies $\delta = \kappa n^{-\gamma}$ for $\kappa > 0$ and $\gamma \in (0, 2/3)$, and assumptions (A1)-(A5) hold. Then, there is an exponentially consistent sequence of tests for*

$$H_0 : \theta = (\mathbf{S}_0^*, \beta^*),$$
$$H_1 : \theta \in \{(\mathbf{S}_0, \beta) \in \mathcal{S}_0 \times \mathbb{R}^d : \|\mathbf{S}_0 - \mathbf{S}_0^*\|_\infty \geq \eta_1, \|\beta - \beta^*\|_2 < \eta_2\}$$

*for any $\eta_1 > 0$ and sufficiently small $\eta_2 > 0$.*

*Proof.* There exist constants $M_1, M_2 \in (0, 1)$ such that $M_1 \leq S_0^*(v) \leq M_2$ for any $v \in [p_{\min}, p_{\max}]$ under assumption (A5). We choose $\eta_1$ to be less than $\min\{1 - M_2, M_1\}$ to ensure that $\{\mathbf{S}_0 \in \mathcal{S}_0 : \|\mathbf{S}_0 - \mathbf{S}_0^*\|_\infty \geq \eta_1\} \neq \emptyset$. Consider the following two groups of hypotheses for each $k \in [K]$,

$$H_0 : \theta = (\mathbf{S}_0^*, \beta^*), \quad H_1 : \theta \in \Theta_{k,1} \tag{15}$$
$$H_0 : \theta = (\mathbf{S}_0^*, \beta^*), \quad H_1 : \theta \in \Theta_{k,2} \tag{16}$$

where $\Theta_{k,1} = \{(\mathbf{S}_0, \beta) \in \mathcal{S}_0 \times \mathbb{R}^d : S_{0,k} \geq S_{0,k}^* + \eta_1, \|\beta - \beta^*\|_2 < \eta_2\}$ and $\Theta_{k,2} = \{(\mathbf{S}_0, \beta) \in \mathcal{S}_0 \times \mathbb{R}^d : S_{0,k} \leq S_{0,k}^* - \eta_1, \|\beta - \beta^*\|_2 < \eta_2\}$.

Fix an arbitrary $k \in [K]$. For the first group of hypotheses (15), define a function $\phi_1(D_t) = \mathbb{1}\{P_t = g_k, Y_t = 1\}$. For any $\beta$ such that $\|\beta - \beta^*\|_2 < \eta_2$, by the Cauchy-Schwartz inequality and the assumption (A2), $|X_t^\top(\beta - \beta^*)| \leq \|X_t\|_2 \|\beta - \beta^*\|_2 < L\eta_2$ almost surely. This implies $\exp(X_t^\top \beta) < \exp(X_t^\top \beta^*) \exp(L\eta_2)$. Then, for any $\theta = (\mathbf{S}_0, \beta) \in \Theta_{k,1}$, we have

$$S_{0,k}^{\exp(X_t^\top \beta)} > (S_{0,k}^* + \eta_1)^{\exp(L\eta_2)\exp(X_t^\top \beta^*)}.$$

It is easy to show that there exists a positive constant $C_1$ depending on $M_1, M_2, L$ and $\eta_1$ such that $C_1 \leq \log\log((S_{0,k}^* + \eta_1/2)^{-1}) - \log\log((S_{0,k}^* + \eta_1)^{-1})$ for any $S_{0,k}^* \in [M_1, M_2]$. If we choose a sufficiently small $\eta_2$ such that $L\eta_2 \leq C_1$, we have $(S_{0,k}^* + \eta_1)^{\exp(L\eta_2)} \geq S_{0,k}^* + \eta_1/2$. Combining this with the previous display,

$$S_{0,k}^{\exp(X_t^\top \beta)} > \left(S_{0,k}^* + \frac{\eta_1}{2}\right)^{\exp(X_t^\top \beta^*)}$$
$$\geq S_{0,k}^{*\exp(X_t^\top \beta^*)} + C_2,$$

where the last inequality holds with a positive constant $C_2$ depending on $M_1, M_2, L, B$ and $\eta_1$ by assumptions (A1), (A2), (A5) and the mean value theorem. Then, for any $\theta = (\mathbf{S}_0, \beta) \in \Theta_{k,1}$, we have

$$\mathbb{E}_\theta[\phi_1(D_t)] = \mathbb{E}_{X_t, P_t}\left[S_0(P_t)^{\exp(X_t^\top \beta)} \mathbb{1}\{P_t = g_k\}\right]$$
$$> \mathbb{E}_{X_t, P_t}\left[S_0^*(P_t)^{\exp(X_t^\top \beta^*)} \mathbb{1}\{P_t = g_k\}\right] + C_2 q(g_k) \tag{17}$$
$$= \mathbb{E}_{\theta^*}[\phi_1(D_t)] + C_2 q(g_k).$$

In addition, for either $\theta \in \Theta_{k,1}$ or $\theta = \theta^*$, we have

$$\mathrm{Var}_\theta(\phi_1(D_t) - \mathbb{E}_\theta(\phi_1(D_t))) = \mathbb{E}_\theta[\phi_1(D_t)](1 - \mathbb{E}_\theta[\phi_1(D_t)])$$
$$\leq \mathbb{E}_\theta[\phi_1(D_t)]$$
$$= \mathbb{E}_{X_t, P_t}\left[S_0(P_t)^{\exp(X_t^\top \beta)} \mathbb{1}\{P_t = g_k\}\right] \tag{18}$$
$$< q(g_k),$$

where $\mathrm{Var}_\theta$ is the variance with respect to the distribution $\mathbb{P}_\theta$. Define tests as follows:

$$\Phi_{k,1}(\mathbf{D}_n) := \mathbb{1}\left\{\sum_{t=1}^n \phi_1(D_t) > \sum_{t=1}^n (\mathbb{E}_{\theta^*}[\phi_1(D_t)] + \mathbb{E}_\theta[\phi_1(D_t)])/2\right\}.$$

Then, we have

$$\mathbb{E}_{\theta^*}^n[\Phi_{k,1}(\mathbf{D}_n)] = \mathbb{P}_{\theta^*}^n\left(\sum_{t=1}^n(\phi_1(D_t) - \mathbb{E}_{\theta^*}[\phi_1(D_t)]) > \sum_{t=1}^n(\mathbb{E}_\theta[\phi_1(D_t)] - \mathbb{E}_{\theta^*}[\phi_1(D_t)])/2\right)$$

$$\leq \mathbb{P}_{\theta^*}^n\left(\sum_{t=1}^n(\phi_1(D_t) - \mathbb{E}_{\theta^*}[\phi_1(D_t)]) > \frac{C_2}{2}nq(g_k)\right)$$

$$\leq \exp\left(-\frac{(C_2^2/8)n^2q(g_k)^2}{\sum_{t=1}^n \mathrm{Var}_{\theta^*}(\phi_1(D_t) - \mathbb{E}_{\theta^*}[\phi_1(D_t)]) + (C_2/6)nq(g_k)}\right)$$

$$\leq \exp\left(-C_3nq(g_k)\right)$$

$$\leq \exp\left(-C_4nq_n\right),$$

(19)

where the first inequality holds by (17), the second inequality holds by Bernstein inequality, the third inequality holds by (18) with a positive constant $C_3 = C_2^2/(8(1 + C_2/6))$, and the last inequality holds because $q(p) \gtrsim q_n$ with

$$q_n = \begin{cases} n^{-\frac{1+\gamma}{2}}(\log n)^{\frac{1}{2}} & \text{if } \gamma < \frac{1}{3}, \\ n^{-\gamma-\frac{1}{3}}(\log n)^{\frac{1}{2}} & \text{if } \gamma \geq \frac{1}{3}, \end{cases}$$

under the assumption (A4). On the other hand, applying (17), (18) and Bernstein inequality to $1 - \phi_1(D_t)$, we have

$$\sup_{\theta\in\Theta_{k,1}} \mathbb{E}_\theta^n[1 - \Phi_{k,1}(\mathbf{D}_n)]$$

$$= \sup_{\theta\in\Theta_{k,1}} \mathbb{P}_\theta^n\left(\sum_{t=1}^n((1 - \phi_1(D_t)) - (1 - \mathbb{E}_\theta[\phi_1(D_t)])) \geq \sum_{t=1}^n(\mathbb{E}_\theta[\phi_1(D_t)] - \mathbb{E}_{\theta^*}[\phi_1(D_t)])/2\right)$$

$$\leq \sup_{\theta\in\Theta_{k,1}} \mathbb{P}_\theta^n\left(\sum_{t=1}^n((1 - \phi_1(D_t)) - (1 - \mathbb{E}_\theta[\phi_1(D_t)])) \geq \frac{C_2}{2}nq(g_k)\right)$$

$$\leq \sup_{\theta\in\Theta_{k,1}} \exp\left(-\frac{(C_2^2/8)n^2q(g_k)^2}{\sum_{t=1}^n \mathrm{Var}_\theta(\mathbb{E}_\theta[\phi_1(D_t)] - \phi_1(D_t)) + (C_2/6)nq(g_k)}\right)$$

$$\leq \exp\left(-C_3nq(g_k)\right)$$

$$\leq \exp\left(-C_4nq_n\right).$$

(20)

The construction of tests for the second group of hypotheses (16) is similar. Define the tests as follows:

$$\Phi_{k,2}(\mathbf{D}_n) := \mathbb{1}\left\{\sum_{t=1}^n \phi_2(D_t) > \sum_{t=1}^n(\mathbb{E}_{\theta^*}[\phi_2(D_t)] + \mathbb{E}_\theta[\phi_2(D_t)])/2\right\},$$

where $\phi_2(D_t) = \mathbb{1}\{P_t = g_k, Y_t = 0\}$ is a function. Similarly, we see that there exists a positive constant $C_5$ depending on $M_1, M_2, L, B, \eta_1$ such that

$$\mathbb{E}_{\theta^*}^n[\Phi_{k,2}(\mathbf{D}_n)] \leq \exp(-C_5nq_n), \quad \sup_{\theta\in\Theta_{k,2}} \mathbb{E}_\theta^n[1 - \Phi_{k,2}(\mathbf{D}_n)] \leq \exp(-C_5nq_n). \quad (21)$$

Note that the union of the sets in the alternative hypotheses (15) and (16) for all $k = 1, \ldots, K$ contains $\Theta_{\eta_1,\eta_2} := \{(\mathbf{S}_0, \beta) \in \mathcal{S}_0 \times \mathbb{R}^d : \|\mathbf{S}_0 - \mathbf{S}_0^*\|_\infty \geq \eta_1, \|\beta - \beta^*\|_2 < \eta_2\}$. We set $\Phi_n := \max_{k\in[K]}\{\Phi_{k,1} \vee \Phi_{k,2}\}$. Combining (19), (20) and (21), we have

$$\mathbb{E}_{\theta^*}^n[\Phi_n(\mathbf{D}_n)] \leq K\exp\left(-C_6nq_n\right)$$

$$= \exp\left(\log K - C_6nq_n\right),$$

$$\sup_{\theta\in\Theta_{\eta_1,\eta_2}} \mathbb{E}_\theta^n[1 - \Phi_n(\mathbf{D}_n)] \leq \exp\left(-C_6nq_n\right),$$

where $C_6 = \min\{C_4, C_5\}$. By the definition of $q_n$, we have $\mathbb{E}_{\theta^*}^n[\Phi_n(\mathbf{D}_n)] \to 0$ and $\sup_{\theta\in\Theta_{\eta_1,\eta_2}} \mathbb{E}_\theta^n[1 - \Phi_n(\mathbf{D}_n)] \to 0$ as $n \to \infty$ when $\gamma < 2/3$. By Lemma D.11 of [16], there exist tests $\Psi_n$ and a constant $C_7 > 0$ such that $\mathbb{E}_{\theta^*}^n[\Psi_n(\mathbf{D}_n)] \leq \exp(-C_7n)$ and $\sup_{\theta\in\Theta_{\eta_1,\eta_2}} \mathbb{E}_\theta^n[1 - \Psi_n(\mathbf{D}_n)] \leq \exp(-C_7n)$. The proof is then complete. □

**Lemma A.5.** *Suppose that the grid resolution satisfies $\delta = \kappa n^{-\gamma}$ for $\kappa > 0$ and $\gamma \in [1/3, 1]$, and assumptions (A1)-(A5) hold. Let $\epsilon = n^{-1/3}$ and $J = \lceil (p_{\max} - p_{\min})/(\kappa\epsilon) \rceil$. Define $(k_1, \ldots, k_J)$ as a subsequence of $[K]$ such that $p_{\min} + (j-1)\kappa\epsilon < g_{k_j} \leq p_{\min} + (j-1)\kappa\epsilon + \delta$ for $j = 1, \ldots, J-1$, and set $k_J = K$. Then, there is an exponentially consistent sequence of tests for*

$$H_0 : \theta = (\mathbf{S}_0^*, \beta^*),$$
$$H_1 : \theta \in \{(\mathbf{S}_0, \beta) \in \mathcal{S}_0 \times \mathbb{R}^d : \max_{2 \leq j \leq J-1} |S_{0,k_j} - S_{0,k_j}^*| \geq \eta_1, \|\beta - \beta^*\|_2 < \eta_2\}$$

*for any $\eta_1 > 0$ and sufficiently small $\eta_2 > 0$.*

*Proof.* We consider the following two groups of hypotheses for each $j = 2, \ldots, J-1$,

$$H_0 : \theta = (\mathbf{S}_0^*, \beta^*), \quad H_1 : \theta \in \Theta_{j,1} \tag{22}$$
$$H_0 : \theta = (\mathbf{S}_0^*, \beta^*), \quad H_1 : \theta \in \Theta_{j,2} \tag{23}$$

where $\Theta_{j,1} = \{(\mathbf{S}_0, \beta) \in \mathcal{S}_0 \times \mathbb{R}^d : S_{0,k_j} \geq S_{0,k_j}^* + \eta_1, \|\beta - \beta^*\|_2 < \eta_2\}$ and $\Theta_{j,2} = \{(\mathbf{S}_0, \beta) \in \mathcal{S}_0 \times \mathbb{R}^d : S_{0,k_j} \leq S_{0,k_j}^* - \eta_1, \|\beta - \beta^*\|_2 < \eta_2\}$. Define the index set between $k_j$ and $k_{j+1}$ as $I_j = \{k \in [K] : k_j \leq k \leq k_{j+1}\}$ for $j = 1, \ldots, J-1$. Given an index set $I \subseteq [K]$, we denote the subset of $\mathcal{G}$ corresponding to $I$ by $\mathcal{G}(I) = \{g_k \in \mathcal{G} : k \in I\}$.

Fix $j = 2, \ldots, J-1$. For the first group of hypotheses (22), define a function $\phi_1(D_t) = \mathbb{1}\{P_t \in \mathcal{G}(I_{j-1}), Y_t = 1\}$. For any $\theta \in \Theta_{j,1}$ and $k \in I_{j-1}$, we have

$$S_{0,k} - S_{0,k}^* \geq S_{0,k_j} - S_{0,k_j}^* + S_{0,k_j}^* - S_{0,k}^*$$
$$\geq \eta_1 - L_0 |g_{k_j} - g_k|$$
$$\geq \eta_1 - L_0(\kappa\epsilon + \delta)$$
$$\geq \eta_1 - 2L_0\kappa\epsilon$$
$$\geq \frac{\eta_1}{2},$$

where the second inequality holds because $\theta \in \Theta_{j,1}$ and $S_0^*$ is $L_0$-Lipschitz continuous under the assumption (A5), the third inequality holds by the definition of $(k_1, \ldots, k_J)$, the fourth inequality holds because $\delta \leq \kappa\epsilon$ when $\gamma \geq 1/3$, and the last inequality holds for sufficiently large $n$ such that $\epsilon \leq \eta_1/(4L_0\kappa)$. By a similar argument as the proof in Lemma A.4, for a sufficiently small $\eta_2$, there exists a positive constant $C_1$ depending on $M_1$, $M_2$, $L$, $B$ and $\eta_1$ such that for $\theta = (\mathbf{S}_0, \beta) \in \Theta_{j,1}$ and $k \in I_{j-1}$,

$$S_{0,k}^{\exp(X_t^\top \beta)} > S_{0,k}^{*\exp(X_t^\top \beta^*)} + C_1.$$

Then, for any $\theta = (\mathbf{S}_0, \beta) \in \Theta_{j,1}$, we have

$$\mathbb{E}_\theta[\phi_1(D_t)] = \mathbb{E}_{X_t, P_t}\left[S_0(P_t)^{\exp(X_t^\top \beta)} \mathbb{1}\{P_t \in \mathcal{G}(I_{j-1})\}\right]$$
$$> \mathbb{E}_{X_t, P_t}\left[S_0^*(P_t)^{\exp(X_t^\top \beta^*)} \mathbb{1}\{P_t \in \mathcal{G}(I_{j-1})\}\right] + C_1 \sum_{k \in I_{j-1}} q(g_k)$$
$$= \mathbb{E}_{\theta^*}[\phi_1(D_t)] + C_1 \sum_{k \in I_{j-1}} q(g_k),$$

In addition, for either $\theta \in \Theta_{j,1}$ or $\theta = \theta^*$, we have

$$\mathrm{Var}_\theta(\phi_1(D_t) - \mathbb{E}_\theta(\phi_1(D_t)) = \mathbb{E}_\theta[\phi_1(D_t)](1 - \mathbb{E}_\theta[\phi_1(D_t)])$$
$$\leq \mathbb{E}_\theta[\phi_1(D_t)]$$
$$= \mathbb{E}_{X_t, P_t}\left[S_0(P_t)^{\exp(X_t^\top \beta)} \mathbb{1}\{P_t \in \mathcal{G}(I_{j-1})\}\right]$$
$$< \sum_{k \in I_{j-1}} q(g_k).$$

Define tests as follows:

$$\Phi_{j,1}(\mathbf{D}_n) := \mathbb{1}\left\{\sum_{t=1}^n \phi_1(D_t) > \sum_{t=1}^n (\mathbb{E}_{\theta^*}[\phi_1(D_t)] + \mathbb{E}_\theta[\phi_1(D_t)])/2\right\}.$$

Combining the last three displays, by Bernstein inequality, we have

$$\mathbb{E}_{\theta^*}^n[\Phi_{j,1}(\mathbf{D}_n)] \leq \mathbb{P}_{\theta^*}^n \left( \sum_{t=1}^n (\phi_1(D_t) - \mathbb{E}_{\theta^*}[\phi_1(D_t)]) > \frac{C_1}{2} n \sum_{k \in I_{j-1}} q(g_k) \right)$$

$$\leq \exp\left( -\frac{(C_1^2/8)n^2(\sum_{k \in I_{j-1}} q(g_k))^2}{\sum_{t=1}^n \mathrm{Var}_{\theta^*}(\phi_1(D_t) - \mathbb{E}_{\theta^*}[\phi_1(D_t)]) + (C_1/6)n \sum_{k \in I_{j-1}} q(g_k)} \right)$$

$$\leq \exp\left( -C_2 n \sum_{k \in I_{j-1}} q(g_k) \right)$$

$$\leq \exp\left( -C_3 n |I_{j-1}| q_n \right)$$

$$\leq \exp\left( -C_3 n^{\gamma + \frac{2}{3}} q_n \right), \tag{24}$$

where $C_2 = C_1^2/(8(1 + C_1/6))$ be a positive constant, the fourth inequalith holds with a positive constant $C_3$ depending on $C_2$ and $q(\cdot)$ because $q(p) \gtrsim q_n$ for any $p \in \mathcal{G}$ with $q_n = n^{-\gamma - 1/3}(\log n)^{1/2}$ under the assumption (A4), and the last inequality holds because $|I_j| \geq K/J \geq n^{\gamma - 1/3}$ for any $j = 1, \dots, J - 1$. Similarly, we have

$$\sup_{\theta \in \Theta_{j,1}} \mathbb{E}_\theta^n[1 - \Phi_{j,1}(\mathbf{D}_n)] \leq \exp\left( -C_3 n^{\gamma + \frac{2}{3}} q_n \right), \tag{25}$$

The construction of tests for the second group of hypotheses (23) is similar. Define the tests as follows:

$$\Phi_{j,2}(\mathbf{D}_n) := \mathbb{1}\left\{ \sum_{t=1}^n \phi_2(D_t) > \sum_{t=1}^n (\mathbb{E}_{\theta^*}[\phi_2(D_t)] + \mathbb{E}_\theta[\phi_2(D_t)])/2 \right\}, \tag{26}$$

where $\phi_2(D_t) = \mathbb{1}\{P_t \in \mathcal{G}(I_j), Y_t = 0\}$ is a function. By a similar argument as the preceding, for any $\theta \in \Theta_{j,2}$, $k \in I_j$, and for sufficiently large $n$ such that $\epsilon \leq \eta_1/(4L_0\kappa)$, we have

$$S_{0,k}^* - S_{0,k} \geq S_{0,k}^* - S_{0,k_j}^* + S_{0,k_j}^* - S_{0,k_j} \geq -L_0|g_k - g_{k_j}| + \eta_1 \geq \frac{\eta_1}{2}.$$

By a similar argument as the proof in Lemma A.4, for a sufficiently small $\eta_2$, there exists a positive constant $C_4$ depending on $M_1$, $M_2$, $L$, $B$ and $\eta_1$ such that for any $\theta \in \Theta_{j,2}$,

$$\mathbb{E}_\theta[\phi_2(D_t)] > \mathbb{E}_{\theta^*}[\phi_2(D_t)] + C_4 \sum_{k \in I_j} q(g_k).$$

In addition, for either $\theta \in \Theta_{j,2}$ or $\theta \in \theta^*$, we have

$$\mathrm{Var}_\theta(\phi_2(D_t) - \mathbb{E}_\theta(\phi_2(D_t))) \leq \mathbb{E}_\theta[\phi_2(D_t)]$$

$$= \mathbb{E}_{X_t, P_t}\left[ \left( 1 - S_0(P_t)^{\exp(X_t^\top \beta)} \right) \mathbb{1}\{P_t \in \mathcal{G}(I_j)\} \right]$$

$$< \sum_{k \in I_j} q(g_k).$$

Combining the last two displays, by Bernstein inequality, there exists a positive constant $C_5$ depending on $C_4$ and $q(\cdot)$ such that

$$\mathbb{E}_{\theta^*}^n[\Phi_{j,2}(\mathbf{D}_n)] \leq \exp\left( -C_5 n^{\gamma + \frac{2}{3}} q_n \right), \quad \sup_{\theta \in \Theta_{j,2}} \mathbb{E}_\theta^n[1 - \Phi_{j,2}(\mathbf{D}_n)] \leq \exp\left( -C_5 n^{\gamma + \frac{2}{3}} q_n \right). \tag{27}$$

Note that the union of the sets in the alternative hypotheses (22) and (23) for all $j = 2, \dots, J - 1$ contains $\Theta_{\eta_1, \eta_2} := \{(\mathbf{S}_0, \beta) \in \mathcal{S}_0 \times \mathbb{R}^d : \max_{2 \leq j \leq J-1} |S_{0,k_j} - S_{0,k_j}^*| \geq \eta_1, \|\beta - \beta^*\|_2 < \eta_2\}$. We set $\Phi_n := \max_{2 \leq j \leq J-1}\{\Phi_{j,1} \vee \Phi_{j,2}\}$. Combining (24), (25) and (27), we have

$$\mathbb{E}_{\theta^*}^n[\Phi_n(\mathbf{D}_n)] \leq J \exp\left( -C_6 n^{\gamma + \frac{2}{3}} q_n \right)$$

$$= \exp\left( \log J - C_6 n^{\gamma + \frac{2}{3}} q_n \right),$$

$$\sup_{\theta \in \Theta_{\eta_1, \eta_2}} \mathbb{E}_\theta^n[1 - \Phi_n(\mathbf{D}_n)] \leq \exp\left( -C_6 n^{\gamma + \frac{2}{3}} q_n \right),$$

where $C_6 = \min\{C_3, C_5\}$. By the definition of $q_n$, we have $\mathbb{E}_{\theta^*}^n[\Phi_n(\mathbf{D}_n)] \to 0$ and $\sup_{\theta \in \Theta_{\eta_1, \eta_2}} \mathbb{E}_\theta^n[1 - \Phi_n(\mathbf{D}_n)] \to 0$ as $n \to \infty$ when $\gamma \geq 1/3$. By Lemma D.11 of [16], there exist tests $\Psi_n$ and a constant $C_7 > 0$ such that $\mathbb{E}_{\theta^*}^n[\Psi_n(\mathbf{D}_n)] \leq \exp(-C_7 n)$ and $\sup_{\theta \in \Theta_{\eta_1, \eta_2}} \mathbb{E}_\theta^n[1 - \Psi_n(\mathbf{D}_n)] \leq \exp(-C_7 n)$. The proof is then complete.

$\square$

*Proof of Lemma A.1.* We proceed with the proof by considering two separate cases: $\gamma < 2/3$ and $\gamma \geq 2/3$. First, we suppose that $\gamma < 2/3$. Let $\epsilon_0 > 0$ be a constant to be chosen later, and define $\Theta_{\epsilon_0} = \{(\mathbf{S}_0, \beta) \in \Theta : \|\mathbf{\Lambda}_0 - \mathbf{\Lambda}_0^*\|_\infty \vee \|\beta - \beta^*\|_2 \leq \epsilon_0\}$. Here, $\mathbf{\Lambda}_0 = (\Lambda_{0,1}, \ldots, \Lambda_{0,K})$ and $\mathbf{\Lambda}_0^* = (\Lambda_{0,1}^*, \ldots, \Lambda_{0,K}^*)$ are $K$-dimensional vectors corresponding to $\mathbf{S}_0$ and $\mathbf{S}_0^*$, respectively, such that $\Lambda_{0,k} = -\log S_{0,k}$ and $\Lambda_{0,k}^* = -\log S_{0,k}^*$ for $k = 1, \ldots, K$. The log-likelihood ratio satisfies

$$\log \frac{p_{\theta^*}}{p_\theta}(x, p, y) = y \log \frac{H_{\theta^*}(x, p)}{H_\theta(x, p)} + (1 - y) \log \frac{1 - H_{\theta^*}(x, p)}{1 - H_\theta(x, p)}$$

$$\leq \max\left\{\log \frac{H_{\theta^*}(x, p)}{H_\theta(x, p)}, \log \frac{1 - H_{\theta^*}(x, p)}{1 - H_\theta(x, p)}\right\}.$$

By assumption (A5), there exist constants $M_1$ and $M_2$ such that $0 < M_1 \leq S_0^*(p_{\max}) < S_0^*(p_{\min}) \leq M_2 < 1$. Note that for $\theta \in \Theta_{\epsilon_0}$, where $\epsilon_0 \leq (M_1 \wedge (1 - M_2)/2) \wedge B$, we have $S_0(v) \in [M_1/2, (1 + M_2)/2]$ for any $v \in [p_{\min}, p_{\max}]$, and $\|\beta\| \leq 2B$ under assumptions (A1) and (A5). Furthermore, by assumption (A2), both $H_{\theta^*}(x, p)$ and $H_\theta(x, p)$ are bounded away from 0 and 1 for any $x \in \mathcal{X}, p \in \mathcal{G}$ and $\theta \in \Theta_{\epsilon_0}$. Since $|\log p - \log q| \leq |p - q| \max\{p^{-1}, q^{-1}\}$ for any $0 < p, q < 1$, we have

$$\left\|\log \frac{p_{\theta^*}}{p_\theta}\right\|_\infty \leq C_0 \|H_{\theta^*} - H_\theta\|_\infty, \tag{28}$$

where $C_0$ is a positive constant depending on $M_1, M_2, L$ and $B$. In addition, by Lemma C.2, there exist positive constants $c_1$ and $c_2$, depending on $M_1, M_2, L$ and $B$, such that for any $x \in \mathcal{X}$ and $p \in \mathcal{G}$,

$$|H_{\theta^*}(x, p) - H_\theta(x, p)| \leq c_1 \|\mathbf{S}_0 - \mathbf{S}_0^*\|_\infty + c_2 \|\beta - \beta^*\|_2$$

$$\leq c_1 \|\mathbf{\Lambda}_0 - \mathbf{\Lambda}_0^*\|_\infty + c_2 \|\beta - \beta^*\|_2,$$

where the last inequality holds because $\|\mathbf{S}_0 - \mathbf{S}_0^*\|_\infty \leq \|\mathbf{\Lambda}_0 - \mathbf{\Lambda}_0^*\|_\infty$. Combining the last two displays, for $\theta \in \Theta_{\epsilon_0}$, we have $K(p_{\theta^*}, p_\theta) \leq \|\log(p_{\theta^*}/p_\theta)\|_\infty < C_1 \epsilon_0$, where $C_1 = C_0(c_1 + c_2)$ is a positive constant. Then, we obtain

$$\Theta_{\epsilon_0} \subseteq \{\theta \in \Theta : K(p_{\theta^*}, p_\theta) < C_1 \epsilon_0\}. \tag{29}$$

We denote the renormalized restriction of $\Pi$ to $\Theta_{\epsilon_0}$ by $\Pi_{\epsilon_0}$. We note that

$$\int_\Theta \prod_{t=1}^n \frac{p_\theta}{p_{\theta^*}}(D_t) d\Pi(\theta) \geq \Pi(\Theta_{\epsilon_0}) \int_{\Theta_{\epsilon_0}} \prod_{t=1}^n \frac{p_\theta}{p_{\theta^*}}(D_t) d\Pi_{\epsilon_0}(\theta)$$

$$\geq \Pi(\Theta_{\epsilon_0}) \exp\left(-\int_{\Theta_{\epsilon_0}} \sum_{t=1}^n \log\left(\frac{p_{\theta^*}}{p_\theta}\right)(D_t) d\Pi_{\epsilon_0}(\theta)\right), \tag{30}$$

where the last inequality holds by Jensen's inequality. Since the log-likelihood ratio is bounded from (28), by Hoeffding's inequality, we have

$$\mathbb{P}_{\theta^*}^n\left(\sum_{t=1}^n \log\left(\frac{p_{\theta^*}}{p_\theta}\right)(D_t) - nK(p_{\theta^*}, p_\theta) < \epsilon_0 n\right) > 1 - \exp\left(-\frac{\epsilon_0^2}{2C_0^2} n\right). \tag{31}$$

Let $\Omega_1$ be the event in the left-hand side of the last display. Thus, on the event $\Omega_1$, we have

$$-\int_{\Theta_{\epsilon_0}} \sum_{t=1}^n \log\left(\frac{p_{\theta^*}}{p_\theta}\right)(D_t) d\Pi_{\epsilon_0}(\theta) > -\int_{\Theta_{\epsilon_0}} nK(p_{\theta^*}, p_\theta) d\Pi_{\epsilon_0}(\theta) - \epsilon_0 n$$

$$> -(C_1 + 1)\epsilon_0 n,$$

where the last inequality holds by (29). Combining this with (30), on the event $\Omega_1$, we have

$$\int_\Theta \prod_{t=1}^n \frac{p_\theta}{p_{\theta^*}}(D_t)d\Pi(\theta) > \Pi(\Theta_{\epsilon_0})\exp\left(-(C_1+1)\epsilon_0 n\right). \tag{32}$$

Let $\epsilon_1 = \epsilon_0 n^{-\gamma}$. By Lemma C.3, with the specified prior (3) and the hyperparameter condition (P2), there exist positive constants $c_3$, $c_4$ and $c_5$ depending on $p_{\min}$, $p_{\max}$, $M_1$, $M_2$, $\underline{\alpha}$, $\overline{\alpha}$, $\rho$ and $\epsilon_0$, such that

$$\Pi\left(\|\mathbf{\Lambda}_0 - \mathbf{\Lambda}_0^*\|_\infty \leq \epsilon_1\right) \geq c_3 \exp(-c_4 K - c_5 K \log_- \epsilon_1).$$

In addition, under the prior condition (P1), we have $\Pi(\|\beta - \beta^*\|_2 \leq \epsilon_0) \geq C_2$, where $C_2$ is a positive constant depending on $d$, $\epsilon_0$ and the lower bound of the prior on a neighborhood of $\beta^*$. Then, we have

$$\Pi(\Theta_{\epsilon_0}) \geq \Pi(\|\mathbf{\Lambda}_0 - \mathbf{\Lambda}_0^*\|_\infty \leq \epsilon_0) \cdot \Pi(\|\beta - \beta^*\|_2 \leq \epsilon_0)$$
$$\geq C_2 c_3 \exp(-c_4 K - c_5 \log_- \epsilon_1 K),$$

where the last inequality follows from the previous display and the fact that $\epsilon_1 \leq \epsilon_0$ for $n \geq 1$. Combining this with (32), on the event $\Omega_1$, we have

$$\int_\Theta \prod_{t=1}^n \frac{p_\theta}{p_{\theta^*}}(D_t)d\Pi(\theta) > C_2 c_3 \exp\left(-c_4 K - c_5 \log_- \epsilon_1 K - (C_1+1)\epsilon_0 n\right)$$

$$> C_2 c_3 \exp\left(-C_3 K \log n - (C_1+1)\epsilon_0 n\right), \tag{33}$$

where the last inequality holds by $C_3 = c_4 + c_5(\log_- \epsilon_0 + 1)$ because $\log_- \epsilon_1 < \log_- \epsilon_0 + \log n$. By Lemma A.3 and A.4, there exist tests $\Phi_n$ such that

$$\mathbb{E}_{\theta^*}^n[\Phi_n] \leq \exp(-C_4 n), \quad \sup_{\theta \in U^c} \mathbb{E}_\theta^n[1 - \Phi_n] \leq \exp(-C_4 n), \tag{34}$$

where $C_4$ is a positive constant depending on $M_1$, $M_2$, $L$, $B$, $p_{\min}$, $p_{\max}$, $\kappa$ and $\epsilon$. Then, we have
$$\mathbb{E}_{\theta^*}^n[\Pi(U^c|\mathbf{D}_n)] \leq \mathbb{E}_{\theta^*}^n[\Phi_n] + \mathbb{E}_{\theta^*}^n[(1 - \Phi_n)\Pi(U^c|\mathbf{D}_n)\mathbb{1}\{\Omega_1\}] + \mathbb{P}_{\theta^*}^n(\Omega_1^c)$$

$$\leq \mathbb{E}_{\theta^*}^n[\Phi_n] + (C_2 c_3)^{-1}\exp(C_3 K \log n + (C_1+1)\epsilon_0 n)\sup_{\theta \in U^c}\mathbb{E}_\theta^n[1 - \Phi_n] + \exp\left(-\frac{\epsilon_0^2}{2C_0^2}n\right)$$

$$\leq \exp(-C_4 n) + C_5 \exp\left(C_3 C_p n^{\frac{2}{3}}\log n - (C_4 - (C_1+1)\epsilon_0)n\right) + \exp\left(-C_6 n\right),$$

where the second inequality holds by (31) and (33), and the last inequality follows from (34), with $C_5 = (C_2 c_3)^{-1}$ and $C_6 = \epsilon_0^2/(2C_0^2)$, and $K \leq C_p n^{2/3}$ for $\gamma < 2/3$, where $C_p$ is a positive constant depending on $p_{\min}$, $p_{\max}$ and $\kappa$. We choose $\epsilon_0 = (C_4/(3(C_1+1))) \wedge ((M_1 \wedge (1 - M_2)/2) \wedge B)$ to ensure that the second term on the right-hand side of the previous display is bounded by $C_5 \exp(-(C_4/3)n)$, provided that $n^{1/3}(\log n)^{-1} \geq 3C_3 C_p/C_4$. Then, we have

$$\mathbb{E}_{\theta^*}^n[\Pi(U^c|\mathbf{D}_n)] \leq \exp(-C_4 n) + C_5 \exp(-(C_4/3)n) + \exp\left(-C_6 n\right)$$
$$\leq C_7 \exp(-C_8 n),$$

where $C_7 = C_5 + 2$ and $C_8 = (C_4/3) \wedge C_6$. By the Markov inequality, for $n \geq (3C_3 C_p/C_4)^3$,

$$\mathbb{P}_{\theta^*}^n(\Pi(U^c|\mathbf{D}_n) \geq C_7 \exp(-C_9 n)) < \exp(-C_9 n),$$

where $C_9 = C_8/2$. This concludes the proof for the case where $\gamma < 2/3$.

Now, we suppose that $\gamma \geq 2/3$. Let $\epsilon_2 = n^{-1/3}$ and $J = \lceil(p_{\max} - p_{\min})/(\kappa\epsilon_2)\rceil$. Define $(k_1, \ldots, k_J)$ as a subsequence of $[K]$ such that $p_{\min} + (j-1)\kappa\epsilon_2 < g_{k_j} \leq p_{\min} + (j-1)\kappa\epsilon_2 + \delta$ for $j = 1, \ldots, J-1$, and set $k_J = K$.

Suppose that $|S_{0,k_j} - S_{0,k_j}^*| < \epsilon/2$ for every $j = 1, \ldots, J$. Then, for any $k \in [K]$ with $k_j \leq k < k_{j+1}$ for $j = 1, \ldots, J-2$, we have

$$S_{0,k}^* - S_{0,k} \leq S_{0,k}^* - S_{0,k_{j+1}}$$
$$\leq |S_{0,k_{j+1}}^* - S_{0,k_{j+1}}| + |S_{0,k}^* - S_{0,k_{j+1}}^*|$$
$$< \epsilon/2 + L_0|g_k - g_{k_{j+1}}|$$
$$\leq \epsilon/2 + L_0(\kappa\epsilon_2 + \delta)$$
$$\leq \epsilon/2 + 2L_0\kappa\epsilon_2$$
$$\leq \epsilon,$$

where the third inequality holds by our assumption and $L_0$-Lipschitz continuity of $S_0^*$, with $L_0$ being a positive constant because (A5) is assumed, the fourth inequality holds by the definition of $k_j$, the fifth inequality holds because $\delta \leq \kappa\epsilon_2$ for $\gamma \geq 2/3$, and the last inequality holds for sufficiently large $n$ so that $\epsilon_2 \leq \epsilon/(4L_0\kappa)$. Note that $|g_{k_{J-1}} - g_{k_J}| < 2\kappa\epsilon_2$ by the definition of $J$. Then, for any $k \in [K]$ with $k_{J-1} \leq k \leq k_J$, we have

$$S_{0,k}^* - S_{0,k} < \epsilon.$$

Combining the preceding two displays, we have $S_{0,k}^* - S_{0,k} < \epsilon$ for any $k \in [K]$. Similarly, for any $k \in [K]$, we have $S_{0,k} - S_{0,k}^* < \epsilon$. Therefore, for $n \geq (4L_0\kappa/\epsilon)^3$, we have

$$\{\mathbf{S}_0 \in \mathcal{S}_0 : \|\mathbf{S}_0 - \mathbf{S}_0^*\|_\infty \geq \epsilon\} \subset \{\mathbf{S}_0 \in \mathcal{S}_0 : \|\mathbf{S}_0 - \mathbf{S}_0^*\|_{\infty,J} \geq \epsilon/2\}. \tag{35}$$

Then, we can decompose

$$\mathbb{E}_{\theta^*}^n\left[\Pi(U^c \mid \mathbf{D}_n)\right] \leq \mathbb{E}_{\theta^*}^n\left[\Pi(\{\theta \in \Theta : \|\mathbf{S}_0 - \mathbf{S}_0^*\|_{\infty,J} \geq \epsilon/2 \text{ or } \|\beta - \beta^*\|_2 \geq \epsilon\} \mid \mathbf{D}_n)\right]$$
$$\leq \underbrace{\mathbb{E}_{\theta^*}^n\left[\Pi(U_1 \mid \mathbf{D}_n)\right]}_{\text{(i)}} + \underbrace{\mathbb{E}_{\theta^*}^n\left[\Pi(U_2 \mid \mathbf{D}_n)\right]}_{\text{(ii)}} + \underbrace{\mathbb{E}_{\theta^*}^n\left[\Pi(U_3 \mid \mathbf{D}_n)\right]}_{\text{(iii)}}, \tag{36}$$

where

$$U_1 = \{\theta \in \Theta : \max_{2 \leq j \leq J-1} |S_{0,k_j} - S_{0,k_j}^*| \geq \epsilon/4 \text{ or } \|\beta - \beta^*\|_2 \geq \epsilon\},$$
$$U_2 = \{\theta \in \Theta : |S_{0,k_1} - S_{0,k_1}^*| \geq \epsilon/2, \max_{2 \leq j \leq J-1} |S_{0,k_j} - S_{0,k_j}^*| < \epsilon/4\},$$
$$U_3 = \{\theta \in \Theta : |S_{0,k_J} - S_{0,k_J}^*| \geq \epsilon/2, \max_{2 \leq j \leq J-1} |S_{0,k_j} - S_{0,k_j}^*| < \epsilon/4\}.$$

The proof for (i) is similar to that of $\gamma < 2/3$. Let $\epsilon_3 > 0$ be a constant to be chosen later. Similarly as in (35), we have

$$\{\mathbf{S}_0 \in \mathcal{S}_0 : \|\mathbf{\Lambda}_0 - \mathbf{\Lambda}_0^*\|_\infty \geq \epsilon_3 n^{-\frac{1}{3}}\} \subset \{\mathbf{S}_0 \in \mathcal{S}_0 : \|\mathbf{\Lambda}_0 - \mathbf{\Lambda}_0^*\|_{\infty,J} \geq C_{10}\epsilon_3 n^{-\frac{1}{3}}\}, \tag{37}$$

where $C_{10}$ is a positive constant depending on $L_0$ and $\kappa$. Then, we have

$$\Pi(\Theta_{\epsilon_3}) \geq \Pi(\|\mathbf{\Lambda}_0 - \mathbf{\Lambda}_0^*\|_\infty \leq \epsilon_3) \cdot \Pi(\|\beta - \beta^*\|_2 \leq \epsilon_3)$$
$$\geq C_{11}\Pi(\|\mathbf{\Lambda}_0 - \mathbf{\Lambda}_0^*\|_{\infty,J} \leq C_{10}\epsilon_3 n^{-\frac{1}{3}})$$
$$\geq C_{11}c_6 \exp\left(-c_7 J - c_8 \log_-(C_{10}\epsilon_3 n^{-\frac{1}{3}})J\right),$$

where the second inequality holds by a positive constant $C_{11}$ depending on $d$, $\epsilon_3$ and the prior's lower bound near $\beta^*$, and the last inequality follows from constants $c_6$, $c_7$ and $c_8$ in Lemma C.3, depending on $p_{\min}$, $p_{\max}$, $M_1$, $M_2$, $\underline{\alpha}$, $\overline{\alpha}$, $\rho$, $C_{10}$ and $\epsilon_3$. Similarly as in (33), there exists the event $\Omega_2$ such that $\mathbb{P}_{\theta^*}^n(\Omega_2^c) \leq \exp(-\epsilon_3^2/(2C_0^2)n)$, and on the event $\Omega_2$, we have

$$\int_\Theta \prod_{t=1}^n \frac{p_\theta}{p_{\theta^*}}(D_t)d\Pi(\theta) > C_{11}c_6 \exp\left(-c_7 J - c_8 \log_-(C_{10}\epsilon_3 n^{-\frac{1}{3}})J - (C_1+1)\epsilon_3 n\right)$$
$$> C_{11}c_6 \exp\left(-C_{12}J\log n - (C_1+1)\epsilon_3 n\right)$$
$$\geq C_{11}c_6 \exp\left(-C_{12}C_p n^{\frac{1}{3}}\log n - (C_1+1)\epsilon_3 n\right),$$

where the second inequality holds by $C_{12} = c_7 + c_8(\log_-(C_{10}\epsilon_3) + 1)$, and the last inequality holds because $J \leq C_p n^{1/3}$ by the definition of $J$. By Lemma A.5, there exist tests $\Phi_{n,1}$ such that

$$\mathbb{E}_{\theta^*}^n\left[\Phi_{n,1}\right] \leq \exp(-C_{13}n), \quad \sup_{\theta \in U_1} \mathbb{E}_\theta^n\left[1 - \Phi_{n,1}\right] \leq \exp(-C_{13}n),$$

where $C_{13}$ is a positive constant depending on $M_1$, $M_2$, $L$, $B$, $p_{\min}$, $p_{\max}$, $\kappa$ and $\epsilon$. We choose $\epsilon_3 = (C_{13}/(3(C_1 + 1))) \wedge ((M_1 \wedge (1 - M_2)/2) \wedge B)$. Combining the last two displays, for $n \geq (3C_{12}C_p/C_{13})^{3/2}$, we have

$$\mathbb{E}_{\theta^*}^n\left[\Pi(U_1 \mid \mathbf{D}_n)\right] \leq C_{14}\exp(-C_{15}n), \tag{38}$$

where $C_{14} = (C_{11}c_6)^{-1} + 2$ and $C_{15} = (C_{13}/3) \wedge (\epsilon_3^2/(2C_0^2))$ are positive constants.

We now consider the term (ii). We split $U_2$ in $U_{2,-}$ and $U_{2,+}$, where

$$U_{2,-} = \{\theta \in \Theta : S_{0,k_1} - S_{0,k_1}^* \leq -\epsilon/2, \max_{2 \leq j \leq J-1} |S_{0,k_j} - S_{0,k_j}^*| < \epsilon/4\},$$

$$U_{2,+} = \{\theta \in \Theta : S_{0,k_1} - S_{0,k_1}^* \geq \epsilon/2, \max_{2 \leq j \leq J-1} |S_{0,k_j} - S_{0,k_j}^*| < \epsilon/4\}.$$

Note that $U_{2,-} \subset U_{2,-}^1 \cup U_{2,-}^2$, where

$$U_{2,-}^1 = \{\theta \in \Theta : \|\beta - \beta^*\|_2 \geq \eta\},$$
$$U_{2,-}^2 = \{\theta \in \Theta : S_{0,k_1} - S_{0,k_1}^* \leq -\epsilon/2, \|\beta - \beta^*\|_2 < \eta\},$$

for some sufficiently small positive constant $\eta$ depending on $\epsilon$. By Lemma A.3, there exist exponentially consistent tests $\Phi_{n,2,1}$ for testing $H_0 : \theta = \theta^*, H_1 : \theta \in U_{2,-}^1$. Similarly as (26) in the proof of Lemma A.5, we construct the tests $\Psi_{n,2,2}$ for $U_{2,-}^2$ by

$$\Psi_{n,2,2} = \mathbb{1}\left\{\sum_{t=1}^n \phi_2(D_t) > \sum_{t=1}^n (\mathbb{E}_{\theta^*}[\phi_2(D_t)] + \mathbb{E}_\theta[\phi_2(D_t)])/2\right\},$$

where $\phi_2(D_t) = \mathbb{1}\{P_t \in \{g_{k_1}, \ldots, g_{k_2}\}, Y_t = 0\}$. By a similar argument as the proof in Lemma A.5, we can show that $\mathbb{E}_{\theta^*}^n[\Psi_{n,2,2}(\mathbf{D}_n)] \to 0$ and $\sup_{\theta \in U_{2,-}^2} \mathbb{E}_\theta^n[1 - \Psi_{n,2,2}(\mathbf{D}_n)] \to 0$ as $n \to \infty$. By Lemma D.11 of [16], there exist exponentially consistent tests $\Phi_{n,2,2}$ for testing $H_0 : \theta = \theta^*, H_1 : \theta \in U_{2,-}^2$. Let $\Phi_{n,2} = \Phi_{n,2,1} \vee \Phi_{n,2,2}$. Then, there exists a positive constant $C_{16}$ depending on $M_1$, $M_2, L, B, p_{\min}, p_{\max}, \kappa$ and $\epsilon$ such that

$$\mathbb{E}_{\theta^*}^n[\Phi_{n,2}] \leq \exp(-C_{16}n), \quad \sup_{\theta \in U_{2,-}} \mathbb{E}_\theta^n[1 - \Phi_{n,2}] \leq \exp(-C_{16}n).$$

Then, by a similar argument as the preceding, for $n \geq (3C_{12}C_p/C_{16})^{3/2}$, it holds that

$$\mathbb{E}_{\theta^*}^n[\Pi(U_{2,-} \mid \mathbf{D}_n)] \leq C_{14}\exp(-C_{17}n), \tag{39}$$

where $C_{17} = (C_{16}/3) \wedge (\epsilon_3^2/(2C_0^2))$ is a positive constant.

We restrict ourselves to vectors $\mathbf{S}_0$ such that $\max_{2 \leq j \leq J-1} |S_{0,k_j} - S_{0,k_j}^*| < \epsilon/4$. Suppose that $S_{0,k_1} - S_{0,k_2} < \epsilon/4$. Then, we have

$$\begin{aligned} S_{0,k_1} - S_{0,k_1}^* &= S_{0,k_1} - S_{0,k_2} + S_{0,k_2} - S_{0,k_1}^* \\ &\leq S_{0,k_1} - S_{0,k_2} + S_{0,k_2} - S_{0,k_2}^* \\ &< \epsilon/4 + \epsilon/4 \\ &= \epsilon/2, \end{aligned}$$

where the first inequality holds by the monotonicity of $S_0^*$, and the last inequality follows from our assumption and the fact that $|S_{0,k_2} - S_{0,k_2}^*| < \epsilon/4$. Thus, it holds that

$$U_{2,+} \subset \{\theta \in \Theta : S_{0,k_1} - S_{0,k_2} \geq \epsilon/4, \max_{2 \leq j \leq J-1} |S_{0,k_j} - S_{0,k_j}^*| < \epsilon/4\}$$

$$\subset \{\theta \in \Theta : S_{0,k_1} - S_{0,k_2} \geq \epsilon/4\}.$$

Then, it is sufficient to show that $\mathbb{E}_{\theta^*}^n[\Pi(U_{2,+}' \mid \mathbf{D}_n)] \to 0$ as $n \to \infty$, where $U_{2,+}' = \{\theta \in \Theta : S_{0,k_1} - S_{0,k_2} \geq \epsilon/4\}$. Let $\epsilon_4 = \epsilon_5 n^{-1/3}$, where $\epsilon_5 > 0$ is a sufficiently small constant to be chosen later. Similarly as in (32), there exists the event $\Omega_3$ such that $\mathbb{P}_{\theta^*}^n(\Omega_3^c) \leq \exp(-\epsilon_4^2/(2C_0^2)n)$, and on the event $\Omega_3$, we have

$$\int_\Theta \prod_{t=1}^n \frac{p_\theta}{p_{\theta^*}}(D_t)d\Pi(\theta) > \Pi(\Theta_{\epsilon_4})\exp(-(C_1+1)\epsilon_4 n).$$

Furthermore, we have

$$\begin{aligned} \Pi(\Theta_{\epsilon_4}) &\geq \Pi(\|\mathbf{\Lambda}_0 - \mathbf{\Lambda}_0^*\|_\infty \leq \epsilon_4) \cdot \Pi(\|\beta - \beta^*\|_2 \leq \epsilon_4) \\ &\geq c_6\exp(-c_7 J - c_8\log_-(C_{10}\epsilon_4)J) \cdot \Pi(\|\beta - \beta^*\|_2 \leq \epsilon_4) \\ &\geq C_{18}c_6\exp(-c_7 J - c_8\log_-(C_{10}\epsilon_4)J) \cdot \epsilon_4^d \\ &= C_{18}c_6\exp(-c_7 J - c_8\log_-(C_{10}\epsilon_4)J + d\log\epsilon_4), \end{aligned}$$

where the second inequality holds by (37) and Lemma C.3, and the last inequality holds because $\Pi(\|\beta - \beta^*\|_2 \leq \epsilon_4) \geq C_{18}\epsilon_4^d$ with a positive constant $C_{18}$ depending on $d$ and the prior's lower bound near $\beta^*$. Combining the last two displays, on the event $\Omega_3$, we have

$$\int_\Theta \prod_{t=1}^n \frac{p_\theta}{p_{\theta^*}}(D_t)d\Pi(\theta) > C_{18}c_6 \exp\left(-c_7 J - c_8 \log_-(C_{10}\epsilon_4)J + d\log\epsilon_4 - (C_1+1)\epsilon_4 n\right)$$

$$\geq C_{18}c_6 \exp\left(-C_{19}n^{\frac{1}{3}}\log n - (C_1+1)\epsilon_5 n^{\frac{2}{3}}\right),$$

where the last inequality holds by a positive constant $C_{19} = C_p(c_7 + c_8(\log_-(C_{10}\epsilon_5)+1)) + d\log_-\epsilon_5$. This implies that

$$\mathbb{E}_{\theta^*}^n\left[\Pi(U'_{2,+} \mid \mathbf{D}_n)\right] \leq \mathbb{E}_{\theta^*}^n\left[\Pi(U'_{2,+} \mid \mathbf{D}_n)\mathbb{1}\{\Omega_3\}\right] + \mathbb{P}_{\theta^*}^n(\Omega_3^c)$$

$$\leq (C_{18}c_6)^{-1}\exp\left(C_{19}n^{\frac{1}{3}}\log n + (C_1+1)\epsilon_5 n^{\frac{2}{3}}\right)\Pi(U'_{2,+}) + \exp\left(-\frac{\epsilon_5^2}{2C_0^2}n^{\frac{1}{3}}\right).$$
$$(40)$$

We now prove that the prior mass of $U'_{2,+}$ is exponentially small. Note that

$$S_{0,k_1} - S_{0,k_2} = \exp(-\Lambda_{0,k_1}) - \exp(-\Lambda_{0,k_2})$$
$$\leq \Lambda_{0,k_2} - \Lambda_{0,k_1}$$
$$= \delta \sum_{k=k_1+1}^{k_2} \lambda_{0,k}.$$

Let $\overline{\lambda} = \sum_{k=k_1+1}^{k_2} \lambda_{0,k}$. By (3), $\overline{\lambda}$ is gamma distributed with parameters $\alpha_0$ and $\rho$, where $\alpha_0 = \sum_{k=k_1+1}^{k_2} \alpha_k$. Then, we have

$$\Pi\left(U'_{2,+}\right) \leq \Pi\left(\delta\overline{\lambda} \geq \frac{\epsilon}{4}\right)$$

$$\leq \Pi\left(\overline{\lambda} \geq \frac{\epsilon}{4C_p}K\right)$$

$$\leq 2^{\alpha_0}\exp\left(-\frac{\rho\epsilon}{8C_p}K\right)$$

$$\leq \exp\left(C_{20}\log 2 \cdot n^{\gamma-\frac{1}{3}} - C_{21}n^\gamma\right)$$

$$\leq \exp\left(-C_{21}/2 \cdot n^\gamma\right),$$

where the second inequality holds because $K \leq C_p\delta^{-1}$, the third inequality follows from Chernoff bounds. Here, the fourth inequality holds because $K \geq C'_p n^\gamma$ and $\alpha_0 \leq K/J \cdot \overline{\alpha} \leq C_{20} \cdot n^{\gamma-1/3}$ under (P2) with positive constants $C_{20}$ depending on $\overline{\alpha}$, $p_{\min}$, $p_{\max}$ and $\kappa$, and $C_{21} = \rho\epsilon C'_p/(8C_p)$. The last inequality holds for $n \geq (2C_{20}\log 2/C_{21})^3$. Combining this with (40), we have

$$\mathbb{E}_{\theta^*}^n\left[\Pi(U'_{2,+} \mid \mathbf{D}_n)\right] \leq C_{22}\exp\left(C_{19}n^{\frac{1}{3}}\log n + (C_1+1)\epsilon_5 n^{\frac{2}{3}} - \frac{C_{21}}{2}n^{\frac{2}{3}}\right) + \exp\left(-\frac{\epsilon_5^2}{2C_0^2}n^{\frac{1}{3}}\right),$$

where the inequality holds because $\gamma \geq 2/3$ with a positive constant $C_{22} = (C_{18}c_6)^{-1}$. We choose $\epsilon_5 = (C_{21}/(6(C_1+1)))\wedge((M_1\wedge(1-M_2)/2)\wedge B)$. Then, for $n \geq (3C_{19}/C_{21})^3\vee(2C_{20}\log 2/C_{21})^3$, we have

$$\mathbb{E}_{\theta^*}^n\left[\Pi(U'_{2,+} \mid \mathbf{D}_n)\right] \leq C_{22}\exp\left(-\frac{C_{21}}{3}n^{\frac{2}{3}}\right) + \exp\left(-\frac{\epsilon_5^2}{2C_0^2}n^{\frac{1}{3}}\right)$$

$$\leq C_{23}\exp\left(-C_{24}n^{\frac{1}{3}}\right),$$
$$(41)$$

where the last inequality holds by postive constants $C_{23} = C_{22}+1$ and $C_{24} = (C_{21}/3)\wedge(\epsilon_5^2/(2C_0^2))$. Combining (39) and (41), for $n \geq (3C_{12}C_p/C_{16})^{3/2} \vee (3C_{19}/C_{21})^3 \vee (2C_{20}\log 2/C_{21})^3$, we have

$$\mathbb{E}_{\theta^*}^n\left[\Pi(U_2 \mid \mathbf{D}_n)\right] \leq C_{14}\exp(-C_{17}n) + C_{23}\exp\left(-C_{24}n^{\frac{1}{3}}\right)$$

$$\leq C_{25}\exp\left(-C_{26}n^{\frac{1}{3}}\right),$$
$$(42)$$

where $C_{25} = C_{14} + C_{23}$ and $C_{26} = C_{17} \wedge C_{24}$ are positive constants.

By a similar argument as (ii), there exist positive constants $C_{27}$ and $C_{28}$ such that

$$\text{(iii)} \leq C_{27} \exp\left(-C_{28} n^{\frac{1}{3}}\right). \tag{43}$$

Combining (36), (38), (42) and (43), we have

$$\mathbb{E}_{\theta^*}^n \left[\Pi(U^c \mid \mathbf{D}_n)\right] \leq C_{29} \exp\left(-C_{30} n^{\frac{1}{3}}\right),$$

where $C_{29} = C_{14} + C_{25} + C_{27}$ and $C_{30} = C_{15} \wedge C_{26} \wedge C_{28}$ are positive constants. By the Markov inequality, we have

$$\mathbb{P}_{\theta^*}^n \left(\Pi(U^c \mid \mathbf{D}_n) \geq C_{29} \exp\left(-C_{31} n^{\frac{1}{3}}\right)\right) < \exp\left(-C_{31} n^{\frac{1}{3}}\right),$$

where $C_{31} = C_{30}/2$. This concludes the proof for the case where $\gamma \geq 2/3$.

$\square$

## A.2 Proof of Theorem 3.1

**Lemma A.6.** *Let* $\Theta' = \{(\mathbf{S}_0, \beta) \in \Theta : S_{0,K} \geq M_1, S_{0,1} \leq M_2, \|\beta\|_2 \leq D\}$, *where* $M_1$, $M_2$ *and* $D$ *are some positve constants such that* $0 < M_1 < M_2 < 1$, *and let* $\mathcal{P}' = \{p_\theta : \theta \in \Theta'\}$. *Under the assumption (A2), there exist positive constants* $C_1$ *and* $C_2$ *depending only on* $M_1, M_2, L$ *and* $D$ *such that for every* $\epsilon > 0$, *it holds that*

$$N(\epsilon, \mathcal{P}', \mathcal{D}_H) \leq (C_1/\epsilon + K)^K (C_2/\epsilon)^d.$$

*Proof.* Let $\mathcal{S}_0' = \{\mathbf{S}_0 = (S_{0,1}, \ldots, S_{0,K}) : M_2 \geq S_{0,1} \geq \cdots \geq S_{0,K} \geq M_1\}$ and $\mathcal{H}_0' = \{\Lambda_0 = (\Lambda_{0,1}, \ldots, \Lambda_{0,K}) : \lambda_2 \leq \Lambda_{0,1} \leq \cdots \leq \Lambda_{0,K} \leq \lambda_1\}$, where $\lambda_2 = -\log M_2$ and $\lambda_1 = -\log M_1$. Then, for any $\Lambda_0$ corresponding to the vector $\mathbf{S}_0 \in \mathcal{S}_0'$, $\Lambda_0$ belongs to $\mathcal{H}_0'$ since $\Lambda_{0,k} = -\log S_{0,k} \in [\lambda_2, \lambda_1]$ for any $k = 1, \ldots, K$. For $\epsilon > 0$, let

$$\mathcal{H}_{0,\epsilon}' = \{\Lambda_0 \in \mathcal{H}_0' : (\Lambda_{0,1}, \ldots, \Lambda_{0,K}) = (m_1 \epsilon, \ldots, m_K \epsilon)$$

$$\text{for some positive integers } m_1, \ldots, m_K \text{ satisfying } m_1 \leq \cdots \leq m_K\}.$$

Then, it is not difficult to show that $\mathcal{H}_{0,\epsilon}'$ is an $\epsilon$-cover of $\mathcal{H}_0'$ with respect to $\|\cdot\|_\infty$. Note that the cardinality of $\mathcal{H}_{0,\epsilon}'$ is the number of $K$-tuples of integers $(m_1, \ldots, m_K)$ satisfying $\lfloor \lambda_1/\epsilon \rfloor \leq m_1 \leq \cdots \leq m_K \leq \lfloor \lambda_2/\epsilon \rfloor$, which is given as $\binom{\lfloor \lambda_2/\epsilon \rfloor - \lfloor \lambda_1/\epsilon \rfloor + K}{K}$ based on simple combinatorics. Hence, we have $N(\epsilon, \mathcal{H}_0', \|\cdot\|_\infty) \leq \binom{\lfloor \lambda_2/\epsilon \rfloor + K}{K} \leq (\lambda_2/\epsilon + K)^K$. Therefore, we have

$$N(\epsilon, \mathcal{H}_0', \|\cdot\|_\infty) \leq (\lambda_2/\epsilon + K)^K. \tag{44}$$

Take any two parameters $\theta = (\mathbf{S}_0, \beta), \theta' = (\mathbf{S}_0', \beta') \in \Theta'$. By Lemma C.1 and C.2, there exist positive constants $c_1$, $c_2$ and $c_3$, depending on $M_1, M_2, L$ and $D$, such that for any $x \in \mathcal{X}$ and $p \in \mathcal{G}$,

$$\mathcal{D}_H(p_\theta, p_{\theta'}) \leq c_1 \|H_\theta - H_{\theta'}\|_\infty$$
$$\leq C_1 \|\Lambda_0 - \Lambda_0'\|_\infty + C_2 \|\beta - \beta'\|_2, \tag{45}$$

where $C_1 = c_1 c_2$ and $C_2 = c_1 c_3$.

Let $m := N(\epsilon/(2C_1), \mathcal{H}_0', \|\cdot\|_\infty)$ and $l := N(\epsilon/(2C_2), \mathcal{B}', \|\cdot\|_2)$, where $\mathcal{B}' = \{\beta \in \mathbb{R}^d : \|\beta\|_2 \leq D\}$. This definition implies that there exist $\Lambda_{0,1}, \ldots, \Lambda_{0,m} \in \mathcal{H}_0'$ such that for every $\Lambda_0 \in \mathcal{H}_0'$, the inequality $\|\Lambda_0 - \Lambda_{0,i}\|_\infty < \epsilon/(2C_1)$ holds for some $1 \leq i \leq m$. Similarly, there exist $\beta_1, \ldots, \beta_l \in \mathcal{B}'$ such that for every $\beta \in \mathcal{B}'$, $\|\beta - \beta_j\|_2 < \epsilon/(2C_2)$ holds for some $1 \leq j \leq l$. Let $\theta_{ij} = (\mathbf{S}_{0,i}, \beta_j) \in \Theta'$, where $\mathbf{S}_{0,i}$ be the vector corresponding to $\Lambda_{0,i}$ for $i = 1, \ldots, m$ and $j = 1, \ldots, l$. By (45), for any $\theta = (\mathbf{S}_0, \beta) \in \Theta'$, there exists $\theta_{ij}$ for some $1 \leq i \leq m$ and $1 \leq j \leq l$ such that

$$\mathcal{D}_H(p_\theta, p_{\theta_{ij}}) \leq C_1 \|\Lambda_0 - \Lambda_{0,i}\|_\infty + C_2 \|\beta - \beta_j\|_2 \leq \epsilon.$$

Consequently, the covering number $N(\epsilon, \mathcal{P}', \mathcal{D}_H)$ is of order $ml$. Note that $m \leq (2C_1 \lambda_2/\epsilon + K)^K$ by (44). Furthermore, by Proposition C.2 of [16], $l \leq (6DC_2/\epsilon)^d$. Therefore, we have

$$N(\epsilon, \mathcal{P}', \mathcal{D}_H) \leq (C_3/\epsilon + K)^K (C_4/\epsilon)^d,$$

where $C_3 = 2C_1 \lambda_2$ and $C_4 = 6DC_2$ are positive constants depending only on $M_1, M_2, L$ and $D$.

$\square$

*Proof of Theorem 3.1.* First, we define the square Kullback-Leibler variation as $V_0(p,q) = \int(\log(p/q) - K(p/q))^2 dP$. For every $\epsilon > 0$, we define neighborhoods of $\theta^*$ by

$$B(\theta^*, \epsilon) = \{\theta \in \Theta : K(p_{\theta^*}, p_\theta) \leq \epsilon^2, V_0(p_{\theta^*}, p_\theta) \leq \epsilon^2\}.$$

We begin by checking the prior mass condition. Note that there exist constants $M_1$ and $M_2$ such that $0 < M_1 \leq S_0^*(v) \leq M_2 < 1$ for any $v \in [p_{\min}, p_{\max}]$ by assumption (A5). Let $U = \{(\mathbf{S}_0, \beta) \in \Theta : \|\mathbf{S}_0 - \mathbf{S}_0^*\|_\infty \vee \|\beta - \beta^*\|_2 < \epsilon_0\}$ be a neighborhood of $\theta^*$, where $\epsilon_0$ is a positive constant that can be chosen as $\epsilon_0 = ((M_1 \wedge (1 - M_2))/2) \wedge B$ to ensure that $U \subseteq \Theta$. By (28) in the proof of Lemma A.1, there exists a positive constant $C_0$ depending on $M_1$, $M_2$, $L$ and $B$ such that for any $\theta \in U$,

$$\left\|\log\frac{p_{\theta^*}}{p_\theta}\right\|_\infty \leq C_0.$$

By Lemma B.2 in [16], the uniformly bounded likelihood ratio implies that

$$
\begin{aligned}
K(p_{\theta^*}, p_\theta) &\leq c_1 \mathcal{D}_H^2(p_{\theta^*}, p_\theta)\left\|\frac{p_{\theta^*}}{p_\theta}\right\|_\infty \leq C_1 \mathcal{D}_H^2(p_{\theta^*}, p_\theta), \\
V_0(p_{\theta^*}, p_\theta) &\leq c_2 \mathcal{D}_H^2(p_{\theta^*}, p_\theta)\left\|\frac{p_{\theta^*}}{p_\theta}\right\|_\infty \leq C_2 \mathcal{D}_H^2(p_{\theta^*}, p_\theta),
\end{aligned}
\tag{46}
$$

where $C_1 = c_1 \exp(C_0)$ and $C_2 = c_2 \exp(C_0)$ for universal constants $c_1$ and $c_2$. By Lemma C.1 and Lemma C.2, there exist postive constants $c_3$, $c_4$ and $c_5$, depending on $M_1$, $M_2$, $L$ and $B$ such that for any $\theta \in U$,

$$
\begin{aligned}
\mathcal{D}_H(p_{\theta^*}, p_\theta) &\leq c_3\|H_{\theta^*} - H_\theta\|_\infty \\
&\leq C_3\|\mathbf{\Lambda}_0 - \mathbf{\Lambda}_0^*\|_\infty + C_4\|\beta - \beta^*\|_2,
\end{aligned}
$$

where $C_3 = c_3 c_4$ and $C_4 = c_3 c_5$. Let $\Theta_n = \{\theta \in \Theta : \|\mathbf{\Lambda}_0 - \mathbf{\Lambda}_0^*\|_\infty \leq C_5\epsilon_n, \|\beta - \beta^*\|_2 \leq C_6\epsilon_n\}$, where $C_5 = 1/(2C_3\sqrt{C_1 \vee C_2})$ and $C_6 = 1/(2C_4\sqrt{C_1 \vee C_2})$. Combining the last two displays, we have

$$\Theta_n \cap U \subseteq B(\theta^*, \epsilon_n) \cap U.$$

Since $\|\mathbf{S}_0 - \mathbf{S}_0^*\|_\infty \leq \|\mathbf{\Lambda}_0 - \mathbf{\Lambda}_0^*\|_\infty$, for sufficiently large $n$ such that $\epsilon_n < \epsilon_0/(C_5 \vee C_6)$, it follows that $\Theta_n \subset U$, implying $\Theta_n \cap U = \Theta_n$. Thus, we see that

$$
\begin{aligned}
\Pi(B(\theta^*, \epsilon_n)) &\geq \Pi(B(\theta^*, \epsilon_n) \cap U) \\
&\geq \Pi(\|\mathbf{\Lambda}_0 - \mathbf{\Lambda}_0^*\|_\infty \leq C_5\epsilon_n) \cdot \Pi(\|\beta - \beta^*\|_2 \leq C_6\epsilon_n).
\end{aligned}
\tag{47}
$$

By Lemma C.3 with the specified prior (3), the first term in the right side of the last display is bounded below by $C_7 \exp(-C_8 K - C_9 K \log_-(C_5\epsilon_n))$, where $C_7$, $C_8$ and $C_9$ are positive constants depending on $p_{\min}$, $p_{\max}$, $M_1$, $M_2$, $\underline{\alpha}$, $\overline{\alpha}$ and $\rho$. Let $V_d(R)$ denote the volume of a $d$-dimensional $L^2$ norm ball of radius $R > 0$. The closed form of $V_d(R)$ is given by $V_d(R) = \pi^{d/2}/\Gamma(\frac{d}{2} + 1) \cdot R^d$ where $\Gamma$ is the gamma function. Note that $\Gamma(\frac{d}{2} + 1) \leq \Gamma(d + 1) = d! \leq d^d$ for $d \geq 1$. Then, the second term in the right side of the last display is bounded below by $C_{10}(\sqrt{\pi}/d)^d(C_6\epsilon_n)^d$ where $C_{10}$ is the lower bound of the prior on a neighborhood of $\beta^*$. Therefore, we have

$$
\begin{aligned}
\Pi(B(\theta^*, \epsilon_n)) &\geq C_7 C_{10} \exp(-C_8 K - C_9 K \log_-(C_5\epsilon_n)) \cdot (\sqrt{\pi}/d)^d(C_6\epsilon_n)^d \\
&\geq C_7 C_{10} \exp(-C_8 K - C_9 K \log_-(C_5\epsilon_n) - d\log d - d\log_-(C_6\epsilon_n)) \\
&\geq C_7 C_{10} \exp(-C_8 C_p n^\gamma - C_9 C_p n^\gamma \log_-(C_5\epsilon_n) - d\log d - d\log_-(C_6\epsilon_n)) \\
&\geq \exp(-C_{11}n\epsilon_n^2),
\end{aligned}
$$

where the third inequality holds because $K \geq C_p n^\gamma$ with a positive constant $C_p$ depending on $p_{\min}$, $p_{\max}$ and $\kappa$ as defined by $K$, and the last inequality holds by a positive constant $C_{11} = |\log(C_7 C_{10})| + C_8 C_p + C_9 C_p(|\log C_5| + 1) + |\log C_6| + 2$ because $\log_-(C\epsilon_n) \leq |\log C| + \log n$ holds for any $C > 0$ and $n^\gamma \log n \leq n\epsilon_n^2$. Thus, by Lemma 10 of [15], there exists an event $\Omega_n$ such that $\mathbb{P}_{\theta^*}^n(\Omega_n) \geq 1 - 1/(n\epsilon_n^2)$, and in $\Omega_n$,

$$\int \exp(\ell_n(\theta) - \ell_n(\theta^*))d\Pi(\theta) \geq \exp(-(C_{11} + 2)n\epsilon_n^2).\tag{48}$$

By the Kullback-Leibler inequality, note that $\mathbb{E}_{\theta^*}[\ell(\theta)]$ is maximized at $\theta = \theta^*$, meaning its first derivative at $\theta^*$ is equal to 0. Additionally, for $\theta = (\mathbf{S}_0, \beta) \in U$, note that $\mathbf{S}_0$ is uniformly bounded away from 0 and 1, and $\beta$ is bounded by assumptions (A1) and (A5). Since the covariate has bounded support by assumption (A2), by a Taylor expansion, there exists a positive constant $c_0$ depending on $M_1$, $M_2$, $L$ and $B$ such that for any $\theta \in U$, we have

$$c_0 \mathcal{D}_{\mathbb{Q}}^2(\theta, \theta^*) \leq \mathbb{E}_{\theta^*}[\ell(\theta^*)] - \mathbb{E}_{\theta^*}[\ell(\theta)] = K(p_{\theta^*}, p_\theta).$$

Combining this with (46), we have

$$C_H \mathcal{D}_{\mathbb{Q}}(\theta, \theta^*) \leq D_H(p_{\theta^*}, p_\theta), \tag{49}$$

where $C_H = \sqrt{c_0/C_1}$. Then, we have

$$\begin{aligned}
&\mathbb{E}_{\theta^*}^n[\Pi(\mathcal{D}_{\mathbb{Q}}(\theta, \theta^*) \geq MJ\epsilon_n \mid \mathbf{D}_n)\mathbb{1}\{\Omega_n\}] \\
&\quad \leq \mathbb{E}_{\theta^*}^n[\Pi(\{\mathcal{D}_{\mathbb{Q}}(\theta, \theta^*) \geq MJ\epsilon_n\} \cap U \mid \mathbf{D}_n)\mathbb{1}\{\Omega_n\}] + \mathbb{E}_{\theta^*}^n[\Pi(U^c \mid \mathbf{D}_n)] \\
&\quad \leq \mathbb{E}_{\theta^*}^n[\Pi(\Gamma_n \mid \mathbf{D}_n)\mathbb{1}\{\Omega_n\}] + c_7 \exp(-c_8 n), \tag{50}
\end{aligned}$$

where $\Gamma_n = \{\theta \in U : \mathcal{D}_H(p_{\theta^*}, p_\theta) \geq C_H M J \epsilon_n\}$ for large constants $M$ and $J$ to be chosen later, and the last inequality holds for $n \geq c_6$ by Lemma A.1 with positive constants $c_6$, $c_7$ and $c_8$ depending on $M_1$, $M_2$, $L$, $B$, $p_{\min}$, $p_{\max}$, $\underline{\alpha}$, $\overline{\alpha}$, $\rho$ and $\kappa$.

Define $\mathcal{P}_U = \{p_\theta : \theta \in U\}$ and $N_n^* = \sup_{\epsilon > \epsilon_n} N(\epsilon/36, \{p_\theta \in \mathcal{P}_U : \theta \in \Gamma_n\}, \mathcal{D}_H)$. By Lemma A.6, there exist positive constants $C_{12}$ and $C_{13}$ depending on $M_1$, $M_2$, $L$, and $B$ such that

$$\begin{aligned}
N_n^* &\leq N(\epsilon_n/36, \mathcal{P}_n^{\text{Sieve}}, \mathcal{D}_H) \\
&\leq (36C_{12}/\epsilon_n + K)^K (36C_{13}/\epsilon_n)^d \\
&\leq \exp(K \cdot \log(36C_{12}/\epsilon_n + K) + d \cdot \log(36C_{13}/\epsilon_n)) \\
&\leq \exp(C_{14} n \epsilon_n^2), \tag{51}
\end{aligned}$$

where the last inequality holds by a positive constant $C_{14}$ depending on $C_{12}$, $C_{13}$, $p_{\min}$, $p_{\max}$ and $\kappa$. In addition, by Lemma 2 of [15] and Lemma 9 of [15], applied with $\epsilon = C_H M \epsilon_n$, where $C_H M \geq 2$, there exist tests $\phi_n$ that satisfy

$$\begin{aligned}
\mathbb{E}_{\theta^*}^n \phi_n &\leq N_n^* \exp\left(-\frac{1}{2} C_H^2 M^2 n \epsilon_n^2\right) \frac{1}{1 - \exp\left(-\frac{1}{2} C_H^2 M^2 n \epsilon_n^2\right)}, \\
\sup_{\theta \in \Gamma_n} \mathbb{E}_\theta^n(1 - \phi_n) &\leq \exp\left(-\frac{1}{2} C_H^2 M^2 J^2 n \epsilon_n^2\right),
\end{aligned} \tag{52}$$

for any $J \geq 1$. Then, by (48), the first term of (50) is upper bounded by

$$\mathbb{E}_{\theta^*}^n[\Pi(\Gamma_n \mid \mathbf{D}_n)\mathbb{1}\{\Omega_n\}] \leq \mathbb{E}_{\theta^*}^n \phi_n + \exp((C_{11} + 2) n \epsilon_n^2) \sup_{\theta \in \Gamma_n} \mathbb{E}_\theta^n(1 - \phi_n). \tag{53}$$

If $M$ is sufficiently large to ensure that $C_H^2 M^2/2 - C_{14} > C_H^2 M^2/4$, by combining (51) and the first line of (52), we have

$$\begin{aligned}
\mathbb{E}_{\theta^*}^n \phi_n &\leq \exp\left(\left(C_{14} - \frac{1}{2} C_H^2 M^2\right) n \epsilon_n^2\right) \frac{1}{1 - \exp\left(-\frac{1}{2} C_H^2 M^2 n \epsilon_n^2\right)} \\
&\leq C_{15} \exp\left(-\frac{1}{4} C_H^2 M^2 n \epsilon_n^2\right),
\end{aligned}$$

where $C_{15} = (1 - \exp(-2C_{14}))^{-1}$ is a positive constant. If we set $J = 1$ and choose $M$ to be sufficiently large such that $C_H^2 M^2/2 - (C_{11} + 2) > C_H^2 M^2/4$, by the second line of (52), the second term in the right hand side of (53) is bounded by

$$\exp\left(-\frac{1}{4} C_H^2 M^2 n \epsilon_n^2\right).$$

Therefore, if we choose $M$ to be sufficiently large such that $M \geq 2\sqrt{(C_{11} + 2) \vee C_{14}}/C_H$, by combining the preceding two displays, (50) and (53), we have

$$\begin{aligned}
\mathbb{E}_{\theta^*}^n[\Pi(\mathcal{D}_{\mathbb{Q}}(\theta, \theta^*) \geq M\epsilon_n \mid \mathbf{D}_n)\mathbb{1}\{\Omega_n\}] &\leq (C_{15} + 1)\exp\left(-C_{16} n \epsilon_n^2\right) + c_7 \exp(-c_8 n) \\
&\leq C_{17} \exp\left(-(C_{16} \wedge c_8) n \epsilon_n^2\right),
\end{aligned}$$

where $C_{17} = C_{15} + c_7 + 1$ and $C_{16} = (C_{11} + 2) \vee C_{14}$ are positive constants. An application of the Markov inequality yields that

$$\mathbb{P}_{\theta^*}^n \left( \Pi(\mathcal{D}_{\mathbb{Q}}(\theta, \theta^*) \geq M\epsilon_n \mid \mathbf{D}_n) \mathbb{1}\{\Omega_n\} \geq C_{17} \exp\left(-C_{18} n \epsilon_n^2\right)\right) \leq \exp\left(-C_{18} n \epsilon_n^2\right),$$

where $C_{18} = (C_{16} \wedge c_8)/2$ be a positive constant. Note that it is easy to show that

$$\mathbb{P}_{\theta^*}^n \left( \{\Pi(\mathcal{D}_{\mathbb{Q}}(\theta, \theta^*) \geq M\epsilon_n \mid \mathbf{D}_n) \geq C_{17} \exp\left(-C_{18} n \epsilon_n^2\right)\} \cap \Omega_n \right)$$

$$\leq \mathbb{P}_{\theta^*}^n \left( \Pi(\mathcal{D}_{\mathbb{Q}}(\theta, \theta^*) \geq M\epsilon_n \mid \mathbf{D}_n) \mathbb{1}\{\Omega_n\} \geq C_{17} \exp\left(-C_{18} n \epsilon_n^2\right)\right).$$

Combining the last two displays and (48), we have

$$\mathbb{P}_{\theta^*}^n \left( \{\Pi(\mathcal{D}_{\mathbb{Q}}(\theta, \theta^*) \geq M\epsilon_n | \mathbf{D}_n) \geq C_{17} \exp\left(-C_{18} n \epsilon_n^2\right)\} \right)$$

$$\leq \mathbb{P}_{\theta^*}^n \left( \{\Pi(\mathcal{D}_{\mathbb{Q}}(\theta, \theta^*) \geq M\epsilon_n | \mathbf{D}_n) \geq C_{17} \exp\left(-C_{18} n \epsilon_n^2\right)\} \cap \Omega_n \right) + \mathbb{P}_{\theta^*}^n \left(\Omega_n\right)$$

$$\leq \mathbb{P}_{\theta^*}^n \left( \Pi(\mathcal{D}_{\mathbb{Q}}(\theta, \theta^*) \geq M\epsilon_n | \mathbf{D}_n) \mathbb{1}\{\Omega_n\} \geq C_{17} \exp\left(-C_{18} n \epsilon_n^2\right)\right) + \mathbb{P}_{\theta^*}^n \left(\Omega_n\right)$$

$$\leq \exp\left(-C_{18} n \epsilon_n^2\right) + \frac{1}{n \epsilon_n^2}.$$

Thus, we have that with probability at least $1 - \left(\exp(-C_{18} n \epsilon_n^2) + 1/n \epsilon_n^2\right)$,

$$\Pi(\mathcal{D}_{\mathbb{Q}}(\theta, \theta^*) \geq M\epsilon_n | \mathbf{D}_n) < C_{17} \exp\left(-C_{18} n \epsilon_n^2\right).$$

If we fix $M = \lceil 2\sqrt{(C_{11} + 2) \vee C_{14}/C_H} \rceil$, then the proof is complete. $\qquad\square$

### A.3 Proof of Theorem 3.2

**Lemma A.7.** *Let $\Theta' = \{(\mathbf{S}_0, \beta) \in \Theta : S_{0,K} \geq M_1, S_{0,1} \leq M_2, \|\beta\|_2 \leq D\}$, where $M_1$, $M_2$ and $D$ are some positve constants such that $0 < M_1 < M_2 < 1$, and let $\mathcal{P}' = \{p_\theta : \theta \in \Theta'\}$. Under the assumption (A2), there exist positive constants $C_1$ and $C_2$ depending only on $M_1, M_2, L$ and $D$ such that for every $\epsilon > 0$, it holds that*

$$N(\epsilon, \mathcal{P}', \mathcal{D}_H) \leq \exp(C_1/\epsilon)(C_2/\epsilon)^d.$$

*Proof.* Let $\mathcal{F}_0$ be the collection of monotone functions $f : (p_{\min}, p_{\max}) \to [\lambda_2, \lambda_1]$, where $\lambda_1 = -\log M_1$ and $\lambda_2 = -\log M_2$. Additionally, let $\Lambda_0$ denote the cumulative hazard functions with respect to the baseline complementary c.d.f. $S_0$. Then, for any $S_0$ corresponding to the vector $\mathbf{S}_0 \in \{\mathbf{S}_0 \in \mathcal{S}_0 : S_{0,K} \geq M_1, S_{0,1} \leq M_2\}$, $\Lambda_0 = -\log S_0$ belongs to $\mathcal{F}_0$.

Take any two parameters $\theta = (\mathbf{S}_0, \beta)$ and $\theta' = (\mathbf{S}_0', \beta') \in \Theta'$. By Lemma C.1 and Lemma C.2, we have

$$\mathcal{D}_H(p_\theta, p_{\theta'}) \leq C_0 \left[ \mathbb{E}_{X,P} |H_\theta(X, P) - H_{\theta'}(X, P)|^2 \right]^{1/2}$$

$$\leq C_1 \|\mathbf{S}_0 - \mathbf{S}_0'\|_{2,\mathbb{Q}} + C_2 \|\beta - \beta'\|_2$$

$$\leq C_1 \|\mathbf{\Lambda}_0 - \mathbf{\Lambda}_0'\|_{2,\mathbb{Q}} + C_2 \|\beta - \beta'\|_2, \qquad (54)$$

where $\mathbb{E}_{X,P}$ denotes the expectation with respect to the covariate $X$ and the price $P$, and $C_0$, $C_1$ and $C_2$ are positive constants depending on $M_1, M_2, L$ and $D$.

Let $m := N(\epsilon/(2C_1), \mathcal{F}_0, \|\cdot\|_{2,\mathbb{Q}})$ and $l := N(\epsilon/(2C_2), \mathcal{B}', \|\cdot\|_2)$, where $\mathcal{B}' = \{\beta \in \mathbb{R}^d : \|\beta\|_2 \leq D\}$. This definition implies that there exist $\Lambda_{0,1}, \ldots, \Lambda_{0,m} \in \mathcal{F}_0$ such that for every $\Lambda_0 \in \mathcal{F}_0$, the inequality $\|\Lambda_0 - \Lambda_{0,i}\|_{2,\mathbb{Q}} < \epsilon/(2C_1)$ holds for some $1 \leq i \leq m$. Similarly, there exist $\beta_1, \ldots, \beta_l \in \mathcal{B}'$ such that for every $\beta \in \mathcal{B}'$, $\|\beta - \beta_j\|_2 < \epsilon/(2C_2)$ holds for some $1 \leq j \leq l$. Let $\mathbf{S}_{0,i} = (S_{0,i}(g_1), \ldots, S_{0,i}(g_K))$ where $S_{0,i} = \exp(-\Lambda_{0,i})$ and $\theta_{ij} = (\mathbf{S}_{0,i}, \beta_j) \in \Theta'$ for $i = 1, \ldots, m$ and $j = 1, \ldots, l$. By (54), for any $\theta = (\mathbf{S}_0, \beta) \in \Theta'$, there exists $\theta_{ij}$ for some $i \in \{1, \ldots, m\}$ and $j \in \{1, \ldots, l\}$ such that

$$\mathcal{D}_H(p_\theta, p_{\theta_{ij}}) \leq C_1 \|\Lambda_0 - \Lambda_{0,i}\|_{2,\mathbb{Q}} + C_2 \|\beta - \beta_j\|_2 \leq \epsilon.$$

Consequently, the covering number $N(\epsilon, \mathcal{P}', \mathcal{D}_H)$ is of order $ml$. By Proposition C.8 of [16], note that $m \leq N_{[]}(\epsilon/(2C_1), \mathcal{F}_0, \|\cdot\|_{2,\mathbb{Q}}) \leq \exp(2C_1 C_3 \lambda_1/\epsilon)$, where $C_3$ is a universal constant. Furthermore, by Proposition C.2 of [16], $l \leq (6DC_2/\epsilon)^d$. Therefore, we have

$$N(\epsilon, \mathcal{P}', \mathcal{D}_H) \leq \exp(C_4/\epsilon)(C_5/\epsilon)^d,$$

where $C_4 = 2C_1 C_3 \lambda_1$ and $C_5 = 6DC_2$ are positive constants depending only on $M_1, M_2, L$ and $D$.

$\qquad\square$

*Proof of Theorem 3.2.* Recall the grid support $\mathcal{G} = \{g_k : k = 1, \ldots, K\}$, where each grid point $g_k$ is defined as $g_k = p_{\min} + k\delta$ with $\delta = \kappa n^{-\gamma}$ for some constant $\kappa > 0$. Let $J = \lceil (p_{\max} - p_{\min})/(\kappa\epsilon_n) \rceil$ and define $(k_1, \ldots, k_J)$ as a subsequence of $[K]$ such that $p_{\min} + (j-1)\kappa\epsilon_n < g_{k_j} \le p_{\min} + (j-1)\kappa\epsilon_n + \delta$ for $j = 1, \ldots, J-1$, and set $k_J = K$. Suppose that $|S_{0,k_j} - S_{0,k_j}^*| < \epsilon_n$ for every $j = 1, \ldots, J$. Then, for any $k \in [K]$ with $k_j \le k < k_{j+1}$ for $j = 1, \ldots, J-2$, we have

$$
\begin{aligned}
S_{0,k}^* - S_{0,k} &\le S_{0,k}^* - S_{0,k_{j+1}} \\
&\le |S_{0,k_{j+1}}^* - S_{0,k_{j+1}}| + |S_{0,k}^* - S_{0,k_{j+1}}^*| \\
&< \epsilon_n + L_0|g_k - g_{k_{j+1}}| \\
&\le \epsilon_n + L_0|g_{k_j} - g_{k_{j+1}}| \\
&\le \epsilon_n + L_0(\kappa\epsilon_n + \delta) \\
&\le (2L_0\kappa + 1)\epsilon_n,
\end{aligned}
$$

where the third inequality holds by our assumption and $L_0$-Lipschitz continuity of $S_0^*$, with $L_0$ being a positive constant because (A5) is assumed, the fifth inequality holds by the definition of $k_j$, and the last inequality holds because $\delta \le \kappa\epsilon_n$. Note that $|g_{k_{J-1}} - g_{k_J}| < 2\kappa\epsilon_n$ by the definition of $J$. Then, for any $k \in [K]$ with $k_{J-1} \le k \le k_J$, we have

$$
S_{0,k}^* - S_{0,k} < (2L_0\kappa + 1)\epsilon_n.
$$

Combining the preceding two displays, there exists a positive constant $C_0 = 2L_0\kappa + 1$ such that $S_{0,k}^* - S_{0,k} < C_0\epsilon_n$ for any $k \in [K]$. Similarly, for any $k \in [K]$, we have $S_{0,k} - S_{0,k}^* < C_0\epsilon_n$. Therefore, we have

$$
\Pi\left(\|\mathbf{S}_0 - \mathbf{S}_0^*\|_\infty \le C_0\epsilon_n\right) \ge \Pi\left(\|\mathbf{S}_0 - \mathbf{S}_0^*\|_{\infty, J} \le \epsilon_n\right), \tag{55}
$$

where $\|\mathbf{S}_0 - \mathbf{S}_0^*\|_{\infty, J} = \max_{j=1,\ldots,J} |S_{0,k_j} - S_{0,k_j}^*|$.

Note that there exist constants $M_1$ and $M_2$ such that $0 < M_1 \le S_0^*(v) \le M_2 < 1$ for any $v \in [p_{\min}, p_{\max}]$ by assumption (A5). Let $U = \{(\mathbf{S}_0, \beta) \in \Theta : \|\mathbf{S}_0 - \mathbf{S}_0^*\|_\infty \vee \|\beta - \beta^*\|_2 < \epsilon_0\}$ be a neighborhood of $\theta^*$, where $\epsilon_0$ is a positive constant that can be chosen as $\epsilon_0 = ((M_1 \wedge (1 - M_2))/2) \wedge B$ to ensure that $U \subseteq \Theta$. Given in (47) of Theorem 3.1, there exist positive constants $C_1$ and $C_2$ depending on $M_1, M_2, L$ and $B$ such that for sufficiently large $n$ with $\epsilon_n < \epsilon_0/(C_1 \vee C_2)$,

$$
\Pi\left(B(\theta^*, \epsilon_n)\right) \ge \Pi\left(\|\mathbf{\Lambda}_0 - \mathbf{\Lambda}_0^*\|_\infty \le C_1\epsilon_n\right) \cdot \Pi\left(\|\beta - \beta^*\|_2 \le C_2\epsilon_n\right).
$$

Note that for $\theta \in U$, $\|\mathbf{S}_0 - \mathbf{S}_0^*\|_\infty \asymp \|\mathbf{\Lambda}_0 - \mathbf{\Lambda}_0^*\|_\infty$ and $\|\mathbf{S}_0 - \mathbf{S}_0^*\|_{\infty, J} \asymp \|\mathbf{\Lambda}_0 - \mathbf{\Lambda}_0^*\|_{\infty, J}$, where constants in $\asymp$ depend on $M_1, M_2, L$ and $B$. Thus, the inequality (55) implies that

$$
\Pi\left(\|\mathbf{\Lambda}_0 - \mathbf{\Lambda}_0^*\|_\infty \le C_0'C_0\epsilon_n\right) \ge \Pi\left(\|\mathbf{\Lambda}_0 - \mathbf{\Lambda}_0^*\|_{\infty, J} \le \epsilon_n\right),
$$

where $C_0'$ is a positive constants depending on $M_1, M_2, L$ and $B$. Combining the preceding two displays, for sufficiently large $n$ such that $\epsilon_n < \epsilon_0/(C_1 \vee C_2)$ and $\delta \le \kappa(C_0'C_0)^{-1}C_1\epsilon_n$, we have

$$
\Pi\left(B(\theta^*, \epsilon_n)\right) \ge \Pi\left(\|\mathbf{\Lambda}_0 - \mathbf{\Lambda}_0^*\|_{\infty, J} \le (C_0'C_0)^{-1}C_1\epsilon_n\right) \cdot \Pi\left(\|\beta - \beta^*\|_2 \le C_2\epsilon_n\right).
$$

Therefore, we have

$$
\begin{aligned}
\Pi\left(B(\theta^*, \epsilon_n)\right) &\ge C_3 C_6 \exp(-C_4 J - C_5 J \log_-((C_0'C_0)^{-1}C_1\epsilon_n) - d\log d - d\log_-(C_2\epsilon_n)) \\
&\ge C_3 C_6 \exp(-C_4 C_7 \epsilon_n^{-1} - C_5 C_7 \epsilon_n^{-1} \log_-((C_0'C_0)^{-1}C_1\epsilon_n) - d\log d - d\log_-(C_2\epsilon_n)) \\
&\ge \exp(-C_8 n\epsilon_n^2),
\end{aligned}
$$

where the first inequality holds by Lemma C.3 with positive constants $C_3, C_4, C_5$ depending on $p_{\min}, p_{\max}, M_1, M_2, \underline{\alpha}, \overline{\alpha}$ and $\rho$, and $C_6$ serving as the lower bound of the prior on a neighborhood of $\beta^*$, the second inequality holds because $J \le C_7 \epsilon_n^{-1}$ holds by the definition of $J$ with a positive constant $C_7$ depending on $p_{\min}, p_{\max}$ and $\kappa$, and the third inequality holds by a positive constant $C_8 = |\log(C_3 C_6)| + C_4 C_7 + C_5 C_7(|\log((C_0'C_0)^{-1}C_1)| + 1) + |\log C_2| + 2$ because $\epsilon_n^{-1}\log_-(\epsilon_n) \le n\epsilon_n^2$ and $d\log_-(\epsilon_n) \le n\epsilon_n^2$. Thus, by Lemma 10 of [15], there exists an event $\Omega_n$ such that $\mathbb{P}_{\theta^*}^n(\Omega_n) \ge 1 - 1/n\epsilon_n^2$, and in $\Omega_n$,

$$
\int \exp(\ell_n(\theta) - \ell_n(\theta^*))d\Pi(\theta) \ge \exp(-(C_8 + 2)n\epsilon_n^2). \tag{56}
$$

By (49) in the proof of Theorem 3.1, there exists a positive constant $C_H$ depending on $M_1$, $M_2$, $L$ and $B$ such that $C_H \mathcal{D}_\mathbb{Q}(\theta, \theta^*) \leq \mathcal{D}_H(p_{\theta^*}, p_\theta)$ for $\theta \in U$. Then, we have

$$
\begin{aligned}
\mathbb{E}_{\theta^*}^n &\left[\Pi(\mathcal{D}_\mathbb{Q}(\theta, \theta^*) \geq MJ\epsilon_n \mid \mathbf{D}_n) \mathbb{1}\{\Omega_n\}\right] \\
&\leq \mathbb{E}_{\theta^*}^n \left[\Pi(\{\mathcal{D}_\mathbb{Q}(\theta, \theta^*) \geq MJ\epsilon_n\} \cap U \mid \mathbf{D}_n) \mathbb{1}\{\Omega_n\}\right] + \mathbb{E}_{\theta^*}^n \left[\Pi(U^c \mid \mathbf{D}_n)\right] \\
&\leq \mathbb{E}_{\theta^*}^n \left[\Pi(\Gamma_n \mid \mathbf{D}_n) \mathbb{1}\{\Omega_n\}\right] + \mathbb{E}_{\theta^*}^n \left[\Pi(U^c \mid \mathbf{D}_n)\right],
\end{aligned}
\tag{57}
$$

where $\Gamma_n = \{\theta \in U : \mathcal{D}_H(p_{\theta^*}, p_\theta) \geq C_H MJ\epsilon_n\}$ for large constants $M$ and $J$ to be chosen later.

Define $\mathcal{P}_U = \{p_\theta : \theta \in U\}$ and $N_n^* = \sup_{\epsilon > \epsilon_n} N(\epsilon/36, \{p_\theta \in \mathcal{P}_U : \theta \in \Gamma_n\}, \mathcal{D}_H)$. By Lemma A.7, there exist positive constants $C_9$ and $C_{10}$ depending on $M_1$, $M_2$, $L$ and $B$ such that

$$
\begin{aligned}
N_n^* &\leq N\left(\epsilon_n/36, \mathcal{P}_n^{\text{Sieve}}, \mathcal{D}_H\right) \\
&\leq \exp(36 C_9 \epsilon_n^{-1})(36 C_{10} \epsilon_n^{-1})^d \\
&\leq \exp(36 C_9 n\epsilon_n^2 + d\log(36 C_{10} n\epsilon_n^2)) \\
&\leq \exp(C_{11} n\epsilon_n^2),
\end{aligned}
\tag{58}
$$

where the third inequality holds because $\epsilon_n^{-1} \leq n\epsilon_n^2$, and the last inequality holds by $C_{11} = 36 C_9 + |\log(36 C_{10})| + 2$ because $d\log(n\epsilon_n^2) \leq 2n\epsilon_n^2$. In addition, by Lemma 2 of [15] and Lemma 9 of [15], applied with $\epsilon = C_H M\epsilon_n$, where $C_H M \geq 2$, there exist tests $\phi_n$ that satisfy

$$
\begin{aligned}
\mathbb{E}_{\theta^*}^n \phi_n &\leq N_n^* \exp\left(-\frac{1}{2} C_H^2 M^2 n\epsilon_n^2\right) \frac{1}{1 - \exp\left(-\frac{1}{2} C_H^2 M^2 n\epsilon_n^2\right)}, \\
\sup_{\theta \in \Gamma_n} \mathbb{E}_\theta^n (1 - \phi_n) &\leq \exp\left(-\frac{1}{2} C_H^2 M^2 J^2 n\epsilon_n^2\right),
\end{aligned}
\tag{59}
$$

for any $J \geq 1$. Then, by (56), the first term of (57) is upper bounded by

$$
\mathbb{E}_{\theta^*}^n \left[\Pi(\Gamma_n \mid \mathbf{D}_n) \mathbb{1}\{\Omega_n\}\right] \leq \mathbb{E}_{\theta^*}^n \phi_n + \exp((C_8 + 2) n\epsilon_n^2) \sup_{\theta \in \Gamma_n} \mathbb{E}_\theta^n (1 - \phi_n).
\tag{60}
$$

If $M$ is sufficiently large to ensure that $C_H^2 M^2/2 - C_{11} > C_H^2 M^2/4$, by combining (58) and the first line of (59), we have

$$
\begin{aligned}
\mathbb{E}_{\theta^*}^n \phi_n &\leq \exp\left(\left(C_{11} - \frac{1}{2} C_H^2 M^2\right) n\epsilon_n^2\right) \frac{1}{1 - \exp\left(-\frac{1}{2} C_H^2 M^2 n\epsilon_n^2\right)} \\
&\leq C_{12} \exp\left(-\frac{1}{4} C_H^2 M^2 n\epsilon_n^2\right),
\end{aligned}
$$

where $C_{12} = (1 - \exp(-2C_{11}))^{-1}$ is a positive constant. If we set $J = 1$ and choose $M$ to be sufficiently large such that $C_H^2 M^2/2 - (2 + C_8) > C_H^2 M^2/4$, by the second line of (59), the second term in the right hand side of (60) is bounded by

$$
\exp\left(-\frac{1}{4} C_H^2 M^2 n\epsilon_n^2\right).
$$

Therefore, if we choose $M$ to be sufficiently large such that $M \geq 2\sqrt{(2 + C_8) \vee C_{11}}/C_H$, by combining the preceding two displays, (57) and (60), we have

$$
\mathbb{E}_{\theta^*}^n \left[\Pi(\mathcal{D}_\mathbb{Q}(\theta, \theta^*) \geq M\epsilon_n \mid \mathbf{D}_n) \mathbb{1}\{\Omega_n\}\right] \leq (C_{12} + 1) \exp\left(-C_{13} n\epsilon_n^2\right) + \mathbb{E}_{\theta^*}^n \left[\Pi(U^c \mid \mathbf{D}_n)\right],
$$

where $C_{13} = (C_8 + 2) \vee C_{11}$ is a positive constant. By Lemma A.1, if $\gamma < 2/3$, the second term on the right-hand side of the last display is bounded by $c_2 \exp(-c_3 n)$ for $n \geq c_1$, and if $\gamma \geq 2/3$, it is bounded by $c_5 \exp(-c_6 n^{1/3})$ for $n \geq c_4$, where $c_1, \ldots, c_6$ are positive constants depending on $M_1$, $M_2$, $L$, $B$, $p_{\min}$, $p_{\max}$, $\kappa$, $\underline{\alpha}$, $\overline{\alpha}$ and $\rho$. Then, we have

$$
\mathbb{E}_{\theta^*}^n \left[\Pi(\mathcal{D}_\mathbb{Q}(\theta, \theta^*) \geq M\epsilon_n \mid \mathbf{D}_n) \mathbb{1}\{\Omega_n\}\right] \leq
\begin{cases}
C_{14} \exp\left(-C_{15} n\epsilon_n^2\right), & \text{if } \gamma < \frac{2}{3}, \\
C_{16} \exp\left(-C_{17} n^{\frac{1}{3}}\right), & \text{if } \gamma \geq \frac{2}{3},
\end{cases}
$$

where $C_{14} = C_{12} + 1 + c_2$, $C_{15} = C_{13} \wedge c_3$, $C_{16} = C_{12} + 1 + c_5$ and $C_{17} = C_{13} \wedge c_6$. By the Markov inequality and (56), we have that if $\gamma < 2/3$,

$$
\Pi(\mathcal{D}_\mathbb{Q}(\theta, \theta^*) \geq M\epsilon_n \mid \mathbf{D}_n) \leq C_{14} \exp\left(-C_{18} n\epsilon_n^2\right),
$$

with probability at least $1 - (\exp(-C_{18}n\epsilon_n^2) + 1/n\epsilon_n^2)$, where $C_{18} = C_{15}/2$, and if $\gamma \geq 2/3$,

$$\Pi(\mathcal{D}_{\mathbb{Q}}(\theta, \theta^*) \geq M\epsilon_n \mid \mathbf{D}_n) \leq C_{16} \exp\left(-C_{19}n^{\frac{1}{3}}\right),$$

with probability at least $1 - (\exp(-C_{19}n^{1/3}) + 1/n\epsilon_n^2)$, where $C_{19} = C_{17}/2$. If we fix $M = \lceil 2\sqrt{(C_8 + 2) \vee C_{11}}/C_H \rceil$, then the proof is complete.

$\square$

## B    Proofs for Section 5

### B.1    Proof of Lemma 5.1

*Proof.* We first consider the epoch $l - 1$ under the condition $\gamma_{l-1} < 1/3$. Let $q_l(\cdot \mid x)$ be the conditional probability mass function of $P_t$ given $X_t = x$ for $t \in \mathcal{E}_l$ in epoch $l$. By Lemma C.5, $q_l(\cdot \mid x)$ satisfies the assumption (A4) for every epoch $l$. Then, by Theorem 3.1, there exist positive constants $c_1, c_2, c_3$ and $c_4$ depending on $L, B, p_{\min}, p_{\max}, \kappa, \alpha, \rho$ such that for $l \geq \lceil \log_2(c_4/n_1) \rceil + 1$,

$$\Pi(\mathcal{D}_{\mathbb{Q}_{l-1}}(\theta, \theta^*) \geq c_1\epsilon_{l-1} \mid \mathbf{D}_{l-1}) \leq c_2 \exp(-c_3 n_{l-1}\epsilon_{l-1}^2) \tag{61}$$

with probability at least $1 - \exp(-c_3 n_{l-1}\epsilon_{l-1}^2) - 1/(n_{l-1}\epsilon_{l-1}^2)$. We partition the parameter space $\widetilde{\Theta}$ into two subsets $\widetilde{\Theta}_{l-1,1} = \{\theta \in \widetilde{\Theta} : \mathcal{D}_{\mathbb{Q}_{l-1}}(\theta, \theta^*) < c_1\epsilon_{l-1}\}$ and $\widetilde{\Theta}_{l-1,2} = \{\theta \in \widetilde{\Theta} : \mathcal{D}_{\mathbb{Q}_{l-1}}(\theta, \theta^*) \geq c_1\epsilon_{l-1}\}$. Then, we can decompose $\widehat{\theta}^{l-1}$ as

$$\begin{aligned}
\widehat{\theta}^{l-1} &= \int_{\widetilde{\Theta}} \theta \, d\widetilde{\Pi}(\theta \mid \mathbf{D}_{l-1}) \\
&= \int_{\widetilde{\Theta}_{l-1,1}} \theta \, d\widetilde{\Pi}(\theta \mid \mathbf{D}_{l-1}) + \int_{\widetilde{\Theta}_{l-1,2}} \theta \, d\widetilde{\Pi}(\theta \mid \mathbf{D}_{l-1}) \\
&= (1 - \tau_{l-1})\widehat{\theta}_1^{l-1} + \tau_{l-1}\widehat{\theta}_2^{l-1}, \tag{62}
\end{aligned}$$

where $\tau_{l-1} = \widetilde{\Pi}(\widetilde{\Theta}_{l-1,2} \mid \mathbf{D}_{l-1})$. Here, $\widehat{\theta}_1^{l-1}$ and $\widehat{\theta}_2^{l-1}$ are the mean estimates of the probability measures resulting from the restriction and normalization of the truncated posterior distribution on the sets $\widetilde{\Theta}_{l-1,1}$ and $\widetilde{\Theta}_{l-1,2}$, respectively. It is easy to check that the function $\theta \mapsto \mathcal{D}_{\mathbb{Q}_{l-1}}(\theta, \theta^*)$ is convex and bounded over the domain $\widetilde{\Theta}$. By Jensen's inequality, we have

$$\begin{aligned}
\mathcal{D}_{\mathbb{Q}_{l-1}}(\widehat{\theta}_1^{l-1}, \theta^*) &\leq \int_{\widetilde{\Theta}_{l-1,1}} \mathcal{D}_{\mathbb{Q}_{l-1}}(\theta, \theta^*) \, d\widetilde{\Pi}_1(\theta \mid \mathbf{D}_{l-1}) \\
&< c_1\epsilon_{l-1}, \tag{63}
\end{aligned}$$

where $\widetilde{\Pi}_1(\cdot \mid \mathbf{D}_{l-1})$ be the probability measure obtained by restricting and renormalizing $\widetilde{\Pi}(\cdot \mid \mathbf{D}_{l-1})$ to $\widetilde{\Theta}_{l-1,1}$, and the last inequality holds by the definition of $\widetilde{\Theta}_{l-1,1}$. On the event that the inequality (61) holds, we have that with probability at least $1 - \exp(-c_3 n_{l-1}\epsilon_{l-1}^2) - 1/(n_{l-1}\epsilon_{l-1}^2)$, for $l \geq \lceil \log_2(c_4/n_1) \rceil + 1$, it follows that

$$\begin{aligned}
\mathcal{D}_{\mathbb{Q}_{l-1}}(\widehat{\theta}^{l-1}, \theta^*) &\leq (1 - \tau_{l-1})\mathcal{D}_{\mathbb{Q}_{l-1}}(\widehat{\theta}_1^{l-1}, \theta^*) + \tau_{l-1}\mathcal{D}_{\mathbb{Q}_{l-1}}(\widehat{\theta}_2^{l-1}, \theta^*) \\
&< c_1\epsilon_{l-1} + \frac{\Pi(\widetilde{\Theta}_{l-1,2} \mid \mathbf{D}_{l-1})}{\Pi(\widetilde{\Theta} \mid \mathbf{D}_{l-1})}\mathcal{D}_{\mathbb{Q}_{l-1}}(\widehat{\theta}_2^{l-1}, \theta^*) \\
&\leq c_1\epsilon_{l-1} + \frac{c_2 \exp(-c_3 n_{l-1}\epsilon_{l-1}^2)}{1 - c_2 \exp(-c_3 n_{l-1}\epsilon_{l-1}^2)} \cdot (1 + \sqrt{d}(a \vee b) + B) \\
&\leq C_1\epsilon_{l-1},
\end{aligned}$$

where the first inequality holds because of the convexity of the function $\theta \mapsto \mathcal{D}_{\mathbb{Q}_{l-1}}(\theta, \theta^*)$ and (62), and the second inequality holds by (63) and the definition of $\tau_{l-1}$. The third inequality follows from $\Pi(\widetilde{\Theta} \mid \mathbf{D}_{l-1}) \geq 1 - \Pi(\widetilde{\Theta}_{l-1,2} \mid \mathbf{D}_{l-1})$, combined with inequality (61) and the boundedness of $\mathcal{D}_{\mathbb{Q}_{l-1}}$ over $\widetilde{\Theta}$ under the assumption (A1). The last inequality holds with a positive constant $C_1$ depending on $c_1, c_2, c_3, a, b$ and $B$, since $\sqrt{d}\exp(-c_3 n_{l-1}\epsilon_{l-1}^2)/(1 - c_2 \exp(-c_3 n_{l-1}\epsilon_{l-1}^2)) \lesssim \epsilon_{l-1}$.

By similar arguments as before, for the epoch $l-1$ under the condition $\gamma_{l-1} \geq 1/3$, by Theorem 3.2, there exist positive constants $c_5, c_6, c_7$ and $c_8$ depending on $L, B, p_{\min}, p_{\max}, \kappa, \alpha$ and $\rho$ such that for $l \geq \lceil \log_2(c_8/n_1) \rceil + 1$,

$$\Pi(\mathcal{D}_{\mathbb{Q}_{l-1}}(\theta, \theta^*) \geq c_5 \epsilon_{l-1} \mid \mathbf{D}_{l-1}) \leq c_6 \xi_{k-1},$$

where

$$\xi_{k-1} = \begin{cases} \exp(-c_7 n_{l-1} \epsilon_{l-1}^2) & \text{if } \frac{1}{3} \leq \gamma_{l-1} < \frac{2}{3}, \\ \exp(-c_7 n_{l-1}^{1/3}) & \text{if } \gamma_{l-1} \geq \frac{2}{3}, \end{cases}$$

with probability at least $1 - \xi_{k-1} - 1/(n_{l-1} \epsilon_{l-1}^2)$. Since $\exp(-c_7 n_{l-1} \epsilon_{l-1}^2) \leq \exp(-c_7 n_{l-1}^{1/3})$, we unify the cases where $\gamma_{l-1}$ is either greater than or less than $2/3$ and obtain the bound

$$\Pi(\mathcal{D}_{\mathbb{Q}_{l-1}}(\theta, \theta^*) \geq c_5 \epsilon_{l-1} \mid \mathbf{D}_{l-1}) \leq c_6 \exp(-c_7 n_{l-1}^{1/3}), \tag{64}$$

with probability at least $1 - \exp(-c_7 n_{l-1}^{1/3}) - 1/(n_{l-1} \epsilon_{l-1}^2)$. Similarly, for $l \geq \lceil \log_2(c_8/n_1) \rceil + 1$, we have

$$\mathcal{D}_{\mathbb{Q}_{l-1}}(\widehat{\theta}^{l-1}, \theta^*) \leq C_2 \epsilon_{l-1},$$

with probability at least $1 - \exp(-c_7 n_{l-1}^{1/3}) - 1/(n_{l-1} \epsilon_{l-1}^2)$, where $C_2$ is a positive constant depending on $c_5, c_6, c_7, a, b$ and $B$. The proof is then complete. $\square$

## B.2 Proof of Theorem 5.2

**Lemma B.1.** *Suppose that assumptions (A1)-(A3), (A5) and (B1)-(B2) hold. Suppose that the prior distribution $\Pi$ is specified as in (4), and the policy $\pi_l$ for each epoch $l$ is defined by (5). Then, there exist positive constants $C_1, C_2, C_3$ and $C_4$ depending on $L, B, p_{\min}, p_{\max}, \kappa, \alpha, \rho, a, b, \eta_1, \eta_2$ and $n_1$ such that for $l \geq C_1$,*

$$\sum_{t \in \mathcal{E}_l} r(t) \leq C_2 n_l \mathcal{D}_{\mathbb{Q}_{l-1}}(\widehat{\theta}^{l-1}, \theta^*) + C_3 \left( n_l^{\frac{1+\gamma_l}{2}} \wedge n_l^{\frac{2}{3}} \right) (\log n_l)^{\frac{1}{2}}$$

*with probability at least $1 - (\exp(-C_4 n_l^{1/3}) + 3/n_l^2)$.*

*Proof.* The regret in epoch $l$ is decomposed and upper bounded by

$$\sum_{t \in \mathcal{E}_l} r(t) = \sum_{t \in \mathcal{E}_l} \left( P_t^* H_{\theta^*}(X_t, P_t^*) - P_t H_{\theta^*}(X_t, P_t) \right)$$

$$= \sum_{t \in \mathcal{E}_l} \left\{ \left( P_t^* H_{\theta^*}(X_t, P_t^*) - P_t^* H_{\widehat{\theta}^{l-1}}(X_t, P_t^*) \right) + \left( P_t^* H_{\widehat{\theta}^{l-1}}(X_t, P_t^*) - P_t H_{\widehat{\theta}^{l-1}}(X_t, P_t) \right) \right.$$

$$\left. + \left( P_t H_{\widehat{\theta}^{l-1}}(X_t, P_t) - P_t H_{\theta^*}(X_t, P_t) \right) \right\}$$

$$\leq \underbrace{\sum_{t \in \mathcal{E}_l} \left( P_t^* H_{\widehat{\theta}^{l-1}}(X_t, P_t^*) - P_t H_{\widehat{\theta}^{l-1}}(X_t, P_t) \right)}_{(i)} + \underbrace{p_{\max} \sum_{t \in \mathcal{E}_l} \left| H_{\widehat{\theta}^{l-1}}(X_t, P_t^*) - H_{\theta^*}(X_t, P_t^*) \right|}_{(ii)}$$

$$+ \underbrace{p_{\max} \sum_{t \in \mathcal{E}_l} \left| H_{\widehat{\theta}^{l-1}}(X_t, P_t) - H_{\theta^*}(X_t, P_t) \right|}_{(iii)}, \tag{65}$$

where the last inequality holds because any $P_t$ and $P_t^*$ lie in $\mathcal{G} \subset [p_{\min}, p_{\max}]$ almost surely. Note that $\{(X_t, P_t, P_t^*)\}_{t \in \mathcal{E}_l}$ is an i.i.d. sample of joint distribution which satisfies $P_t \sim \mathbb{Q}_l$, $P_t^* \sim \mathbb{Q}^*$ and $X_t \sim \mathbb{P}_X$. Since $P_t^* H_{\widehat{\theta}^{l-1}}(X_t, P_t^*) - P_t H_{\widehat{\theta}^{l-1}}(X_t, P_t) \in [-p_{\max}, p_{\max}]$, by Hoeffding's inequality, it holds that

$$\text{(i)} < 2 p_{\max} n_l^{\frac{1}{2}} (\log n_l)^{\frac{1}{2}} + \mathbb{E} \left[ \sum_{t \in \mathcal{E}_l} \left( P_t^* H_{\widehat{\theta}^{l-1}}(X_t, P_t^*) - P_t H_{\widehat{\theta}^{l-1}}(X_t, P_t) \right) \right], \tag{66}$$

with probability at least $1 - 1/n_l^2$. Let $\widehat{P}_t \in \text{argmax}_{p \in \mathcal{G}} \, pH_{\widehat{\theta}^{l-1}}(X_t, p)$, and let $U$ denote a uniform random variable over $\mathcal{G}$. By the design of Algorithm 1, we have $P_t = R\widehat{P}_t + (1 - R)U$, where $R$ is a Bernoulli random variable with success probability $1 - \eta_l$. By the law of total expectation, we have

$$\mathbb{E}\left[P_t H_{\widehat{\theta}^{l-1}}(X_t, P_t)\right] = (1 - \eta_l)\mathbb{E}\left[\widehat{P}_t H_{\widehat{\theta}^{l-1}}(X_t, \widehat{P}_t)\right] + \eta_l \mathbb{E}\left[U H_{\widehat{\theta}^{l-1}}(X_t, U)\right].$$

By substituting this in (66), the second term of (66) is bounded by

$$\mathbb{E}\left[\sum_{t \in \mathcal{E}_l} \left(P_t^* H_{\widehat{\theta}^{l-1}}(X_t, P_t^*) - P_t H_{\widehat{\theta}^{l-1}}(X_t, P_t)\right)\right] = (1 - \eta_l)\sum_{t \in \mathcal{E}_l} \mathbb{E}\left[P_t^* H_{\widehat{\theta}^{l-1}}(X_t, P_t^*) - \widehat{P}_t H_{\widehat{\theta}^{l-1}}(X_t, \widehat{P}_t)\right]$$

$$+ \eta_l \sum_{t \in \mathcal{E}_l} \mathbb{E}\left[P_t^* H_{\widehat{\theta}^{l-1}}(X_t, P_t^*) - U H_{\widehat{\theta}^{l-1}}(X_t, U)\right]$$

$$\leq \eta_l \sum_{t \in \mathcal{E}_l} p_{\max}$$

$$\leq C_0 \left(n_l^{\frac{1+\gamma_l}{2}} \wedge n_l^{\frac{2}{3}}\right)(\log n_l)^{\frac{1}{2}}$$

where the first inequality holds because $P_t^* H_{\widehat{\theta}^{l-1}}(X_t, P_t^*) - \widehat{P}_t H_{\widehat{\theta}^{l-1}}(X_t, \widehat{P}_t) \leq 0$ by the definition of $\widehat{P}_t$, and the last inequality holds by the definition of $\eta_l$ (6) and (7) with a positive constant $C_0$ depending on $p_{\min}, p_{\max}, \kappa, \eta_1$ and $\eta_2$. Combining this with (66), we have

$$\text{(i)} < C_1 \left(n_l^{\frac{1+\gamma_l}{2}} \wedge n_l^{\frac{2}{3}}\right)(\log n_l)^{\frac{1}{2}}, \tag{67}$$

with probability at least $1 - 1/n_l^2$, where $C_1 = 2p_{\max} + C_0$ is a positive constant.

For (ii) and (iii), by Lemma C.6, for every $\epsilon > 0$, there exist positive constants $c_1$ and $c_2$ depending on $L, B, p_{\min}, p_{\max}, \kappa, \alpha, \rho, a, b, n_1$ and $\epsilon$ such that for large $l \geq c_1$, we have

$$\|\widehat{\mathbf{S}}_0^{l-1} - \mathbf{S}_0^*\|_\infty + \|\widehat{\beta}^{l-1} - \beta^*\|_2 < \epsilon,$$

with probability at least $1 - \exp(-c_2 n_{l-1}^{1/3})$. Note that there exist constants $M_1$ and $M_2$ such that $0 < M_1 \leq S_0^*(p_{\max}) < S_0^*(p_{\min}) \leq M_2 < 1$ by assumption (A5). Take $\epsilon = C_2$, where $C_2 = ((M_1 \wedge (1 - M_2))/2 \wedge B)$. On the event that the preceding inequality holds, we have

$$\|\widehat{\mathbf{S}}_0^{l-1} - \mathbf{S}_0^*\|_\infty + \|\widehat{\beta}^{l-1} - \beta^*\|_2 < C_2.$$

This implies that $\widehat{S}_{0,1}^{l-1} > M_1/2 > 0$, $\widehat{S}_{0,K}^{l-1} < (1 + M_2)/2 < 1$ and $\|\widehat{\beta}^{l-1}\|_2 < 2B$. Then, by Lemma C.2, for any $p \in \mathcal{G}$ and $l \geq c_1$, with probability at least $1 - \exp(-c_2 n_{l-1}^{1/3})$, we have

$$|H_{\widehat{\theta}^{l-1}}(X_t, p) - H_{\theta^*}(X_t, p)| \leq C_3|\widehat{S}_0^{l-1}(p) - S_0^*(p)| + C_4\|\widehat{\beta}^{l-1} - \beta^*\|_2, \tag{68}$$

where $C_3$ and $C_4$ are positive constants depending on $M_1, M_2, L$ and $B$.

Let $\Omega_1$ be the event that (68) holds. For (ii), under the event $\Omega_1$, we have

$$\text{(ii)} < C_3 \sum_{t \in \mathcal{E}_l} |\widehat{S}_0^{l-1}(P_t^*) - S_0^*(P_t^*)| + C_4 \sum_{t \in \mathcal{E}_l} \|\widehat{\beta}^{l-1} - \beta^*\|_2$$

$$= C_3 \sum_{t \in \mathcal{E}_l} |\widehat{S}_0^{l-1}(P_t^*) - S_0^*(P_t^*)| + C_4 n_l \|\widehat{\beta}^{l-1} - \beta^*\|_2.$$

Since $|\widehat{S}_0^{l-1}(P_t^*) - S_0^*(P_t^*)| \leq 1$, by Hoeffding's inequality, there exists an event $\Omega_2$ such that $\mathbb{P}(\Omega_2) \geq 1 - 1/n_l^2$, and in $\Omega_2$,

$$\sum_{t \in \mathcal{E}_l} |\widehat{S}_0^{l-1}(P_t^*) - S_0^*(P_t^*)| \leq n_l \mathbb{E}\left[|\widehat{S}_0^{l-1}(P_t^*) - S_0^*(P_t^*)|\right] + n_l^{\frac{1}{2}}(\log n_l)^{\frac{1}{2}}.$$

Recall the definition of $P_c$ from (87). It easy to see that if $P_t^* = p$ for some $p \in \mathcal{G}$, then $P_c \in (p - \delta, p + \delta)$. Thus, we have $\mathbb{P}(P_c \in (p - \delta, p + \delta)) \geq \mathbb{P}(P_t^* = p)$. Let $P_l$ be a random

variable distributed from $\mathbb{Q}_l$ in epoch $l$. By Lemma C.7 and Portmanteau theorem, we obtain $\lim_{l\to\infty} \mathbb{P}(P_l \in (p-\delta, p+\delta)) = \mathbb{P}(P_c \in (p-\delta, p+\delta))$. Combining these results, we have $\lim_{l\to\infty} q_l(p) = \lim_{l\to\infty} \mathbb{P}(P_l = p) \geq \mathbb{P}(P_t^* = p) = q^*(p)$. Then, for sufficiently large $l$, we have

$$\mathbb{E}\left[|\widehat{S}_0^{l-1}(P_t^*) - S_0^*(P_t^*)|\right] = \sum_{p\in\mathcal{G}} |\widehat{S}_0^{l-1}(p) - S_0^*(p)|q^*(p)$$

$$= \sum_{p\in\mathcal{G}} |\widehat{S}_0^{l-1}(p) - S_0^*(p)|\frac{q^*(p)}{q_{k-1}(p)}q_{k-1}(p)$$

$$\leq C_5 \sum_{p\in\mathcal{G}} |\widehat{S}_0^{l-1}(p) - S_0^*(p)|q_{k-1}(p)$$

$$\leq C_5 \left(\sum_{p\in\mathcal{G}} |\widehat{S}_0^{l-1}(p) - S_0^*(p)|^2 q_{k-1}(p)\right)^{\frac{1}{2}}$$

$$= C_5 \|\widehat{\mathbf{S}}_0^{l-1} - \mathbf{S}_0^*\|_{2,\mathbb{Q}_{l-1}},$$

where the last inequality holds by Jensen's inequality, and $C_5$ be a positive constant. Combining the last three displays, under the event $\Omega_1 \cap \Omega_2$, we have

$$(\text{ii}) < C_6 n_l \left(\|\widehat{\mathbf{S}}_0^{l-1} - \mathbf{S}_0^*\|_{2,\mathbb{Q}_{l-1}} + \|\widehat{\beta}^{l-1} - \beta^*\|_2\right) + C_3 n_l^{\frac{1}{2}}(\log n_l)^{\frac{1}{2}}$$

$$= C_6 n_l \mathcal{D}_{\mathbb{Q}_{l-1}}\left((\widehat{\mathbf{S}}_0^{l-1}, \widehat{\beta}^{l-1}), (\mathbf{S}_0^*, \beta^*)\right) + C_3 n_l^{\frac{1}{2}}(\log n_l)^{\frac{1}{2}} \tag{69}$$

where $C_6 = C_3 C_5 \vee C_4$ be a positive constant.

Similarly, for (iii), under the event $\Omega_1$, we have

$$(\text{iii}) < C_3 \sum_{t\in\mathcal{E}_l} |\widehat{S}_0^{l-1}(P_t) - S_0^*(P_t)| + C_4 n_l \|\widehat{\beta}^{l-1} - \beta^*\|_2.$$

By Hoeffding's inequality, there exists an event $\Omega_3$ such that $\mathbb{P}(\Omega_3) \geq 1 - 1/n_l^2$, and in $\Omega_3$,

$$\sum_{t\in\mathcal{E}_l} |\widehat{S}_0^{l-1}(P_t) - S_0^*(P_t)| \leq n_l \mathbb{E}\left[|\widehat{S}_0^{l-1}(P_t) - S_0^*(P_t)|\right] + n_l^{\frac{1}{2}}(\log n_l)^{\frac{1}{2}}.$$

Note that if $P_c \in (p-\delta, p+\delta)$ for some $p \in \mathcal{G}$, then $P_t^* \in \{p-\delta, p, p+\delta\}$. By Lemma C.7 and Portmanteau theorem, we have $\lim_{l\to\infty} q_l(p) = \lim_{l\to\infty} \mathbb{P}(P_l \in (p-\delta, p+\delta)) = \mathbb{P}(P_c \in (p-\delta, p+\delta)) \leq \mathbb{P}(P_t^* = p-\delta) + \mathbb{P}(P_t^* = p) + \mathbb{P}(P_t^* = p+\delta) \lesssim \delta$, where the last inequality holds by Assumption (B2). Then, for sufficiently large $l$, we have

$$\mathbb{E}\left[|\widehat{S}_0^{l-1}(P_t) - S_0^*(P_t)|\right] = \sum_{p\in\mathcal{G}} |\widehat{S}_0^{l-1}(p) - S_0^*(p)|q_l(p)$$

$$= \sum_{p\in\mathcal{G}} |\widehat{S}_0^{l-1}(p) - S_0^*(p)|\frac{q_l(p)}{q^*(p)}\frac{q^*(p)}{q_{k-1}(p)}q_{k-1}(p)$$

$$\leq C_7 \sum_{p\in\mathcal{G}} |\widehat{S}_0^{l-1}(p) - S_0^*(p)|q_{k-1}(p)$$

$$\leq C_7 \left(\sum_{p\in\mathcal{G}} |\widehat{S}_0^{l-1}(p) - S_0^*(p)|^2 q_{k-1}(p)\right)^{\frac{1}{2}}$$

$$= C_7 \|\widehat{\mathbf{S}}_0^{l-1} - \mathbf{S}_0^*\|_{2,\mathbb{Q}_{l-1}}.$$

Combining the last three displays, under the event $\Omega_1 \cap \Omega_3$, we have

$$(\text{iii}) < C_8 n_l \mathcal{D}_{\mathbb{Q}_{l-1}}\left((\widehat{\mathbf{S}}_0^{l-1}, \widehat{\beta}^{l-1}), (\mathbf{S}_0^*, \beta^*)\right) + C_3 n_l^{\frac{1}{2}}(\log n_l)^{\frac{1}{2}}, \tag{70}$$

where $C_8 = C_3 C_7 \vee C_4$ be a positive constant.

From (65), (67), (69) and (70), for sufficiently large $l$, with probability at least $1 - (\exp(-c_2/2^{1/3} \cdot n_l^{1/3}) + 3/n_l^2)$, it holds that

$$\sum_{t \in \mathcal{E}_l} r(t) \leq C_1 \left( n_l^{\frac{1+\gamma_l}{2}} \wedge n_l^{\frac{2}{3}} \right) (\log n_l)^{\frac{1}{2}} + p_{\max} \left( C_6 n_l \mathcal{D}_{\mathbb{Q}_{l-1}} \left( (\widehat{\mathbf{S}}_0^{l-1}, \widehat{\beta}^{l-1}), (\mathbf{S}_0^*, \beta^*) \right) + C_3 n_l^{\frac{1}{2}} (\log n_l)^{\frac{1}{2}} \right)$$

$$+ p_{\max} \left( C_8 n_l \mathcal{D}_{\mathbb{Q}_{l-1}} \left( (\widehat{\mathbf{S}}_0^{l-1}, \widehat{\beta}^{l-1}), (\mathbf{S}_0^*, \beta^*) \right) + C_3 n_l^{\frac{1}{2}} (\log n_l)^{\frac{1}{2}} \right)$$

$$= C_9 n_l \mathcal{D}_{\mathbb{Q}_{l-1}} \left( (\widehat{\mathbf{S}}_0^{l-1}, \widehat{\beta}^{l-1}), (\mathbf{S}_0^*, \beta^*) \right) + C_{10} \left( n_l^{\frac{1+\gamma_l}{2}} \wedge n_l^{\frac{2}{3}} \right) (\log n_l)^{\frac{1}{2}},$$

where $C_9 = p_{\max}(C_6 + C_8)$ and $C_{10} = C_1 + 2p_{\max}C_3$ are positive constants. Then, the proof is complete.

$\square$

**Lemma B.2.** *Suppose that assumptions (A1)-(A3), (A5) and (B1)-(B2) hold. Suppose that the prior distribution $\Pi$ is specified as in (4), and the policy $\pi_l$ for each epoch $l$ is defined by (5). Then, there exist positive constants $C_1, \ldots, C_5$ depending on $L$, $B$, $p_{\min}$, $p_{\max}$, $\kappa$, $\alpha$, $\rho$, $a$, $b$, $\eta_1$, $\eta_2$ and $n_1$ such that for $l \geq C_1$,*

$$\sum_{t \in \mathcal{E}_l} r(t) \leq \begin{cases} C_2 d^{\frac{1}{2}} n_l^{\frac{1}{2}} (\log(d \vee n_l))^{\frac{1}{2}} + C_3 n_l^{\frac{1+\gamma_l}{2}} (\log n_l)^{\frac{1}{2}} & \text{if } \gamma_{l-1} < \frac{1}{3} \\ C_2 d^{\frac{1}{2}} n_l^{\frac{1}{2}} (\log(d \vee n_l))^{\frac{1}{2}} + C_3 n_l^{\frac{2}{3}} (\log n_l)^{\frac{1}{2}} & \text{if } \gamma_{l-1} \geq \frac{1}{3}, \end{cases}$$

*with probability at least $1 - \zeta_l$, where*

$$\zeta_l = \begin{cases} C_4/n_{l-1}^{\gamma_{l-1}} & \text{if } \gamma_{l-1} < \frac{1}{3} \\ C_5/n_{l-1}^{\frac{1}{3}} & \text{if } \gamma_{l-1} \geq \frac{1}{3}. \end{cases}$$

*Proof.* By Lemma B.1, there exist positive constants $c_1$, $c_2$, $c_3$ and $c_4$ depending on $L$, $B$, $p_{\min}$, $p_{\max}$, $\kappa$, $\alpha$, $\rho$, $a$, $b$, $\eta_1$, $\eta_2$ and $n_1$ such that for $l \geq c_1$,

$$\sum_{t \in \mathcal{E}_l} r(t) \leq c_2 n_l \mathcal{D}_{\mathbb{Q}_{l-1}}(\widehat{\theta}^{l-1}, \theta^*) + c_3 \left( n_l^{\frac{1+\gamma_l}{2}} \wedge n_l^{\frac{2}{3}} \right) (\log n_l)^{\frac{1}{2}} \tag{71}$$

with probability at least $1 - \exp(-c_4 n_l^{1/3}) - 3/n_l^2$. In addition, by Lemma 5.1, there exist positive constants $c_5$, $c_6$, $c_7$ and $c_8$ depending on $L$, $B$, $p_{\min}$, $p_{\max}$, $\kappa$, $\alpha$, $\rho$, $a$, $b$ and $n_1$ such that for $l \geq c_5$,

$$\mathcal{D}_{\mathbb{Q}_{l-1}}(\widehat{\theta}^{l-1}, \theta^*) \leq c_6 \epsilon_{l-1} \tag{72}$$

with probability at least $1 - \zeta_{l-1} - 1/(n_{l-1} \epsilon_{l-1}^2)$, where

$$\epsilon_l = \begin{cases} \sqrt{\frac{d}{n_l}} \sqrt{\log(d \vee n_l)} + n_l^{-\frac{1-\gamma_l}{2}} \sqrt{\log n_l} & \text{if } \gamma_l < \frac{1}{3}, \\ \sqrt{\frac{d}{n_l}} \sqrt{\log(d \vee n_l)} + \left( \frac{\log n_l}{n_l} \right)^{\frac{1}{3}} & \text{if } \gamma_l \geq \frac{1}{3}. \end{cases}$$

and

$$\zeta_l = \begin{cases} \exp(-c_7 n_l \epsilon_l^2) & \text{if } \gamma_l < \frac{1}{3}, \\ \exp(-c_8 n_l^{\frac{1}{3}}) & \text{if } \gamma_l \geq \frac{1}{3}. \end{cases}$$

Consider the epoch $l-1$ satisfying $\gamma_{l-1} < 1/3$. On the event both (71) and (72) hold, for epoch $l \geq c_1 \vee c_5$, we have

$$\sum_{t \in \mathcal{E}_l} r(t) \leq c_2 c_6 n_l \epsilon_{l-1} + c_3 \left( n_l^{\frac{1+\gamma_l}{2}} \wedge n_l^{\frac{2}{3}} \right) (\log n_l)^{\frac{1}{2}}$$

$$\leq 2 c_2 c_6 \left( \sqrt{n_{l-1} d} \sqrt{\log(d \vee n_{l-1})} + n_{l-1}^{\frac{1+\gamma_{l-1}}{2}} \sqrt{\log n_{l-1}} \right) + c_3 \left( n_l^{\frac{1+\gamma_l}{2}} \wedge n_l^{\frac{2}{3}} \right) (\log n_l)^{\frac{1}{2}}$$

$$\leq C_1 d^{\frac{1}{2}} n_l^{\frac{1}{2}} (\log(d \vee n_l))^{\frac{1}{2}} + C_2 n_l^{\frac{1+\gamma_l}{2}} (\log n_l)^{\frac{1}{2}},$$

where the second inequality holds by substituting $n_l = 2n_{l-1}$ and $\epsilon_{l-1}$, and the last inequality holds with positive constants $C_1 = 2c_2c_6$ and $C_2 = 2c_2c_6c_p + c_3$ because it holds that $n_{l-1}^{\gamma_{l-1}} \asymp n_l^{\gamma_l}$ due to (7), where $c_p$ is a positive constant depending on $p_{\min}$, $p_{\max}$ and $\kappa$. Similarly, for epoch $l \geq c_1 \vee c_5$ satisfying $\gamma_{l-1} \geq 1/3$, we have

$$\sum_{t \in \mathcal{E}_l} r(t) \leq c_2c_6n_l\epsilon_{l-1} + c_3 \left( n_l^{\frac{1+\gamma_l}{2}} \wedge n_l^{\frac{2}{3}} \right) (\log n_l)^{\frac{1}{2}}$$

$$\leq 2c_2c_6 \left( \sqrt{n_{l-1}d}\sqrt{\log(d \vee n_{l-1})} + n_{l-1}^{\frac{2}{3}}(\log n_{l-1})^{\frac{1}{3}} \right) + c_3 \left( n_l^{\frac{1+\gamma_l}{2}} \wedge n_l^{\frac{2}{3}} \right) (\log n_l)^{\frac{1}{2}}$$

$$\leq C_1 d^{\frac{1}{2}} n_l^{\frac{1}{2}} (\log(d \vee n_l))^{\frac{1}{2}} + C_2 n_l^{\frac{2}{3}} (\log n_l)^{\frac{1}{2}}.$$

Let $\Omega_1$ and $\Omega_2$ denote the events where inequalities (71) and (72) hold, respectively. Then, for epoch $l-1$ satisfying $\gamma_{l-1} < 1/3$, we obtain

$$\mathbb{P}(\Omega_1^c \cup \Omega_2^c) \leq \exp(-c_4 n_l^{1/3}) + 3/n_l^2 + \exp(-c_7 n_{l-1}\epsilon_{l-1}^2) + 1/(n_{l-1}\epsilon_{l-1}^2)$$

$$\leq 1/(2^{1/3}c_4 n_{l-1}^{1/3}) + 3/(4n_{l-1}^2) + 1/(c_7 n_{l-1}^{\gamma_{l-1}}) + 1/n_{l-1}^{\gamma_{l-1}}$$

$$\leq C_3/n_{l-1}^{\gamma_{l-1}},$$

where the second inequality holds because $\exp(-x) \leq 1/x$ for any $x > 0$ and $n_l\epsilon_l^2 \geq n_l^{\gamma_l}$, and the last inequality holds since $\gamma_{l-1} < 1/3$, and $C_3$ is a positive constant defined as $C_3 = 1/(2^{1/3}c_4) + 1/c_7 + 7/4$. Similarly, for epoch $l-1$ satisfying $\gamma_{l-1} \geq 1/3$, we have

$$\mathbb{P}(\Omega_1^c \cup \Omega_2^c) \leq \exp(-c_4 n_l^{1/3}) + 3/n_l^2 + \exp(-c_8 n_{l-1}^{1/3}) + 1/(n_{l-1}\epsilon_{l-1}^2)$$

$$\leq 1/(2^{1/3}c_4 n_{l-1}^{1/3}) + 3/(4n_{l-1}^2) + 1/(c_8 n_{l-1}^{1/3}) + 1/n_{l-1}^{1/3}$$

$$\leq C_4/n_{l-1}^{1/3},$$

where the second inequality holds because $n_l\epsilon_l^2 \geq n_l^{1/3}$, and the last inequality holds by a positive constant $C_4 = 1/(2^{1/3}c_4) + 1/c_8 + 7/4$. Then, the proof is complete. $\qquad \square$

*Proof of Theorem 5.2.* Before proceeding, we may without loss of generality assume that the last epoch is complete (i.e., $T = n_1(2^N - 1)$ for some integer $N \geq 1$). If not (i.e., $n_1(2^{N-1} - 1) < T < n_1(2^N - 1)$), the regret associated with the incomplete last epoch will be no greater than if it were completed. Thus, the number of epochs $N$ and $T$ satisfies $T = n_1(2^N - 1)$, equivalently $N = \log_2(T/n_1 + 1)$.

We first consider the case where $\gamma < 1/3$. We define $N_{\gamma_0} := \lfloor \log_2(T^{\gamma_0}/n_1) \rfloor + 2$, where $\gamma_0 \in (0, 1)$ is a constant to be chosen later. Note that $N_{\gamma_0} < N$ for a sufficiently large $T \geq 2^{2/(1-\gamma_0)}$. For epoch $l \leq N_{\gamma_0}$, we have

$$l \leq \lfloor \log_2(T^{\gamma_0}/n_1) \rfloor + 2 \leq \log_2(T^{\gamma_0}/n_1) + 2,$$

and hence $n_{l-1} = n_1 2^{l-2} \leq T^{\gamma_0}$. Therefore, by the equation (7), we have

$$K_{k-1} = \left\lfloor \frac{p_{\max} - p_{\min}}{\kappa}T^\gamma \right\rfloor \geq \left\lfloor \frac{p_{\max} - p_{\min}}{\kappa}n_{l-1}^{\gamma/\gamma_0} \right\rfloor,$$

for all $l \leq N_{\gamma_0}$. If we set $\gamma_0 = 3\gamma$, then we have

$$\left\lfloor \frac{p_{\max} - p_{\min}}{\kappa}n_{l-1}^{\gamma_{l-1}} \right\rfloor \geq \left\lfloor \frac{p_{\max} - p_{\min}}{\kappa}n_{l-1}^{1/3} \right\rfloor.$$

Then, the condition $\gamma_{l-1} \geq 1/3$ is sufficient to hold the preceding inequality. On the other hand, for epoch $l > N_{\gamma_0}$, since $l$ is an integer value, we have $l \geq \lfloor \log_2(T^{\gamma_0}/n_1) \rfloor + 3 > \log_2(T^{\gamma_0}/n_1) + 2$, and hence $n_{l-1} = n_1 2^{l-2} > T^{\gamma_0}$. Therefore, by the equation (7) and setting $\gamma_0 = 3\gamma$, we have

$$\left\lfloor \frac{p_{\max} - p_{\min}}{\kappa}n_{l-1}^{\gamma_{l-1}} \right\rfloor < \left\lfloor \frac{p_{\max} - p_{\min}}{\kappa}n_{l-1}^{1/3} \right\rfloor$$

for all $l > N_{\gamma_0}$, and the condition $\gamma_{l-1} < 1/3$ is sufficient to hold this inequality. By Lemma B.2, there exit positive constants $c_1, \ldots, c_5$ depending on $L$, $B$, $p_{\min}$, $p_{\max}$, $\kappa$, $\alpha$, $\rho$, $a$, $b$, $\eta_1$, $\eta_2$ and $n_1$ such that for $l \geq c_1$,

$$\sum_{t \in \mathcal{E}_l} r(t) \leq \begin{cases} c_2 d^{\frac{1}{2}} n_l^{\frac{1}{2}} (\log(d \vee n_l))^{\frac{1}{2}} + c_3 n_l^{\frac{1+\gamma_l}{2}} (\log n_l)^{\frac{1}{2}} & \text{if } l > N_{3\gamma} \\ c_2 d^{\frac{1}{2}} n_l^{\frac{1}{2}} (\log(d \vee n_l))^{\frac{1}{2}} + c_3 n_l^{\frac{2}{3}} (\log n_l)^{\frac{1}{2}} & \text{if } l \leq N_{3\gamma}, \end{cases} \tag{73}$$

with probability at least $1 - \zeta_l$, where

$$\zeta_l = \begin{cases} c_4 / n_{l-1}^{\gamma_{l-1}} & \text{if } l > N_{3\gamma} \\ c_5 / n_{l-1}^{\frac{1}{3}} & \text{if } l \leq N_{3\gamma}. \end{cases}$$

For unity of notation, we denote $N_0 := \lceil c_1 \rceil - 1$. Note that $N_0 < N_{3\gamma}$ for sufficiently large $T \geq (2^{N_0+1} n_1)^{1/(3\gamma)}$. Let $\Omega_{1,l}$ and $\Omega_{2,l}$ denote the events where the first and second inequalities in (73) are satisfied for each epoch $l$, respectively. Then, we have

$$\mathbb{P}\left[ \left\{ \bigcap_{l=N_0+1}^{N_{3\gamma}} \Omega_{1,l} \right\} \cap \left\{ \bigcap_{l=N_{3\gamma}+1}^{N} \Omega_{2,l} \right\} \right] \geq 1 - \sum_{l=N_0+1}^{N_{3\gamma}} c_5 n_{l-1}^{-\frac{1}{3}} - \sum_{l=N_{3\gamma}+1}^{N} c_4 n_{l-1}^{-\gamma_{l-1}}$$

$$> 1 - c_4 \vee c_5 \cdot \sum_{l=N_0+1}^{N} n_{l-1}^{-\gamma_{l-1}}$$

$$\geq 1 - (c_4 \vee c_5) c_p^{-1} \cdot \log(T/n_1 + 1) T^{-\gamma}, \tag{74}$$

where the second inequality holds because $\gamma_{l-1} < 1/3$ for $l > N_{3\gamma}$, and the last inequality follows from $n_{l-1}^{\gamma_{l-1}} \geq c_p T^\gamma$ by (7). Here, $c_p$ is a positive constant depending on $p_{\min}$, $p_{\max}$ and $\kappa$.

Now, we decompose the cumulative regret as

$$R(T) = \underbrace{\sum_{l=1}^{N_0} \sum_{t \in \mathcal{E}_l} r(t)}_{(i)} + \underbrace{\sum_{l=N_0+1}^{N_{3\gamma}} \sum_{t \in \mathcal{E}_l} r(t)}_{(ii)} + \underbrace{\sum_{l=N_{3\gamma}+1}^{N} \sum_{t \in \mathcal{E}_l} r(t)}_{(iii)}.$$

For (i), note that $p S_0^*(p)^{\exp(\mathbf{x}^\top \beta^*)}$ is upper bounded by a positive constant $C_1 := \max\{p S_0^*(p)^{\exp(v)} : p \in [p_{\min}, p_{\max}], v \in [-BL, BL]\}$ depending on $p_{\min}$, $p_{\max}$, $B$ and $L$. Then, we have

$$(i) \leq \sum_{l=1}^{N_0} \sum_{t \in \mathcal{E}_l} C_1 = C_2,$$

where $C_2 = n_1(2^{N_0} - 1)C_1$ be a constant and does not depend on $T$. Let $\Omega$ be the event in the probability notation in the display (74). For (ii), under the event $\Omega$, by (73), we have

$$(ii) \leq \sum_{l=N_0+1}^{N_{3\gamma}} \left\{ c_2 d^{\frac{1}{2}} n_l^{\frac{1}{2}} (\log(d \vee n_l))^{\frac{1}{2}} + c_3 n_l^{\frac{2}{3}} (\log n_l)^{\frac{1}{2}} \right\}$$

$$\leq C_3 \cdot d^{\frac{1}{2}} T^{\frac{3}{2}\gamma} (\log(d \vee T))^{\frac{1}{2}} + C_4 \cdot T^{2\gamma} (\log T)^{\frac{1}{2}},$$

where $C_3 = 2^{\frac{5}{2}} c_2$ and $C_4 = 2^3 c_3$ are positive constants. Similarly, for (iii), under the event $\Omega$, we obtain

$$(iii) \leq \sum_{l=N_{3\gamma}+1}^{N} \left\{ c_2 d^{\frac{1}{2}} n_l^{\frac{1}{2}} (\log(d \vee n_l))^{\frac{1}{2}} + c_3 n_l^{\frac{1+\gamma_l}{2}} (\log n_l)^{\frac{1}{2}} \right\}$$

$$\leq C_5 \cdot d^{\frac{1}{2}} T^{\frac{1}{2}} (\log(d \vee T))^{\frac{1}{2}} + c_3 \sum_{l=1}^{N} n_l^{\frac{\gamma_l}{2}} n_l^{\frac{1}{2}} (\log n_l)^{\frac{1}{2}}$$

$$\leq C_5 \cdot d^{\frac{1}{2}} T^{\frac{1}{2}} (\log(d \vee T))^{\frac{1}{2}} + c_3 C_p^{\frac{1}{2}} \cdot T^{\frac{\gamma}{2}} \sum_{l=1}^{N} n_l^{\frac{1}{2}} (\log n_l)^{\frac{1}{2}}$$

$$\leq C_5 \cdot d^{\frac{1}{2}} T^{\frac{1}{2}} (\log(d \vee T))^{\frac{1}{2}} + C_6 \cdot T^{\frac{\gamma+1}{2}} (\log T)^{\frac{1}{2}},$$

where the first inequality follows from (73), the second inequality holds for a positive constant $C_5 = 2^2 c_2$, the third inequality holds because $n_l^{\gamma_l} \leq C_p T^\gamma$ for any $l = 1, \ldots, N$ by (7) with a positive constant $C_p$ depending on $p_{\min}$, $p_{\max}$ and $\kappa$, and the last inequality holds for a positive constant $C_6 = 2^2 c_3 C_p^{\frac{1}{2}}$. Combining the last five displays, for sufficiently large $T \geq C_7$, it holds that

$$R(T) \leq C_2 + (C_3 + C_5) \cdot d^{\frac{1}{2}} T^{\frac{1}{2}} (\log(d \vee T))^{\frac{1}{2}} + C_4 \cdot T^{2\gamma} (\log T)^{\frac{1}{2}} + C_6 \cdot T^{\frac{\gamma+1}{2}} (\log T)^{\frac{1}{2}}$$
$$\leq C_8 d^{\frac{1}{2}} T^{\frac{1}{2}} (\log(d \vee T))^{\frac{1}{2}} + C_9 T^{\frac{\gamma+1}{2}} (\log T)^{\frac{1}{2}},$$

with probability at least $1 - C_{10} \log(T/n_1 + 1)/T^\gamma$, where $C_7 = (2^{2/(1-3\gamma)}) \vee ((2^{N_0+1} n_1)^{1/(3\gamma)})$, $C_8 = C_2 + C_3 + C_5$, $C_9 = C_4 + C_6$ and $C_{10} = (c_4 \vee c_5) c_p^{-1}$ are positive constants, and the last inequality holds because $T^{2\gamma} \leq T^{\frac{\gamma+1}{2}}$ for $\gamma < 1/3$.

Next, we consider the case where $\gamma \geq 1/3$. For any epoch $l \leq N$, we have $2^{l-2} < 2^l - 1 \leq T/n_1$, and hence $n_{l-1} = n_1 2^{l-2} < T$. Therefore, by the equation (7), we have

$$\left\lfloor \frac{p_{\max} - p_{\min}}{\kappa} n_{l-1}^{\gamma_{l-1}} \right\rfloor = \left\lfloor \frac{p_{\max} - p_{\min}}{\kappa} T^\gamma \right\rfloor \geq \left\lfloor \frac{p_{\max} - p_{\min}}{\kappa} n_{l-1}^\gamma \right\rfloor.$$

Then, the condition $\gamma_{l-1} \geq \gamma \geq 1/3$ is sufficient to hold the last display. By Lemma B.2, the event $\Omega_{2,l}$ in (73) holds for all $l > N_0$, and we have

$$\mathbb{P}(\Omega_{2,l}) \geq 1 - c_5/n_{l-1}^{\frac{1}{3}} \quad \text{for } l > N_0.$$

We define $N_1 := \lfloor \log_2(T^{2/3}/n_1) \rfloor + 1$. Note that $N_0 < N_1$ for sufficiently large $T > (2^{N_0} n_1)^{3/2}$. Then, by the preceding display, we obtain

$$\mathbb{P}\left[\left\{\bigcap_{l=N_1+1}^N \Omega_{2,l}\right\}\right] \geq 1 - \sum_{l=N_1+1}^N c_5 n_{l-1}^{-\frac{1}{3}}$$
$$> 1 - 2^{\frac{1}{3}} c_5 \sum_{l=N_1+1}^N T^{-\frac{2}{9}}$$
$$\geq 1 - 2^{\frac{1}{3}} c_5 \cdot \log(T/n_1 + 1) T^{-\frac{2}{9}},$$

where the second inequality holds because $n_{l-1} > 2^{-1} T^{2/3}$ for $l \geq N_1 + 1$. Let $\Omega'$ be the event in the probability notation in the preceding display.

Now, we decompose the cumulative regret as

$$R(T) = \underbrace{\sum_{l=1}^{N_1} \sum_{t \in \mathcal{E}_l} r(t)}_{\text{(I)}} + \underbrace{\sum_{l=N_1+1}^N \sum_{t \in \mathcal{E}_l} r(t)}_{\text{(II)}}.$$

For (I), we have

$$\text{(I)} \leq \sum_{l=1}^{N_1} \sum_{t \in \mathcal{E}_l} C_1 \leq C_1 \sum_{l=1}^{N_1} n_l \leq 2C_1 \cdot T^{\frac{2}{3}}.$$

For (II), under the event $\Omega'$, we obtain

$$\text{(II)} \leq \sum_{l=N_1+1}^N \left\{ c_2 d^{\frac{1}{2}} n_l^{\frac{1}{2}} (\log(d \vee n_l))^{\frac{1}{2}} + c_3 n_l^{\frac{2}{3}} (\log n_l)^{\frac{1}{2}} \right\}$$
$$\leq C_{11} \cdot d^{\frac{1}{2}} T^{\frac{1}{2}} (\log(d \vee T))^{\frac{1}{2}} + C_{12} \cdot T^{\frac{2}{3}} (\log T)^{\frac{1}{2}},$$

where $C_{11} = 2^2 c_2$ and $C_{12} = 2^{\frac{7}{3}} c_3$ are positive constants. Combining the last four displays, for sufficiently large $T \geq C_{13}$, it holds that

$$R(T) \leq C_{11} d^{\frac{1}{2}} T^{\frac{1}{2}} (\log(d \vee T))^{\frac{1}{2}} + C_{14} T^{\frac{2}{3}} (\log T)^{\frac{1}{2}},$$

with probability at least $1 - C_{15} \log(T/n_1 + 1)/T^{2/9}$, where $C_{13} = (2^{N_0} n_1)^{3/2} + 1$, $C_{14} = 2C_1 + C_{12}$ and $C_{15} = 2^{1/3} c_5$ are positive constants. This completes the proof. $\qquad\square$

## B.3 Proof of Theorem 5.3

*Proof.* Recall the grid support $\mathcal{G} = \{g_i : i = 1, \ldots, K\}$, where $g_i = p_{\min} + i\delta$, $\delta = \kappa T^{-\gamma}$ for some $\kappa > 0$, and $K = \lfloor (p_{\max} - p_{\min})/\delta \rfloor$. Let $p_t$, $r_t$ and $y_t$ be the offered price, the revenue and the feedback at time $t$, respectively. Let $h_t = (p_1, r_1, p_2, r_2, \ldots, p_t, r_t)$ be a history over $t$ times. Note that $h_t$ can be induced by $(p_1, y_1, \ldots, p_t, y_t)$ since $r_t = p_t y_t$. Define a policy $\pi = (\pi_t)_{t=1}^T$, where $\pi_t$ is a conditional distribution of price $p_t$ given $h_{t-1}$ supported on $\mathcal{G}$. We denote the conditional distribution of revenue $r_t$ given $p_t$ with respect to the complementary c.d.f. $S(p) = 1 - F(p)$ by $P_{p_t}^S$. With abuse of notation we view $\pi_t : \mathcal{G} \to [0,1]$ and $P_{p_t}^S : \{0, p_t\} \to [0,1]$ as probability mass function. In addition, we abuse notation by writing $P_i^S = P_{p_t}^S$ if $p_t = g_i$ for some $g_i \in \mathcal{G}$. For given $S(\cdot)$, note that $r_t = p_t y_t$ where $y_t \sim \text{Bin}(1, S(p_t))$. Then, we have

$$P_{p_t}^S(r_t) = S(p_t)^{\frac{r_t}{p_t}} (1 - S(p_t))^{1 - \frac{r_t}{p_t}}$$
$$= S(p_t)^{y_t} (1 - S(p_t))^{1 - y_t}. \tag{75}$$

For given $S$, let $v_S = (P_1^S, P_2^S, \ldots, P_K^S)$ be the reward distributions associated with a $K$-armed bandit. For given policy $\pi$ and bandit $v_S$, we denote the joint distribution of $(p_1, r_1, \ldots, p_T, r_T)$ by $P_{v_S \pi}$. Then, the probability of obtaining a fixed configuration $(p_1, r_1, \ldots, p_T, r_T)$ is given by

$$P_{v_S \pi}(p_1, r_1, \ldots, p_T, r_T) = \prod_{t=1}^T \pi_t(p_t | h_{t-1}) P_{p_t}^S(r_t). \tag{76}$$

Based on this, we define the expected regret by

$$R(T, S) := \mathbb{E}_{v_S \pi} \left[ \sum_{t=1}^T r(p^*, S) - r(p_t, S) \right],$$

where $r(p, S) := pS(p)$ be the expected revenue with respect to $S$ for $p \in \mathcal{G}$, $p^* = \text{argmax}_{p \in \mathcal{G}} \{pS(p)\}$ be the optimal price and $\mathbb{E}_{v_S \pi}$ denotes the expectation under $P_{v_S \pi}$. Further, we define the suboptimality gap of index $i$ by $\Delta_i^S := r(p^*, S) - r(g_i, S)$ for $i = 1, \ldots, K$. For the simplicity of notation we use $P_S$ and $\mathbb{E}_S$ in place of $P_{v_S \pi}$ and $\mathbb{E}_{v_S \pi}$, respectively, for a fixed policy $\pi$.

Now, we first construct two bandits $v_{S_1}$ and $v_{S_1'}$ for $S_1, S_1' \in \mathcal{S} := \{S : \mathcal{G} \to [0,1] \mid 1 > M_2 > S(g_1) \geq \cdots \geq S(g_K) > M_1 > 0 \text{ for some } 0 < M_1 < M_2 < 1\}$ in the following description. Fix a policy $\pi$ and suppose that $\gamma < 1/3$. Let $\epsilon > 0$ be some constant to be chosen later. We define a bandit $v_{S_1} = (P_1^{S_1}, \ldots, P_K^{S_1})$ such that for some $j_1 \in [K]$,

$$\begin{cases} S_1(g_i) = (c + \epsilon) \cdot g_i^{-1} & \text{if } i = j_1 \\ S_1(g_i) = c \cdot g_i^{-1} & \text{otherwise,} \end{cases} \tag{77}$$

where $c > 0$ be a constant so that $S_1 \in \mathcal{S}$. Note that $c$ only depends on $M_1, M_2, p_{\min}$ and $p_{\max}$. For $i = 1, \ldots, K$, let $N_i(t) := \sum_{s=1}^t \mathbb{1}\{p_s = g_i\}$ be the number of times price $g_i$ was chosen by the policy over $t$ times, and $j_1' = \text{argmin}_{i \neq j_1} \mathbb{E}_{S_1}[N_i(T)]$. Since $\sum_{i=1}^K \mathbb{E}_{S_1}[N_i(T)] = T$, it holds that $\mathbb{E}_{S_1}[N_{j_1'}(T)] \leq \frac{T}{K-1}$.

The second bandit $v_{S_1'} = (P_1^{S_1'}, \ldots, P_K^{S_1'})$ is defined by

$$\begin{cases} S_1'(g_i) = (c + \epsilon) \cdot g_i^{-1} & \text{if } i = j_1 \\ S_1'(g_i) = (c + 2\epsilon) \cdot g_i^{-1} & \text{if } i = j_1' \\ S_1'(g_i) = c \cdot g_i^{-1} & \text{otherwise.} \end{cases} \tag{78}$$

Therefore, $r(g_i, S_1) = r(g_i, S_1')$ except at index $j_1'$ and the optimal price in $v_{S_1}$ is $g_{j_1}$, while in $v_{S_1'}$, $g_{j_1'}$ is the optimal price. Then, we have

$$
\begin{aligned}
R(T, S_1) &= \mathbb{E}_{S_1} \left[ \sum_{t=1}^{T} r(g_{j_1}, S_1) - r(p_t, S_1) \right] \\
&= \sum_{i=1}^{K} \Delta_i^{S_1} \mathbb{E}_{S_1} [N_i(T)] \\
&= \sum_{i \in [K], i \neq j_1} \epsilon \cdot \mathbb{E}_{S_1} [N_i(T)] \\
&= \epsilon \left( T - \mathbb{E}_{S_1} [N_{j_1}(T)] \right) \\
&\geq \frac{T\epsilon}{2} \cdot P_{S_1} \left( N_{j_1}(T) \leq \frac{T}{2} \right),
\end{aligned}
$$

where the second equality holds by the regret decomposition and the third equality holds because $\Delta_i^{S_1} = (c + \epsilon) - c = \epsilon$ for $i \neq j_1$. Similarly, we have

$$
\begin{aligned}
R(T, S_1') &= \sum_{i=1}^{K} \Delta_i^{S_1'} \mathbb{E}_{S_1'} [N_i(T)] \\
&> \epsilon \cdot \mathbb{E}_{S_1'} [N_{j_1}(T)] \\
&> \frac{T\epsilon}{2} \cdot P_{S_1'} \left( N_{j_1}(T) > \frac{T}{2} \right).
\end{aligned}
$$

Combining the last two displays and Lemma 2.6 in [42], we have

$$
\begin{aligned}
R(T, S_1) + R(T, S_1') &> \frac{T\epsilon}{2} \left( P_{S_1} \left( N_{j_1}(T) \leq \frac{T}{2} \right) + P_{S_1'} \left( N_{j_1}(T) > \frac{T}{2} \right) \right) \\
&\geq \frac{T\epsilon}{4} \exp \left( -K(P_{S_1}, P_{S_1'}) \right).
\end{aligned}
$$

By Lemma 1 in [14], the KL divergence $K(P_{S_1}, P_{S_1'})$ is bounded by

$$
\begin{aligned}
K(P_{S_1}, P_{S_1'}) &= \sum_{i=1}^{K} \mathbb{E}_{S_1} [N_i(T)] K(P_i^{S_1}, P_i^{S_1'}) \\
&= \mathbb{E}_{S_1} [N_{j_1'}(T)] K(P_{j_1'}^{S_1}, P_{j_1'}^{S_1'}) \\
&\leq \frac{T}{K-1} K(P_{j_1'}^{S_1}, P_{j_1'}^{S_1'}),
\end{aligned}
$$

where the first equality holds by (76), the second equality holds by the definition of $S_1$ and $S_1'$, and the first inequality holds by the definition of index $j_1'$. Note that (75) implies that $P_{j_1'}^{S_1}$ and $P_{j_1'}^{S_1'}$ are distributions of Bernoulli random variables with parameters $S_1(g_{j_1'})$ and $S_1'(g_{j_1'})$, respectively. Therefore, by Corollary 3.1 in [40], we have

$$
\begin{aligned}
K(P_{j_1'}^{S_1}, P_{j_1'}^{S_1'}) &\leq \frac{(S_1(g_{j_1'}) - S_1'(g_{j_1'}))^2}{S_1'(g_{j_1'})(1 - S_1'(g_{j_1'}))} \\
&< \frac{(2\epsilon g_{j_1'}^{-1})^2}{M_1 M_2} \\
&\leq \frac{4}{p_{\min}^2 M_1 M_2} \epsilon^2,
\end{aligned}
$$

where the second inequality holds by the definition of $S_1$ and $S_1'$, and the last inequality holds because $g_i \in [p_{\min}, p_{\max}]$ for any $i \in [K]$.

Now, it remains to choose $\epsilon$. Due to the monotonicity of the distribution functions $S_1$ and $S_1'$, there are additional constraints in choosing $\epsilon$. Specifically, by the definition of $S_1$ (77), the following

condition must hold: $(c + \epsilon) \cdot g_{j_1}^{-1} \leq c \cdot g_{j_1-1}^{-1}$. By the direct calculations, we obtain $\epsilon \leq \frac{c}{g_{j_1-1}}\delta$. Since $g_i \in [p_{\min}, p_{\max}]$ for any $i \in [K]$, it is sufficient to choose $\epsilon \leq \frac{c}{p_{\max}}\delta$ to satisfy this condition. Similarly, we consider the monotonicity of $S_1'$, but before doing so, we divide it into two cases: (a) $j_1 < j_1'$ and (b) $j_1 > j_1'$. For the case (a), it is necessary that $S_1'(g_{j_1'}) \leq S_1'(g_{j_1}) \leq S_1'(g_{j_1-1})$ holds, and for the case (b), $S_1'(g_{j_1'}) \leq S_1'(g_{j_1'-1})$ must hold. By the simple calculations, $\epsilon \leq \frac{c}{2p_{\max}}\delta$ is sufficient to satisfy the above conditions. Since $\sqrt{K/T} \ll \delta$ if $\gamma < 1/3$, it is sufficient to choose $\epsilon = C\sqrt{K/T}$ for a small enough constant $C > 0$. Combining this with the three preceding displays, there exists a constant $C_1 > 0$ such that

$$R(T, S_1) + R(T, S_1') \geq C_1\sqrt{KT}$$
$$\gtrsim T^{\frac{1+\gamma}{2}}.$$

It completes the proof for the case $\gamma < 1/3$.

In the second case that $\gamma \geq 1/3$, we construct another pair of bandits $v_{S_2}$ and $v_{S_2'}$ for $S_2, S_2' \in \mathcal{S}$ in the following description. Let $\epsilon_2 = \kappa T^{-\frac{1}{3}}$ and $i_1, \ldots, i_J$ be positive integers such that $p_{\min} + j\epsilon_2 - \delta < g_{i_j} \leq p_{\min} + j\epsilon_2$ for $j = 1, \ldots, J$, where $J = \lfloor(p_{\max} - p_{\min})/\epsilon_2\rfloor$. We define partitions $I_j$ of index set $[K]$ by $I_j = \{i \in [K] : i_{j-1} < i \leq i_j\}$ for $j = 1, \ldots, J$, where $i_0 = 0$ with $g_0 = p_{\min}$. Then, we define a bandit $v_{S_2} = (P_1^{S_2}, \ldots, P_K^{S_2})$ such that for some $j_2 \in [J]$,

$$\begin{cases} S_2(g_i) = (c + \epsilon_2) \cdot g_{i_{j_2}}^{-1} & \text{if } i \in I_{j_2} \\ S_2(g_i) = c \cdot g_{i_j}^{-1} & \text{if } i \in I_j \text{ for } j \in [J] \text{ except at } j_2. \end{cases} \tag{79}$$

Let $M_j(t) := \sum_{i \in I_j} N_i(t)$ for $j = 1, \ldots, J$, and $j_2' = \arg\min_{j \neq j_2} \mathbb{E}_{S_2}[M_j(T)]$. Since $\sum_{j=1}^{J} \mathbb{E}_{S_2}[M_j(T)] = T$, it holds that $\mathbb{E}_{S_2}[M_{j_2'}(T)] \leq \frac{T}{J-1}$. Then, the second bandit $v_{S_2'} = (P_1^{S_2'}, \ldots, P_K^{S_2'})$ is defined by

$$\begin{cases} S_2'(g_i) = (c + \epsilon_2) \cdot g_{i_{j_2}}^{-1} & \text{if } i \in I_{j_2} \\ S_2'(g_i) = (c + 2\epsilon_2) \cdot g_{i_{j_2'}}^{-1} & \text{if } i \in I_{j_2'} \\ S_2'(g_i) = c \cdot g_{i_j}^{-1} & \text{if } i \in I_j \text{ for } j \in [J] \text{ except at } j_2 \text{ and } j_2'. \end{cases} \tag{80}$$

Therefore, $r(g_i, S_2) = r(g_i, S_2')$ except at index $i \in I_{j_2'}$ and the optimal price in $v_{S_2}$ is $g_{i_{j_2}}$, while in $v_{S_2'}$, $g_{i_{j_2'}}$ is the optimal price. For $j = 1, \ldots, J$ except at $j_2$, note that $\Delta_i^{S_2} \geq \epsilon_2$ for $i \in I_j$ since $r(g_{i_{j_2}}, S_2) = c + \epsilon_2$ and $r(g_i, S_2) \leq c$ by the definition of $S_2$. Then, we have

$$R(T, S_2) = \sum_{i=1}^{K} \Delta_i^{S_2} \mathbb{E}_{S_2}[N_i(T)]$$
$$= \sum_{j \in [J], j \neq j_2} \sum_{i \in I_j} \Delta_i^{S_2} \mathbb{E}_{S_2}[N_i(T)]$$
$$\geq \sum_{j \in [J], j \neq j_2} \epsilon_2 \cdot \mathbb{E}_{S_2}[M_j(T)]$$
$$= \epsilon_2(T - \mathbb{E}_{S_2}[M_{j_2}(T)])$$
$$\geq \frac{T\epsilon}{2} \cdot P_{S_2}\left(M_{j_2}(T) \leq \frac{T}{2}\right),$$

where the first inequality holds by the definition of $M_j(T)$. Similarly, we have

$$R(T, S_2') = \sum_{j \in [J], j \neq j_2'} \sum_{i \in I_j} \Delta_i^{S_2'} \mathbb{E}_{S_2'}[N_i(T)]$$
$$> \epsilon_2 \cdot \mathbb{E}_{S_2'}[M_{j_2}(T)]$$
$$> \frac{T\epsilon_2}{2} \cdot P_{S_2'}\left(M_{j_2}(T) > \frac{T}{2}\right).$$

Combining the two preceding displays and Lemma 2.6 in [42], we have

$$R(T, S_2) + R(T, S_2') > \frac{T\epsilon_2}{2}\left(P_{S_2}\left(M_{j_2}(T) \le \frac{T}{2}\right) + P_{S_2'}\left(M_{j_2}(T) > \frac{T}{2}\right)\right)$$

$$\ge \frac{T\epsilon_2}{4}\exp\left(-K(P_{S_2}, P_{S_2'})\right). \tag{81}$$

By Lemma 1 in [14], we can decompose the KL divergence $K(P_{S_2}, P_{S_2'})$ as

$$K(P_{S_2}, P_{S_2'}) = \sum_{j=1}^{J}\sum_{i \in I_j} \mathbb{E}_{S_2}\left[N_i(T)\right]K(P_i^{S_2}, P_i^{S_2'})$$

$$= \sum_{i \in I_{j_2'}} \mathbb{E}_{S_2}\left[N_i(T)\right]K(P_i^{S_2}, P_i^{S_2'}).$$

By Corollary 3.1 in [40] and the definition of $S_2, S_2'$, we have

$$K(P_i^{S_2}, P_i^{S_2'}) \le \frac{(S_2(g_i) - S_2'(g_i))^2}{S_2'(g_i)(1 - S_2'(g_i))}$$

$$< \frac{(2\epsilon_2 g_{i_{j_2'}}^{-1})^2}{M_1 M_2}$$

$$\le \frac{4}{p_{\min}^2 M_1 M_2}\epsilon_2^2$$

for any $i \in I_{j_2'}$. Then, by combining the two preceding displays, we have

$$K(P_{S_2}, P_{S_2'}) = \sum_{i \in I_{j_2'}} \mathbb{E}_{S_2}\left[N_i(T)\right]K(P_i^{S_2}, P_i^{S_2'}).$$

$$< c_2\epsilon_2^2 \cdot \mathbb{E}_{S_2}[M_{j_2'}(T)] \tag{82}$$

$$\le c_2\frac{T}{J-1}\epsilon_2^2,$$

where $c_2 = \frac{4}{p_{\min}^2 M_1 M_2}$. It is easy to check that $\epsilon_2 = \kappa T^{-\frac{1}{3}}$ is sufficient to satisfy the monotonicity constraints of $S_2$ and $S_2'$. Therefore, by combining (81), (82) and $J \asymp \epsilon_2^{-1}$, there exists a constant $C_2 > 0$ such that

$$R(T, S_2) + R(T, S_2') \ge C_2 T \cdot T^{-\frac{1}{3}}$$

$$\gtrsim T^{\frac{2}{3}}.$$

It completes the proof for the case $\gamma \ge 1/3$.

$\square$

## C  Technical Lemmas

**Lemma C.1.** *Let $\Theta' = \{(\mathbf{S}_0, \beta) \in \Theta : S_{0,K} \ge M_1, S_{0,1} \le M_2, \|\beta\|_2 \le D\}$, where $M_1$, $M_2$ and $D$ are some positve constants such that $0 < M_1 < M_2 < 1$. Suppose that the assumption (A2) holds. Then, it holds that for any $\theta_1, \theta_2 \in \Theta'$,*

$$\mathcal{D}_H(p_{\theta_1}, p_{\theta_2}) \asymp \left(\int_{x \in \mathcal{X}}\sum_{p \in \mathcal{G}}|H_{\theta_1}(x, p) - H_{\theta_2}(x, p)|^2\, q(p|x)p_X(x)dx\right)^{\frac{1}{2}},$$

*where constants in $\asymp$ depend only on $M_1$, $M_2$, $D$ and $L$.*

*Proof.* By the assumption (A2), there exist positive constants $H_1$ and $H_2$, depending on $M_1$, $M_2$, $D$ and $L$, such that $0 < H_1 < H_\theta(x,p) < H_2 < 1$ for all $x \in \mathcal{X}$, $p \in \mathcal{G}$ and $\theta \in \Theta'$. Then, for any $\theta_1, \theta_2 \in \Theta'$, we have

$$
\begin{aligned}
\mathcal{D}_H^2(p_{\theta_1}, p_{\theta_2}) &= \int_{x \in \mathcal{X}} \sum_{p \in \mathcal{G}} \sum_{y=0,1} \left( \sqrt{p_{\theta_1}(x,p,y)} - \sqrt{p_{\theta_2}(x,p,y)} \right)^2 dx \\
&= \int_{x \in \mathcal{X}} \sum_{p \in \mathcal{G}} \left[ \left( \sqrt{H_{\theta_1}(x,p)} - \sqrt{H_{\theta_2}(x,p)} \right)^2 \right. \\
&\qquad \left. + \left( \sqrt{1 - H_{\theta_1}(x,p)} - \sqrt{1 - H_{\theta_2}(x,p)} \right)^2 \right] q(p|x)p_X(x)dx \\
&\asymp \int_{x \in \mathcal{X}} \sum_{p \in \mathcal{G}} |H_{\theta_1}(x,p) - H_{\theta_2}(x,p)|^2 \, q(p|x)p_X(x)dx,
\end{aligned}
$$

where the third identity holds because the derivative of the map $t \mapsto \sqrt{t}$ is bounded below and above by positive constants on the interval $[H_1, H_2]$.

$\square$

**Lemma C.2.** *Let $\Theta' = \{(\mathbf{S}_0, \beta) \in \Theta : S_{0,K} \geq M_1, S_{0,1} \leq M_2, \|\beta\|_2 \leq D\}$, where $M_1$, $M_2$ and $D$ are some positve constants such that $0 < M_1 < M_2 < 1$. Suppose that the assumption (A2) holds. Then, there exist positive constants $C_1$ and $C_2$ depending on $M_1, M_2, D$ and $L$ such that for any $\theta_1 = (\mathbf{S}_{0,1}, \beta_1), \theta_2 = (\mathbf{S}_{0,2}, \beta_2) \in \Theta'$ and $p \in \mathcal{G}$, it holds that*

$$
|H_{\theta_1}(X,p) - H_{\theta_2}(X,p)| \leq C_1|S_{0,1}(p) - S_{0,2}(p)| + C_2\|\beta_1 - \beta_2\|_2
$$

*almost surely.*

*Proof.* We decompose the term $|H_{\theta_1}(X,p) - H_{\theta_2}(X,p)|$ as

$$
\begin{aligned}
|H_{\theta_1}(X,p) - H_{\theta_2}(X,p)| &= |S_{0,1}(p)^{\exp(X^\top \beta_1)} - S_{0,2}(p)^{\exp(X^\top \beta_2)}| \\
&\leq |S_{0,1}(p)^{\exp(X^\top \beta_1)} - S_{0,2}(p)^{\exp(X^\top \beta_1)}| + |S_{0,2}(p)^{\exp(X^\top \beta_1)} - S_{0,2}(p)^{\exp(X^\top \beta_2)}|.
\end{aligned}
$$
(83)

For the first term of the preceding display, the mean value theorem on a map $t \mapsto t^c$ ($c > 0$ a constant) yields

$$
|S_{0,1}(p)^{\exp(X^\top \beta_1)} - S_{0,2}(p)^{\exp(X^\top \beta_1)}| = \exp(X^\top \beta_1)\overline{S}_0(p)^{\exp(X^\top \beta_1)-1}|S_{0,1}(p) - S_{0,2}(p)|,
$$

for some $\overline{S}_0(p)$ in $(S_{0,1}(p), S_{0,2}(p))$. Under the assumption (A2), by the Cauchy-Schwart inequality and the boundedness of $\beta_1$, we have $|X^\top \beta_1| \leq \|X\|_2\|\beta_1\|_2 \leq LD$ almost surely. Furthermore, $\overline{S}_0(p)$ is bounded away from 0 and 1. Then, there exists a positive constant $C_1$, depending on $M_1$, $M_2$, $D$ and $L$, such that $\exp(X^\top \beta_1)\overline{S}_0(p)^{\exp(X^\top \beta_1)-1} < C_1$. Therefore, the first term of (83) is bounded by $C_1|S_{0,1}(p) - S_{0,2}(p)|$. Similarly, for the second term of (83), applying the mean value theorem to the map $t \mapsto c'^{\exp(t)}$ ($c' > 0$ a constant) gives

$$
|S_{0,2}(p)^{\exp(X^\top \beta_1)} - S_{0,2}(p)^{\exp(X^\top \beta_2)}| = |\log S_{0,2}(p)|S_{0,2}(p)^{\exp(X^\top \bar\beta)} \exp(X^\top \bar\beta)|X^\top(\beta_1 - \beta_2)|,
$$

for some $\bar\beta$ between $\beta_1$ and $\beta_2$. Note that by the Cauchy-Schwartz inequality, $|X^\top(\beta_1 - \beta_2)| \leq \|X\|_2\|\beta_1 - \beta_2\|_2$. By assumption (A2) and the boundedness of $S_{0,2}$ and $\bar\beta$, there exists a positive constant $C_2$, depending on $M_1$, $M_2$, $D$ and $L$, such that $|\log S_{0,2}(p)|S_{0,2}(p)^{\exp(X^\top \bar\beta)} \exp(X^\top \bar\beta)\|X\|_2 < C_2$ almost surely. Combining these results with (83), we have

$$
|H_{\theta_1}(X,p) - H_{\theta_2}(X,p)| < C_1|S_{0,1}(p) - S_{0,2}(p)| + C_2\|\beta_1 - \beta_2\|_2
$$

almost surely for any $p \in \mathcal{G}$.

$\square$

**Lemma C.3.** *Suppose that the assumption (A5) holds. If $\lambda_{0,k} \sim Gamma(\alpha_k, \rho)$ are independent for $k = 1, \ldots, K$, where $A\epsilon^b \leq \alpha_k \leq M$, and $K\epsilon \leq N$ for some positive constants $A, \epsilon, b, M, N$ and $\rho$, then there exist positive constants $C_1$, $C_2$ and $C_3$ depending only on $p_{\min}, p_{\max}, A, b, M, N$ and $\rho$ such that*

$$\Pi\left(\|\boldsymbol{\Lambda}_0 - \boldsymbol{\Lambda}_0^*\|_\infty \leq \epsilon\right) \geq C_1 \exp\left(-C_2 K - C_3 K \log_- \epsilon\right).$$

*Proof.* First, we assume that $M = 1$, so that $\alpha_k \leq 1$ for all $k = 1, \ldots, K$. Fix $k \in \{1, \ldots, K\}$. In the model (2), we can represent $\Lambda_0(g_k)$ as $\delta \sum_{s=1}^k \lambda_{0,s}$. Similarly, $\Lambda_0^*(g_k)$ is given by $\delta \sum_{s=1}^k \Delta_{0,s}^*$, where $\Delta_{0,1}^* = \Lambda_0^*(g_1)/\delta$ and $\Delta_{0,s}^* = (\Lambda_0^*(g_s) - \Lambda_0^*(g_{s-1}))/\delta$ for $s = 2, \ldots, K$. By combining this and the preceding display, we have

$$\|\boldsymbol{\Lambda}_0 - \boldsymbol{\Lambda}_0^*\|_\infty = \max_{1 \leq k \leq K} |\Lambda_0(g_k) - \Lambda_0^*(g_k)|$$

$$\leq \max_{1 \leq k \leq K} \delta \sum_{s=1}^k \left|\lambda_{0,s} - \Delta_{0,s}^*\right|$$

$$= \delta \sum_{k=1}^K \left|\lambda_{0,k} - \Delta_{0,k}^*\right|.$$

Therefore, the probability on the left side of the lemma is bounded below by

$$\Pi\left(\|\boldsymbol{\Lambda}_0 - \boldsymbol{\Lambda}_0^*\|_\infty \leq \epsilon\right) \geq \Pi\left(\delta \sum_{k=1}^K \left|\lambda_{0,k} - \Delta_{0,k}^*\right| \leq \epsilon\right)$$

$$\geq \prod_{k=1}^K \Pi\left(\left|\lambda_{0,k} - \Delta_{0,k}^*\right| \leq C_p \epsilon\right),$$

where $C_p := (K\delta)^{-1}$ be a postive constant depending only on $p_{\min}$ and $p_{\max}$. Since $\lambda_{0,1}, \ldots, \lambda_{0,K}$ are independent variables with $\lambda_{0,k} \sim Gamma(\alpha_k, \rho)$, we have

$$\Pi\left(\|\boldsymbol{\Lambda}_0 - \boldsymbol{\Lambda}_0^*\|_\infty \leq \epsilon\right)$$

$$\geq \frac{\rho^{\sum_{k=1}^K \alpha_k}}{\prod_{k=1}^K \Gamma(\alpha_k)} \int_{\max\left(\Delta_{0,K}^* - C_p\epsilon, 0\right)}^{\Delta_{0,K}^* + C_p\epsilon} \cdots \int_{\max\left(\Delta_{0,1}^* - C_p\epsilon, 0\right)}^{\Delta_{0,1}^* + C_p\epsilon} \prod_{k=1}^K u_k^{\alpha_k - 1} \exp\left(-\rho \sum_{k=1}^K u_k\right) du_1 \cdots du_K.$$

By assumption (A5), there exist constants $M_1$ and $M_2$ such that $0 < M_1 \leq S_0^*(p_{\max}) < S_0^*(p_{\min}) \leq M_2 < 1$, and it holds that $N_2 \leq \Lambda_0(v) \leq N_1$ for all $v \in [p_{\min}, p_{\max}]$, where $N_1 = -\log M_1$ and $N_2 = -\log M_2$. Note that within the interval of integration, $\sum_{k=1}^K u_k \leq \Lambda_0^*(g_K)/\delta + C_p K\epsilon < N_1 C_p K + C_p K\epsilon$. Furthermore, for any $0 < \alpha_k \leq 1$, it holds that $\alpha_k \Gamma(\alpha_k) = \Gamma(\alpha_k + 1) \leq 1$. Therefore, the right side of the preceding display is bounded below by

$$\rho^{\sum_{k=1}^K \alpha_k} \exp(-\rho C_p K(N_1 + \epsilon)) \prod_{k=1}^K \left\{(\Delta_{0,k}^* + C_p\epsilon)^{\alpha_k} - (\max(\Delta_{0,k}^* - C_p\epsilon, 0))^{\alpha_k}\right\}.$$

By the mean value theorem, the terms of the product in the preceding display is bounded below by $\alpha_k(\overline{\Delta}_{0,k}^*)^{\alpha_k - 1} C_p\epsilon$ for some $\overline{\Delta}_{0,k}^* \in (\max(\Delta_{0,k}^* - C_p\epsilon, 0), \Delta_{0,k}^* + C_p\epsilon)$. Since $\overline{\Delta}_{0,k}^* < N_1 C_p K + C_p\epsilon$ and $\alpha_k - 1 \leq 0$ for all $k = 1, \ldots, K$, by combining the last two displays, we have

$$\Pi\left(\|\boldsymbol{\Lambda}_0 - \boldsymbol{\Lambda}_0^*\|_\infty \leq \epsilon\right)$$

$$\geq \rho^{\sum_{k=1}^K \alpha_k} \exp(-\rho C_p N_1 K)) \exp(-\rho C_p K\epsilon) \cdot (C_p\epsilon)^K (N_1 C_p K + C_p\epsilon)^{\sum_{k=1}^K \alpha_k - K} \prod_{k=1}^K \alpha_k.$$

Note that $N_1 C_p K + C_p\epsilon = 1/\epsilon \cdot (N_1 C_p K\epsilon + C_p\epsilon^2) \leq 1/(C'\epsilon)$ where $C' := 1/(N_1 N C_p + C_p(A^{2/b})^{-1})$ is a positive constant, as $K\epsilon \leq N$ and $A\epsilon^b \leq 1$ by assumption. Therefore, we have

$$\Pi\left(\|\boldsymbol{\Lambda}_0 - \boldsymbol{\Lambda}_0^*\|_\infty \leq \epsilon\right)$$

$$\geq \rho^{\sum_{k=1}^K \alpha_k} \exp(-\rho C_p N_1 K)) \exp(-\rho C_p N)(C_p\epsilon)^K (1/(C'\epsilon))^{\sum_{k=1}^K \alpha_k - K} (A\epsilon^b)^K$$

$$\geq C_1 \exp(-C_2 K - C_3 K \log_- \epsilon),$$

for positive constants $C_1 := \exp(-\rho C_p N)$, $C_2 := \rho C_p N_1 + \log_- \rho + \log_- A + \log_- C_p + \log_- C'$ and $C_3 := b+2$, where the first inequality holds because $K\epsilon \leq N$ and $A\epsilon^b \leq \alpha_k \leq 1$ by assumption, the second inequality holds because $\log x \geq -\log_- x$ for any $x > 0$, where $\log_-$ denotes the negative parts of logarithm. This concludes the proof in the case that $M = 1$.

We may assume without loss of generality that a $M$ is an positive integer. For each $k = 1, \ldots, K$, we can represent the $\lambda_{0,k}$ as the sum of independent random variables $(\lambda_{0,k,m} : m = 1, \ldots, M)$, where $\lambda_{0,k,m}$ is distributed from Gamma distribution with parameters $\alpha_{k,m} = \alpha_k/M$ and $\rho$ for $m = 1, \ldots, M$. Then, it satisfies the conditions of the lemma in the case of $M = 1$, with $K$ and $A$ being adjusted to $MK$ and $A/M$, respectively. The proof is then complete.

$\square$

**Lemma C.4.** *Under the assumption (A3), for a random sample $X_t = (X_{t,1}, \ldots, X_{t,d})$, there is a constant $\epsilon > 0$ such that $\mathbb{P}(|X_{t,j}| > \epsilon) > 0$ for all $j = 1, \ldots, d$.*

*Proof.* Suppose that for any $\epsilon > 0$, there exists $j' \in \{1, \ldots, d\}$ such that $\mathbb{P}(|X_{t,j'}| > \epsilon) = 0$. It follows that $\mathbb{P}(\|X_t\|_\infty > \epsilon) = 0$. Since $\epsilon > 0$ is an arbitrary number, we have $\mathbb{P}(\|X_t\|_\infty = 0) = 1$. Take $j^* = \mathrm{argmax}_{j=1,\ldots,d} |X_{t,j}|$ and $\beta_1, \beta_2$ such that $\beta_{1,j^*} \neq \beta_{2,j^*}$ and $\beta_{1,j} = \beta_{2,j}$ for $j \in [d] \setminus \{j^*\}$. Note that $X_t^\top(\beta_1 - \beta_2) = X_{t,j^*}(\beta_{1,j^*} - \beta_{1,j^*})$. Since $\mathbb{P}(|X_{t,j^*}| = 0) = 1$, we have $\mathbb{P}(X_t^\top(\beta_1 - \beta_2) = 0) = 1$. This contradicts the fact that $\mathbb{P}(X_t^\top(\beta_1 - \beta_2) \neq 0) > 0$ from the assumption (A3), and therefore the proof is complete.

$\square$

**Lemma C.5.** *If the pricing policy $\pi_l$ for each epoch $l$ is specified as in (5), then the assumption (A4) is satisfied.*

*Proof.* Let $q_l(\cdot \mid x)$ be the conditional probability mass function of $P_t$ given $X_t = x$ for $t \in \mathcal{E}_l$. Note that for any $x \in \mathcal{X}$ and $p \in \mathcal{G}$, we have

$$
\begin{aligned}
q_l(p \mid x) &= \pi_l(x)(\{p\}) \\
&\geq \eta_l/K \\
&\gtrsim \eta_l \cdot n_l^{-\gamma_l} \\
&\gtrsim \begin{cases} n_l^{-\frac{1+\gamma_l}{2}}(\log n_l)^{\frac{1}{2}} & \text{if } \gamma_l < \frac{1}{3}, \\ n_l^{-\gamma_l - \frac{1}{3}}(\log n_l)^{\frac{1}{2}} & \text{if } \gamma_l \geq \frac{1}{3}, \end{cases}
\end{aligned}
$$

where the first inequality holds by (5), the second inequality holds by (7), and the last inequality holds by (6). Thus, the conditional probability mass function $q_l(\cdot \mid x)$, parameterized by the policy $\pi_l$, satisfies the assumption (A4).

$\square$

**Lemma C.6.** *Suppose that the prior distribution $\Pi$ is specified as in (4), and the policy $\pi_l$ for each epoch $l$ is defined by (5). Suppose that assumptions (A1)-(A3), (A5) hold. Then, for every $\epsilon > 0$, there exist positive constants $C_1$ and $C_2$ depending on $L, B, p_{\min}, p_{\max}, \kappa, \alpha, \rho, a, b, n_1$ and $\epsilon$, such that for $l \geq C_1$,*

$$
\|\widehat{\mathbf{S}}_0^{l-1} - \mathbf{S}_0^*\|_\infty + \|\widehat{\beta}^{l-1} - \beta^*\|_2 \leq \epsilon
$$

*with probability at least $1 - \exp(-C_2 n_{l-1}^{1/3})$.*

*Proof.* We define the distance $\mathcal{D}_\infty(\theta_1, \theta_2)$ between $\theta_1 = (\mathbf{S}_{0,1}, \beta_1)$ and $\theta_2 = (\mathbf{S}_{0,2}, \beta_2)$ on $\Theta$ as

$$
\mathcal{D}_\infty(\theta_1, \theta_2) = \|\mathbf{S}_{0,1} - \mathbf{S}_{0,2}\|_\infty + \|\beta_1 - \beta_2\|_2.
$$

Let $q_l(\cdot \mid x)$ be the conditional probability mass function of $P_t$ given $X_t = x$ for $t \in \mathcal{E}_l$ in epoch $l$. By Lemma C.5, $q_l(\cdot \mid x)$ satisfies the assumption (A4) for every epoch $l$. Then, by Lemma A.1, for every $\epsilon > 0$ and $\gamma_{l-1} \in (0, 1]$, there exist positive constants $c_1, c_2$ and $c_3$ depending on $L, B, p_{\min}, p_{\max}, \kappa, \alpha, \rho$ and $\epsilon$ such that for $l \geq \lceil \log_2(c_1/n_1) \rceil + 1$,

$$\Pi(\mathcal{D}_\infty(\theta, \theta^*) \geq \epsilon/2 \mid \mathbf{D}_{l-1}) < c_2 \exp\left(-c_3 n_{l-1}^{\frac{1}{3}}\right) \tag{84}$$

with probability at least $1 - \exp(-c_3 n_{l-1}^{1/3})$.

We partition the parameter space $\widetilde{\Theta}$ into two subsets

$$\widetilde{\Theta}_1 = \{\theta \in \widetilde{\Theta} : \mathcal{D}_\infty(\theta, \theta^*) < \epsilon/2\},$$
$$\widetilde{\Theta}_2 = \{\theta \in \widetilde{\Theta} : \mathcal{D}_\infty(\theta, \theta^*) \geq \epsilon/2\}.$$

Then, we can decompose $\widehat{\theta}^{l-1}$ as

$$\begin{aligned}
\widehat{\theta}^{l-1} &= \int_{\widetilde{\Theta}} \theta \, d\widetilde{\Pi}(\theta \mid \mathbf{D}_{l-1}) \\
&= \int_{\widetilde{\Theta}_1} \theta \, d\widetilde{\Pi}(\theta \mid \mathbf{D}_{l-1}) + \int_{\widetilde{\Theta}_2} \theta \, d\widetilde{\Pi}(\theta \mid \mathbf{D}_{l-1}) \\
&= (1 - \tau_{l-1})\widehat{\theta}_1^{l-1} + \tau_{l-1}\widehat{\theta}_2^{l-1}, \tag{85}
\end{aligned}$$

where $\tau_{l-1} = \widetilde{\Pi}(\widetilde{\Theta}_2 \mid \mathbf{D}_{l-1})$. Here, $\widehat{\theta}_1^{l-1}$ and $\widehat{\theta}_2^{l-1}$ are the mean estimates of the probability measures resulting from the restriction and normalization of the truncated posterior distribution on the sets $\widetilde{\Theta}_1$ and $\widetilde{\Theta}_2$, respectively. It is easy to check that the function $\theta \mapsto \mathcal{D}_\infty(\theta, \theta^*)$ is convex and bounded over the domain $\widetilde{\Theta}$. By Jensen's inequality, we have

$$\begin{aligned}
\mathcal{D}_\infty(\widehat{\theta}_1^{l-1}, \theta^*) &\leq \int_{\widetilde{\Theta}_1} \mathcal{D}_\infty(\theta, \theta^*) \, d\widetilde{\Pi}_1(\theta \mid \mathbf{D}_{l-1}) \\
&< \epsilon/2, \tag{86}
\end{aligned}$$

where $\widetilde{\Pi}_1(\cdot \mid \mathbf{D}_{l-1})$ be the probability measure obtained by restricting and renormalizing $\widetilde{\Pi}(\cdot \mid \mathbf{D}_{l-1})$ to $\widetilde{\Theta}_1$, and the last inequality holds by the definition of $\widetilde{\Theta}_1$. On the event that the inequality (84) holds, for $l \geq \lceil \log_2(c_1/n_1) \rceil + 1$, we have

$$\begin{aligned}
\mathcal{D}_\infty(\widehat{\theta}^{l-1}, \theta^*) &\leq (1 - \tau_{l-1})\mathcal{D}_\infty(\widehat{\theta}_1^{l-1}, \theta^*) + \tau_{l-1}\mathcal{D}_\infty(\widehat{\theta}_2^{l-1}, \theta^*) \\
&< \frac{\epsilon}{2} + \frac{\Pi(\widetilde{\Theta}_2 \mid \mathbf{D}_{l-1})}{\Pi(\widetilde{\Theta} \mid \mathbf{D}_{l-1})} \mathcal{D}_\infty(\widehat{\theta}_2^{l-1}, \theta^*) \\
&\leq \frac{\epsilon}{2} + \frac{c_2 \exp(-c_3 n_{l-1}^{1/3})}{1 - c_2 \exp(-c_3 n_{l-1}^{1/3})} \cdot (1 + \sqrt{d}(a \vee b) + B),
\end{aligned}$$

where the first inequality holds because of the convexity of the function $\theta \mapsto \mathcal{D}_\infty(\theta, \theta^*)$ and (85), and the second inequality holds by (86) and the definition of $\tau_{l-1}$. The last inequality follows from $\Pi(\widetilde{\Theta} \mid \mathbf{D}_{l-1}) \geq 1 - \Pi(\widetilde{\Theta}_{l-1,2} \mid \mathbf{D}_{l-1})$, combined with inequality (84) and the boundedness of $\mathcal{D}_\infty$ over $\widetilde{\Theta}$ under the assumption (A1). Note that the second term on the right of the preceding display is upper bounded by $\epsilon/2$ for $n_{l-1} \geq (\log(c_2(1 + C_1)/C_1)/c_3)^3$, where $C_1 = \epsilon/(2(1 + \sqrt{d}(a \vee b) + B))$. Combining this result with the preceding display, we conclude that for $l \geq (\lceil \log_2(c_1/n_1) \rceil + 1) \vee (\lceil \log_2(C_2/n_1) \rceil + 1)$,

$$\mathcal{D}_\infty(\widehat{\theta}^{l-1}, \theta^*) < \epsilon,$$

with probability at least $1 - \exp(-c_3 n_{l-1}^{1/3})$, where $C_2 = (\log(c_2(1 + C_1)/C_1)/c_3)^3$. The proof is then complete. $\qquad\square$

**Lemma C.7.** *Suppose that assumptions (A1)-(A3), (A5) and (B1) hold. Let observations* $\mathbf{D}_l = \{(X_t, P_t, Y_t)\}_{t \in \mathcal{E}_l}$ *be i.i.d. copies of* $(X, P_l, Y)$, *where* $P_l$ *is a random variable distributed from* $\mathbb{Q}_l$ *as specified in Algorithm 1. We consider the collection of random variables* $\{\mathbb{M}(p) : p \in (p_{\min}, p_{\max})\}$, *where* $\mathbb{M}(p) := pS_0^*(p)^{\exp(X^\top \beta^*)}$. *Let* $P_c$ *denote the points of maximum of* $\mathbb{M}(p)$ *over* $(p_{\min}, p_{\max})$ *such that*

$$P_c \in \underset{p \in (p_{\min}, p_{\max})}{\operatorname{argmax}} \mathbb{M}(p). \tag{87}$$

*Then,* $P_l$ *converges to* $P_c$ *in distribution as* $l \to \infty$.

*Proof.* We consider the collection $\{\mathbb{M}_k(p) : p \in (p_{\min}, p_{\max})\}$ of random variables, where $\mathbb{M}_k(p) := p\widehat{S}_0^{l-1}(p)^{\exp(X^\top \widehat{\beta}^{l-1})}$. For each $l$, define the point of maximum of $\mathbb{M}_k(p)$ over $p \in \mathcal{G}$ by

$$\widehat{P}_l \in \underset{p \in \mathcal{G}}{\operatorname{argmax}} \mathbb{M}_k(p).$$

We first show that $\widehat{P}_l$ converges weakly to $P_c$. To see this, we need to verify the conditions of Theorem 1 of [8]. We say that $\mathcal{G}$ Painlevé-Kuratowski (PK) converges to $(p_{\min}, p_{\max})$ if

$$\{p \in (p_{\min}, p_{\max}) : \liminf_{n \to \infty} \inf_{g \in \mathcal{G}} |p - g| = 0\} = \{p \in (p_{\min}, p_{\max}) : \limsup_{n \to \infty} \inf_{g \in \mathcal{G}} |p - g| = 0\} = (p_{\min}, p_{\max}).$$

Let $N$ be the number of epochs for a given horizon $T$, satisfying $N = \log_2(T/n_1 + 1)$. As $l \to \infty$ implies $T \to \infty$, it is easy to see that the grid set $\mathcal{G}$ PK converges to the continuous interval $(p_{\min}, p_{\max})$.

We denote the conditional distribution of $P_l$ given $X$ by $\mathbb{Q}_l(\cdot \mid X)$, where $\mathbb{Q}_l(\cdot \mid X) = \pi_l(X)(\cdot)$ as defined in Algorithm 1. By the design of Algorithm 1 and the definition of $\eta_l$, the conditional distribution $\mathbb{Q}_l(\cdot \mid X)$ satisfies the assumption (A4) for all $l$. Then, by Lemma A.1 and Theorem 6.8 of [16], for $\epsilon > 0$, there exist positive constants $c_1$, $c_2$ and $c_3$ such that for sufficiently large $l$ with $n_{l-1} \geq c_1$ and for any $\gamma$, we have

$$\|\widehat{\mathbf{S}}_0^{l-1} - \mathbf{S}_0^*\|_\infty + \|\widehat{\beta}^{l-1} - \beta^*\|_2 < \epsilon + c_2 \exp\left(-c_3 n_{l-1}^{\frac{1}{3}}\right), \tag{88}$$

with probability at least $1 - \exp(-c_3 n_{l-1}^{1/3})$. Note that there exist constants $M_1$ and $M_2$ such that $0 < M_1 \leq S_0^*(p_{\max}) < S_0^*(p_{\min}) \leq M_2 < 1$ by assumption (A5). Let $C_0 := ((M_1 \wedge (1 - M_2))/2 \wedge B)$ and take $\epsilon < C_0/2$. For large $l$ such that $n_{l-1} \geq c_1 \vee (c_3^{-1} \log(2c_2/C_0))^3$, by (88), we have

$$\|\widehat{\mathbf{S}}_0^{l-1} - \mathbf{S}_0^*\|_\infty + \|\widehat{\beta}^{l-1} - \beta^*\|_2 < C_0,$$

with probability at least $1 - \exp(-c_3 n_{l-1}^{1/3})$. This implies that $\widehat{S}_{0,1}^{l-1} > M_1/2 > 0$, $\widehat{S}_{0,K}^{l-1} < (1 + M_2)/2 < 1$ and $\|\widehat{\beta}^{l-1}\|_2 < 2B$. Then, by Lemma C.2, for any $p \in (p_{\min}, p_{\max})$, we can decompose as

$$|\mathbb{M}_k(p) - \mathbb{M}(p)| = p|\widehat{S}_0^{l-1}(p)^{\exp(X^\top \widehat{\beta}^{l-1})} - S_0^*(p)^{\exp(X^\top \beta^*)}|$$
$$\leq C_1|\widehat{S}_0^{l-1}(p) - S_0^*(p)| + C_2\|\widehat{\beta}^{l-1} - \beta^*\|_2, \tag{89}$$

where $C_1$ and $C_2$ are positive constants depending on $M_1$, $M_2$, $L$, $B$ and $p_{\max}$, and the inequality holds almost surely. For each $p \in (p_{\min}, p_{\max})$, there exists $s \in \{2, ..., K\}$ such that $p \in [g_{s-1}, g_s]$. Then, we have

$$\widehat{S}_0^{l-1}(p) - S_0^*(p) \leq \widehat{S}_{0,s-1}^{l-1} - S_{0,s-1}^* + S_{0,s-1}^* - S_0^*(p)$$
$$\leq |\widehat{S}_{0,s-1}^{l-1} - S_{0,s-1}^*| + L_0\delta,$$

where the first inequality holds by the monotonicity of $\widehat{S}_0^{l-1}$, and the last inequality holds by $L_0$-Lipschitz continuity of $S_0^*$ under the assumption (A5). Similarly, we have $S_0^*(p) - \widehat{S}_0^{l-1}(p) \leq |\widehat{S}_{0,s}^{l-1} - S_{0,s}^*| + L_0\delta$. By combining this with (89),

$$|\mathbb{M}_k(p) - \mathbb{M}(p)| \leq C_3(\|\widehat{\mathbf{S}}_0^{l-1} - \mathbf{S}_0^*\|_\infty + \|\widehat{\beta}^{l-1} - \beta^*\|_2) + C_1L_0\delta,$$

where $C_3 = C_1 \vee C_2$ is a positive constant. Combining this with (88), for sufficiently large $l$ and $T$, we have

$$|\mathbb{M}_k(p) - \mathbb{M}(p)| \leq (C_3 + 1)\epsilon,$$

with probability at least $1 - \exp(-c_3 n_{l-1}^{1/3})$. Thus, for each $p \in (p_{\min}, p_{\max})$, $\mathbb{M}_k(p) \to \mathbb{M}(p)$ as $l \to \infty$ in probability, implying convergence in distribution. Since $p$ is arbitrary, $\mathbb{M}_k$ converges weakly $\mathbb{M}$ in $\ell^\infty(A)$ for every compact $A \subset (p_{\min}, p_{\max})$, where $\ell^\infty(A)$ denote the space of real-valued bounded functions on $A$. By the assumption (B1) and Theorem 1 of [8], we conclude that $\widehat{P}_l$ converges weekly to $P_c$.

By the design of policy $\pi_l$ in Algorithm 1, the random variable $P_l$ is defined as $P_l = R\widehat{P}_l + (1-R)U$, where $R$ is Bernoulli distributed with success probability $1 - \eta_l$. The variable $U$ is uniformly distributed on $\mathcal{G}$. Let $f : (p_{\min}, p_{\max}) \to \mathbb{R}$ be any bounded $L_1$-Lipschitz continuous function for some positive constant $L_1$. Then, we have

$$\begin{aligned}
|\mathbb{E}[f(P_l)] - \mathbb{E}[f(P_c)]| &= |\mathbb{E}[f(P_l)] - \mathbb{E}[f(\widehat{P}_l)] + \mathbb{E}[f(\widehat{P}_l)] - \mathbb{E}[f(P_c)]| \\
&\leq \mathbb{E}[L_1|(R-1)\widehat{P}_l + (1-R)U|] + |\mathbb{E}[f(\widehat{P}_l)] - \mathbb{E}[f(P_c)]| \\
&\leq 2L_1\eta_l p_{\max} + |\mathbb{E}[f(\widehat{P}_l)] - \mathbb{E}[f(P_c)]|,
\end{aligned}$$

where the first inequality holds because $f$ is $L_1$-Lipschitz function, and the last inequality holds because $\widehat{P}_l$ and $U$ are less than $p_{\max}$ almost surely. By the definition of $\eta_l$ and Portmanteau theorem, the right-hand side of the preceding display converges to 0 as $n \to \infty$. Then, we apply the Portmanteau theorem to conclude that $P_l$ converges weekly to $P_c$.

$\square$

## D  Extension to nonuniform grids

The assumption of equally spaced prices is made solely for technical convenience in developing the theory. However, with some additional technical work, our results can be readily extended to more general discrete price sets. For instance, one may consider a nonuniform grid $\mathcal{G} = \{g_1, \ldots, g_K\}$ satisfying

$$a\delta \leq |g_{k+1} - g_k| \leq b\delta \quad \text{for all } k = 1, \ldots, K,$$

where $\delta \asymp T^{-\gamma}$ and $a, b > 0$ are constants. This more general setting implies that our theoretical findings can be extended to more practical settings.

Importantly, since the regret rate in our analysis depends on the discrete set only through the sparsity level $\gamma$, this generalization does not fundamentally change the regret behavior. Therefore, while such an extension increases technical complexity, it does not yield additional theoretical insights.

## E  Discussion on the Cox PH model assumption

We here provide additional discussion on our choice of the Cox PH model, addressing both its suitability and potential concerns about model misspecification.

The key distinction between the Cox PH model and standard linear demand models lies in the use of the hazard function, which is a central concept in survival analysis. Unlike linear models, which model the conditional mean of a random variable, the Cox PH model focuses on modeling the hazard rate, a quantity that fully characterizes the survival distribution and is particularly well-suited to censored data settings. A key advantage of the Cox PH model is that it permits separate analysis of $\lambda_0$ and $\beta$, enabling theoretical development under minimal assumptions on the functional form of $\lambda_0$. This makes the Cox PH model an appropriate and principled choice for contextual dynamic pricing.

At the same time, we note that every model carries some risk of misspecification. Models that directly target the mean (e.g., linear or log-linear) are often highly sensitive to tail behavior and thus more vulnerable to misspecification. By contrast, models focusing on the hazard rate, such as the Cox PH model, tend to be more robust in these settings. A fully distribution-free approach might be preferable in order to avoid the risk of model misspecification. However, such an approach does not appear to be suitable in our context, as interval-censored data contains very limited information about the valuation distribution.

## F  Discussion on the variational Bayes estimator

In our theoretical analysis, the regret bounds are derived under the assumption that the estimator $\hat{\theta}^{l-1}$ corresponds to the posterior mean of the true Bayesian posterior, which contracts to the ground truth at the rates established in Theorems 3.1 and 3.2. In practice, we employ a variational Bayes (VB) approximation to obtain this estimator due to its computational efficiency in high-dimensional and nonparametric settings. The variational family used in our implementation is sufficiently expressive so that the VB posterior mean closely approximates the true posterior mean. As empirically demonstrated in [29], the considered VB approach performs comparably to, or better than, traditional MCMC in terms of estimation accuracy.

From a theoretical perspective, the regret bound depends directly on the convergence rate of $\hat{\theta}^{l-1}$. Therefore, if the VB posterior attains the same contraction rate as the true posterior, the regret guarantees remain valid. Recent advances in the theoretical study of VB methods (e.g., [46, 1, 45]) provide sufficient conditions under which the VB posterior achieves the same contraction rate as the full posterior. Although a rigorous regret analysis for VB-based estimation in our specific setting remains open, these results indicate that our regret guarantees continue to hold under appropriate contraction assumptions.

## G  Additional discussion on estimator replacement

Although we do not provide a formal proof, the Bayes estimator in our proposed algorithm could potentially be replaced by the NPMLE. However, even if so, careful selection of the exploration parameter $\eta_l$ is crucial for designing an optimal pricing policy. As empirically demonstrated in Section 6, our choice of the exploration parameter (6) substantially improves cumulative regret compared to the parameter choice in [7] (denoted as $\alpha_k$ in their notation), which employed the NPMLE.

## H  Details of the experimental setup

We use a Gamma prior with $\alpha_1 = \cdots = \alpha_K = \rho = 10^{-5}$, For a prior of $\beta$, we use a multivariate normal distribution, $N(\mathbf{0}, \mathbf{I}_d)$, where $\mathbf{I}_d$ denotes the $d \times d$ identity matrix. The truncated point estimator is computed within a parameter space of $\beta$ truncated to $[-10, 10]^d$. The proposed algorithm involves three hyperparameters: the first-epoch size $n_1$, the exploration parameters $\eta_1$ and $\eta_2$. These are tuned through grid search over the ranges $n_1 \in \{64, 128, 256\}$, $\eta_1 \in \{2^{-4/3}, 2^{-3/3}, 2^{-2/3}, 2^{-1/3}, 2^0\}$ and $\eta_2 \in \{2^{-12/2}, 2^{-11/2}, 2^{-10/2}, 2^{-9/2}\}$ with an initial period of $T_0 = 3000$ for each combination.

For a fair comparison, the hyperparameters of [7] are also tuned using the grid search procedure. We use the CoxCP algorithm with the $\epsilon$-greedy heuristic, as described in their experiments. This algorithm involves two hyperparameters: the first-epoch size $\tau_1$ and the degree of exploration parameter $\tau$. The hyperparameters are tuned over the ranges $\tau_1 \in \{64, 128, 256, 512, 1024\}$ and $\tau \in \{2^{-4}, 2^{-3}, 2^{-2}, 2^{-1}, 2^0\}$, following the procedure outlined in their work.

## I  Computational resources used

All experiments in this paper were conducted using a machine equipped with an Intel(R) Core(TM) i9-10900X CPU. No GPU was used.

