# OpenReview forum: "A Bayesian Approach to Contextual Dynamic Pricing using the Proportional Hazards Model with Discrete Price Data"
_NeurIPS.cc/2025/Conference — NeurIPS 2025 poster_

### Official Review · Reviewer_pkNR · 2025-06-13

**Clarity:** 3
**Significance:** 3
**Originality:** 3
**Rating:** 4
**Confidence:** 3

**Summary:**

This paper proposes BayesCoxCP, a novel semi-parametric contextual dynamic pricing algorithm. The algorithm is based on a Bayesian formulation of the Cox proportional hazards (PH) model and is specifically designed for settings where feasible prices are drawn from a discrete set rather than a continuous range. It shows that leveraging this discreteness can lead to improved regret performance, particularly when the price grid is sparse relative to the time horizon. The paper also establishes a minimax lower bound that matches the upper bound achieved by the BayesCoxCP algorithm.

**Questions:**

1. I was wondering why, when $\gamma = 1$, the regret upper bound $\tilde{O}(T^{2/3}+\sqrt{dT})$ still outperforms the bound $\tilde{O}(T^{2/3}d)$ from [6] in terms of the dimension $d$. What is the reason behind this?

2. Could the algorithm handle a price space containing discrete subintervals? For instance, each subinterval is $[g_k - \delta', g_k + \delta']$, with $\delta'$ potentially very small. As $\delta' \rightarrow 0$, this would result in a discrete price space. I am not seeking formal proofs; any discussion on this point would be appreciated.

[6] Young-Geun Choi, Gi-Soo Kim, Yunseo Choi, Wooseong Cho, Myunghee Cho Paik, and Min-hwan Oh. Semi-parametric contextual pricing algorithm using Cox proportional hazards model. In Proc. International Conference on Machine Learning, pages 5771–5786, 2023.

**Ethical Concerns:**

["NO or VERY MINOR ethics concerns only"]

**Final Justification:**

The paper is generally well-written and structured. Studying dynamic pricing problems in discrete price spaces introduces an interesting and novel perspective. The paper also proposes a new algorithm supported by solid theoretical results. However, it still has the limitation on the potential model misspecification. So I keep the borderline accept.

**Limitations:**

See weaknesses.

**Quality:**

3

**Strengths And Weaknesses:**

Strengths: The paper is generally well-written and structured. Studying dynamic pricing problems in discrete price spaces introduces an interesting and novel perspective. The paper also proposes a new algorithm supported by solid theoretical results.

Weaknesses:
1. The assumption of a known functional form for the hazard function (specifically, the Cox proportional hazards model) might limit practical applicability if the actual demand structure significantly deviates from this assumption.
2. The assumption of a (sparse) discrete price space might be restrictive. In practice, it is common to encounter closely spaced price candidates or even price subintervals (see questions below).


Minor comments:

1. If my understanding is correct, the derived regret bound would remain valid even for non-uniform grids in the price interval, provided a certain sparsity condition is satisfied (as discussed in Appendix D). Highlighting this generalization explicitly in the main text would enhance the paper's practical relevance. Additionally, formally presenting this result as a corollary could further strengthen the paper.

2. The experiments currently refer readers to [6] for performance comparisons for benchmark algorithms (linear and log-linear models). Providing direct comparisons or summarizing performance metrics in the current paper itself would improve readability and more clearly demonstrate the potential advantages of the proposed approach.

[6] Young-Geun Choi, Gi-Soo Kim, Yunseo Choi, Wooseong Cho, Myunghee Cho Paik, and Min-hwan Oh. Semi-parametric contextual pricing algorithm using Cox proportional hazards model. In Proc. International Conference on Machine Learning, pages 5771–5786, 2023.

---

> ### Author Rebuttal · Authors · 2025-07-31
>
> **W1** Concern about model misspecification under the Cox PH assumption.
>
> **A1** Indeed, every model carries some risk of misspecification. The primary motivation for using a semi-parametric model such as the Cox model is to mitigate this risk. Nevertheless, model misspecification is practically unavoidable, making robustness to such misspecification a critical concern for practitioners. We first offer several general perspectives on this issue, with a particular focus on censored data.
>
> Modeling the hazard rate, rather than the mean of a random variable, has a long-standing tradition in survival analysis. Estimating the mean of a survival random variable is challenging because it is often not directly observable due to censoring. In particular, the tail of the distribution plays a crucial role in determining the mean, yet it is rarely observed in censored data. This issue becomes especially severe in the case of interval-censored data, where no direct observations are available. Consequently, models that directly target the mean, such as linear or log-linear models, are highly vulnerable to model misspecification, particularly when the tail behavior is misspecified. In contrast, models targeting the hazard rate, such as the Cox model, tend to be more robust to model misspecification.
>
> A fully distribution-free approach might be preferable in order to avoid the risk of model misspecification. Unfortunately, however, such an approach does not appear to be suitable in our context, as interval-censored data contains very limited information about the valuation distribution. As a result, estimating $F(\cdot \mid X_t)$ (or $\lambda(\cdot \mid X_t)$) without strong structural assumptions becomes extremely challenging.
>
> In summary, we believe the Cox PH model offers a robust foundation for handling censored observations. Exploring alternative models, such as the proportional odds model, would be an interesting direction for future research.
>
> We thank the reviewer for raising this important point. We will consider including a more detailed discussion of model misspecification issues and possible alternatives in the final version to improve clarity.
>
> ---
>
> **W2 \& Q2** Question on extending the algorithm to handle discrete subintervals in the price space.
>
> **A2** We thank the reviewer for the insightful question. As we understand, you are considering a setting where the price set is a union of small subintervals centered around discrete grid points:
>
> $$
> \mathcal{G} = \bigcup_{k \in [K]} [g_k - \delta', g_k + \delta'],
> $$
>
> where $g_k - g_{k-1} = \delta$, $\delta \asymp n^{-\gamma}$, and $\delta' \asymp n^{-\gamma'}$. While we do not provide a formal analysis for this generalized case, we offer the following conjecture based on theoretical insights.
>
> We first assume that the subintervals are disjoint, which corresponds to $\gamma' \geq \gamma$. If $\gamma \geq 1/3$, the grid becomes sufficiently dense so that the union of subintervals behaves like a continuous price regime. In this case, we conjecture that the estimation error would match the rate for continuous pricing, i.e., $n^{-1/3}$.
>
> More intriguingly, when $\gamma < 1/3$, the regime becomes more subtle. In the extreme case where $\gamma = \gamma' = 0$, the price set becomes a finite union of non-shrinking subintervals, which resembles a continuous regime. Building on insights from Tang et al. (2012), the confidence interval width around each grid point in the discrete regime is roughly $n^{-(1-\gamma)/2}$. If each subinterval has width $\delta' \asymp n^{-\gamma'}$ satisfying $\gamma' \geq (1-\gamma)/2$, then each subinterval may lie entirely within the confidence region. This suggests that the algorithm could still behave similarly to the discrete regime.
>
> We emphasize that this remains a conjecture, and we leave the formal theoretical verification as an exciting direction for future work.
>
> ---
>
> **Q1** Question on why the regret bound shows better dimension dependence then Choi et al. (2023) when $\gamma = 1$.
>
> **A3** The key reason lies in the tightness of the entropy (log-covering number) bound used in the convergence analysis. In Appendix B.1 of Choi et al. (2023), the authors leverage an entropy bound derived from Huang (1996), which states that the $\epsilon$-covering number of the relevant function class is of the order $(1/\epsilon^d) (e^{1/\epsilon})$. Choi et al. (2023) approximated the log-covering number as $d/\epsilon$, but this approximation is loose. A more accurate bound is given by  $d \log(1/\epsilon) + 1/\epsilon$, which yields tighter control in the empirical process analysis. If this refined estimate is applied to their regret analysis, this leads to a tighter dependency on $d$. In particular, it would match the regret bound $\tilde{O}(T^{2/3} + \sqrt{dT})$ that we obtain in our setting when $\gamma = 1$. This suggests that the dimension dependence in Choi et al. (2023) is not intrinsic, but rather an artifact of a conservative entropy approximation.
>
> ---
>
> **minor1** Suggestion to highlight and formalize the extension to nonuniform grids
>
> **A4** We thank the reviewer for this helpful suggestion. As correctly noted, the derived regret bounds can be extended to the case of nonuniform grids, provided a certain sparsity condition is satisfied. Due to space constraints, we did not emphasize the nonuniform case in the main text. We agree that highlighting this generalization could enhance the paper’s practical relevance. We will revise the manuscript to emphasize this point more clearly, and we will consider formally presenting the result in Appendix D as a corollary in the final version.
>
> ---
>
> **minor2** Suggestion to include direct benchmark comparisons in the experimental results
>
> **A5** We thank the reviewer for the suggestion. As noted in Section 6, our current experiments are primarily designed to highlight the benefits of leveraging discrete support information. To demonstrate this aspect, we conducted a comparative evaluation with Choi et al. (2023), which also uses the Cox PH model under a well-specified setting.
>
> That said, we agree with the reviewer that providing direct comparisons with benchmark models such as linear and log-linear approaches would significantly improve the readability and clarity of our experimental section, rather than referring to external results. We will incorporate these comparisons into the main text in future revisions by adding additional experiments.
>
> ---
>
> **References**
> - Choi et al. Semi-parametric contextual pricing algorithm using Cox proportional hazards model. ICML, 2023.
> - Huang. Efficient estimation for the proportional hazards model with interval censoring. Annals of Statistics, 1996.
> - Tang et al. Likelihood based inference for current status data on a grid: A boundary phenomenon and an adaptive inference procedure. Annals of Statistics, 2012.

---

> > ### Comment · Reviewer_pkNR · 2025-08-05
> >
> > I appreciate the rebuttal, which provides helpful explanations and clarifications. I maintain my positive score for this work, which is the borderline for acceptance.

---

### Official Review · Reviewer_nRn3 · 2025-06-23

**Clarity:** 3
**Significance:** 2
**Originality:** 2
**Rating:** 4
**Confidence:** 4

**Summary:**

This paper studies contextual dynamic pricing when prices are restricted to a discrete set. The authors propose a Bayesian approach using a semi-parametric proportional hazards model, showing that incorporating discrete pricing information can significantly enhance performance depending on the relationship between the price support size and the time horizon. They derive both upper and lower regret bounds for the proposed method.

**Questions:**

1. Technical Novelty: The core contribution appears to be extending the semi-parametric Cox proportional hazards model for pricing (Choi et al., 2023) from continuous to discrete price settings. The key technical results build on generalizing Chae (2023), which deals with non-contextual interval-censored data, to the proportional hazards setting. The authors should better articulate the technical novelty. What specific challenges arise in the discrete and contextual setting that are not addressed by existing works? Why are existing techniques insufficient, and what new theoretical tools or results are developed to address these gaps?

References:

Choi et al. (2023): Semi-parametric contextual pricing with Cox PH model. ICML.

Chae (2023): Adaptive Bayesian inference for current status data on a grid. Bernoulli.


2. Modeling Assumptions and Practicality: The current formulation/algorithm design departs from realistic pricing scenarios in three key aspects:

(1) As stated in Section 2.1, the model assumes $(X_t, P_t, Y_t)$ are i.i.d., which is impractical in dynamic pricing—prices are typically adapted based on historical observations.

(2) Algorithm 1 assumes that in the exploration phase, the seller can uniformly sample prices from the known set $\mathcal{G}$. However, in practice, $\mathcal{G}$ may not be known in advance or may change over time and across contexts. In non-contextual settings, it is reasonable to explore a large price set early and narrow it down over time. In contextual settings, however, the appropriate price set is often time- and context-dependent, making the assumption of a fixed known $\mathcal{G}$ unrealistic.

(3) The algorithm relies on three tuning parameters: the first-epoch size \( n_1 \), and the exploration parameters \( \eta_1 \) and \( \eta_2 \). In Appendix Section F, the authors mention that these parameters are selected via grid search. However, in an online learning setting, it is unclear how such tuning would be performed in practice. What is the computational cost associated with jointly tuning all three parameters? Additionally, how should a practitioner determine appropriate ranges for \( \eta_1 \) and \( \eta_2 \), especially without access to validation data?


3. Estimation Methodology:
The paper states that a variational Bayesian method is used to obtain the estimator $\hat{\theta}^{l-1}$. How does the theoretical regret guarantee depend on the accuracy of this estimator? Are there any conditions required for the variational Bayesian method to ensure that the regret bounds hold?

**Ethical Concerns:**

["NO or VERY MINOR ethics concerns only"]

**Final Justification:**

After rebuttal: The authors have partially addressed my major concerns on the technical novelty and the model assumptions. Hence I have updated the rating accordingly.

**Limitations:**

yes

**Quality:**

2

**Strengths And Weaknesses:**

Strengths:

1. The problem setting is practically relevant and underexplored. While most recent work focuses on continuous pricing, discrete pricing is often more realistic in real-world applications.

2. The paper’s primary contribution lies in analyzing the estimation rate of the Bayesian proportional hazards model in this discrete setting and applying it to contextual dynamic pricing.

3. Despite its technical depth, the paper is generally accessible and well written.

Weaknesses:

1. The technical novelty beyond existing work appears limited.

2. Some key modeling assumptions are overly simplistic or unrealistic for real-world applications (see detailed questions below).

---

> ### Author Rebuttal · Authors · 2025-07-31
>
> **Q1** Technical Novelty
>
> **A1** We appreciate the reviewer’s thoughtful reading and for highlighting the need to better articulate the technical contributions beyond prior work.
>
> Our key contribution lies in conveying the message that, in dynamic pricing problems with a discrete price set, one can leverage the size of the price support to improve revenue. While our work is inspired by Chae (2023) and Choi et al. (2023), achieving the above goal required the following important technical developments.
>
> One of our key technical contributions is the realization that the exploration rate $\eta_l$ must also depend on the support size $|\mathcal{G}|$ to attain the optimal regret rate. Therefore, we utilize the support size information not only to construct the estimator $\hat\theta^{l-1}$ but also to determine the exploration rate, as specified in Eq (6) of the manuscript. This enables our algorithm to adapt to all values of $\gamma \in (0,1]$ and achieve optimal regret uniformly across this range. Balancing estimation accuracy and exploration in this discrete price setting is a technically crucial contribution not addressed in prior work.
>
> Second, although it is natural to expect that the result of Chae (2023) can be extended to survival models incorporating covariates, developing a rigorous extension without strong assumptions requires substantial non-trivial techniques and effort. This challenge is particularly pronounced in the Bayesian setting, where many parts of the proof are highly prior-specific in infinite-dimensional models. In our case, we would like to emphasize that the prior for the baseline hazard function is different from that used in Chae (2023), mainly for computational convenience, and this difference necessitates additional prior-specific technical developments.
>
> To the best of our knowledge, no prior work has established regret bounds for Bayesian contextual pricing algorithms under discrete support, nor have they addressed the interplay between grid sparsity, exploration scheduling, and estimation error. We thank the reviewer again for this important question, which gave us the opportunity to better clarify and articulate the core technical contributions of our work. We believe that these developments collectively address the important technical gaps not covered in existing works.
>
> ---
>
> **Q2** Modeling Assumptions and Practicality
>
> **A2-(1)** We would like to clarify that the i.i.d. assumption is not used for modeling the entire sequential process of dynamic pricing, but only within each epoch for the purpose of theoretical analysis. As described in Section 4, our algorithm adopts an epoch-based design, which partitions the time horizon $T$ into multiple epochs. Within each epoch, a fixed pricing policy is applied. This design allows us to treat the data $\{(X_t, P_t, Y_t)\}_{t \in \mathcal{E}_l}$ as i.i.d. conditioned on past observations up to epoch $1, ..., l-1$.
>     This structure provides significant technical convenience, enabling us to analyze the estimation error of point estimator $\hat\theta^{l-1}$ at each epoch. The use of such an epoch-based design is a common strategy in the contextual pricing literature and serves as a practical bridge between theory and sequential decision-making.
>
> **A2-(2)** We would like to emphasize that our setting assumes $\mathcal{G}$ depends on the level of the entire time horizon $T$, not varying across rounds $t$. This is a practically motivated assumption that sellers in real-world typically construct human-friendly discrete sets (e.g., $\mathcal{G}$=\{\\$5, \\$10, \\$15,...\}) given $T$, and $\mathcal{G}$ should be understood as a prescribed set of all possible prices from which a seller may choose. For instance, one seller may plan to sell products over $T=10000$ rounds using a fine grid of $K=1000$, while another seller may consider a coarser grid with only $K=10$. In this context, the former seller corresponds to a larger grid sparsity level $\gamma$. Hence, $\gamma$ can be viewed as a latent property that reflects the seller's preference in designing the price grid set $\mathcal{G}$. Since this latent $\gamma$ is typically unknown in advance, our key contribution is to develop an algorithm that adapts to all possible values of $\gamma \in (0,1]$, achieving optimal regret uniformly over this range.
>
> In this light, we believe that our discrete price setting is more realistic than previous works assuming a fully continuous price setting. We agree with the reviewer that extending the framework to allow for time- or context-dependent price sets (e.g., $K$ is a function of both $T$ and the context dimension $d$) is an interesting direction for future work. We thank the reviewer again for encouraging us for pointing out a promising avenue for future extensions.
>
> **A2-(3)** We agree with the reviewer that hyperparameter tuning in online learning is inherently non-trivial, which remains an active area of research (refer to Cha and Cho (2025); Lee et al. (2024)). Among recent works in contextual dynamic pricing, Fan et al. (2024) propose tuning hyperparameters at the start of each epoch based on data from previous epochs. In contrast, Choi et al. (2023) adopt a more practical approach, which is similar to ours, where hyperparameters are tuned only once using data collected during an initial period $T_0$. This approach is more computationally efficient, particularly when the total horizon $T$ is large.
>
> To assess the practical cost of tuning, we measured the average computation time required per hyperparameter configuration in our simulation setup, where the initial period length is $T_0=3000$ and the grid size is $K=100$. The average time was 23.4 seconds, making the total tuning procedure computationally efficient in practice.
>
> In setting where a validation set is unavailable, we suggest that practitioners select hyperparameters based on prior beliefs about the desired level of exploration. If stronger exploration is preferred, larger values for $n_1$, $\eta_1$, and $\eta_2$ should be used; for more aggressive exploitation, smaller values are appropriate. As a practical guideline, we suggest choosing $\eta_1$ and $\eta_2$ below 1 to avoid excessive exploration.
>
> ---
>
> **Q3** Estimation Methodology
>
> **A3** We thank the reviewer for raising this important point. In our theoretical analysis, the regret bounds are derived under the assumption that the estimator $\hat\theta^{l-1}$ corresponds to the posterior mean of the true Bayesian posterior, which contracts around the ground truth at the rate characterized in Theorems 3.1 and 3.2.
>
> In practice, we employ a variational Bayes (VB) method to approximate this posterior due to its computational efficiency in high-dimensional and nonparametric settings. While the regret guarantees are not derived explicitly for the VB approximation, our implementation uses a variational family that is sufficiently expressive to closely match the posterior mean in all experiments. As empirically demonstrated in Liu et al. (2024), the considered VB approach performs comparably to, and sometimes even better than, traditional MCMC methods in terms of estimation accuracy.
>
> From a theoretical perspective, as the reviewer points out, the use of a VB estimator (i.e., a point estimator based on the variational posterior distribution) naturally raises questions about its impact on the regret guarantees. Although a formal regret analysis for VB-based estimation is beyond the scope of the present work, we can offer a general theoretical perspective.
>
> According to our regret analysis, the regret bound depends directly on the convergence rate of the estimator $\hat{\theta}^{l-1}$ (see lines 296–297). Therefore, if the VB estimator attains the same convergence rate as required in our analysis, the regret guarantees continue to hold.
>
> In this regard, several general theoretical frameworks have been developed for analyzing the contraction rates of VB posteriors (e.g., Zhang and Gao (2020); Alquier and Ridgway (2020); Yang et al. (2020)). These results provide sufficient conditions under which the VB posterior achieves the same contraction rate as the full posterior, thereby preserving the regret guarantees in our setting. While a rigorous theoretical treatment of VB methods in our specific setting remains an open question, we greatly appreciate the reviewer’s insightful comment and will include a detailed discussion of this point in the final version of the paper.
>
> ---
>
> **References**
> - Alquier and Ridgway. Concentration of tempered posteriors and of their variational approximations. Annals of Statistics, 2020.
> - Cha and Cho. Hyperparameters in continual learning: A reality check. TMLR, 2025.
> - Chae. Adaptive Bayesian inference for current status data on a grid. Bernoulli, 2023.
> - Choi et al. Semi-parametric contextual pricing algorithm using Cox proportional hazards model. ICML, 2023.
> - Fan et al. Policy optimization using semiparametric models for dynamic pricing. JASA, 2024.
> - Lee et al. Hyperparameter selection in continual learning. arXiv, 2024.
> - Liu et al. Variational Bayesian approach for analyzing interval-censored data under the proportional hazards model. Computational Statistics \& Data Analysis, 2024
> - Yang et al. $\alpha$-variational inference with statistical guarantees. Annals of Statistics, 2020.
> - Zhang and Gao. Convergence rates of variational posterior distributions. Annals of Statistics, 2020.

---

> > ### Comment · Reviewer_nRn3 · 2025-08-06
> >
> > After rebuttal: The authors have partially addressed my major concerns on the technical novelty and the model assumptions. Hence I have updated the rating accordingly.

---

### Official Review · Reviewer_aTCm · 2025-06-28

**Clarity:** 2
**Significance:** 3
**Originality:** 2
**Rating:** 4
**Confidence:** 4

**Summary:**

This paper addresses the contextual dynamic pricing problem under discrete price constraints, which better reflects many real-world retail settings where prices must come from a fixed grid. The author proposes a novel Bayesian algorithm, BayesCoxCP, which leverages the semi-parametric Cox proportional hazards model to estimate customer valuation distributions from interval-censored (binary) purchase feedback. The author uses an epoch-based pricing strategy to maintain i.i.d. data generation within every epoch to make the semi-parametric estimation doable and uses adaptive epsilon-greedy algorithm in every epoch. The author also proves a high probability regret upper bound to demonstrate that when the pricing action space is sparse, the regret rate could be better than the continuous or dense action space.

**Questions:**

See Strengths and Weakness

**Ethical Concerns:**

["NO or VERY MINOR ethics concerns only"]

**Final Justification:**

I maintain my score. Thank you.

**Limitations:**

See Strengths and Weakness

**Paper Formatting Concerns:**

no concern

**Quality:**

3

**Strengths And Weaknesses:**

Strength: The paper contains theoretical upper bound, lower bound, and empirical evaluation, which is quite complete and thorough. The algorithm design is concise in principle and easy to understand. This paper also uses COX PH model in a discrete space instead of continuous space and observes a phase transition phenomenon with respect to the sparsity parameter $gamma$, which is interesting.

Weakness: I am not fully convinced by the setting. The grid length $\delta=\kappa n^{-\gamma}$, where $n$ is the number of sample points. In the sequential contextual pricing setting, doesn’t this mean that in different rounds, or at least in different epochs, the action space of the agent is different? This is a little weird to me. Are there any existing papers that model the action space sparsity in such a way? I think the author needs to justify the setting in a more profound way.

I tried to read the technical proof details, and I find it a little difficult to follow. It would be much better if the author could provide a proof sketch or a roadmap about how every lemma is fully used. Could you please provide a sketch about the proofs of thm 3.1, 3.2?

The COX PH model, to my understanding, is a way to handle and model data censoring mechanism. It allows tractable estimation to bypass censored observations. What if the data generation mechanism is more general than that? Do you have a more principled way to handle it, rather than this restricted COX model with Gamma prior? I think some discussion about that is needed at least. I feel that there should be a challenge similar to data censorship in this paper, but the author doesn’t mention that explicitly and bypasses it through COX model.

---

> ### Author Rebuttal · Authors · 2025-07-31
>
> **W1** Justification of our setting
>
> **A1** We clarify that $\delta = \kappa n^{-\gamma}$ is only used in the i.i.d. setup discussed in Sections 2 and 3, where $n$ denotes the number of observations. In contrast, our sequential contextual pricing setting is introduced in Section 4, where the grid resolution is defined as $\delta = \kappa T^{-\gamma}$ given time horizon $T$. Importantly, under this formulation, the discrete price set (action space) $\mathcal{G}$ remains unchanged across all epochs and rounds.
>
> In our setup, given $T$, the discrete price set $\mathcal{G}$ should be interpreted as a pre-specified set of all possible prices from which a seller may choose. In practice, prices are often selected from human-friendly discrete sets (e.g., $\mathcal{G}$=\{\\$5, \\$10, …\}), and multiple observations are typically available for each. For instance, one seller may plan to sell a product over $T=10000$ rounds using a fine grid of $K=1000$, while another seller may consider a coarser grid with only $K=10$. In this example, the former seller corresponds to a larger value of $\gamma$, whereas the latter corresponds to a smaller $\gamma$. Thus, $\gamma$ can be interpreted as a latent property that captures the seller's preference in designing the price set $\mathcal{G}$. Since this latent $\gamma$ is typically unknown in advance, our key contribution is to develop an algorithm that adapts to all possible values of $\gamma \in (0, 1]$, achieving optimal regret uniformly over this range.
>
> We emphasize that our framework does not allow the price set $\mathcal{G}$ to change over time as $t$ increases; it remains fixed throughout all epochs and rounds. Of course, it is possible that, up to a certain point, some prices in $\mathcal{G}$ may not have been observed. Exploring such settings would be an interesting direction for future research.
>
> ---
>
> **W2** Request for a proof sketch or roadmap
>
> **A2** We sincerely thank the reviewer for carefully engaging with the technical sections and for this insightful suggestion. We appreciate your effort to understand the proof structure and agree that providing a clear roadmap can improve the accessibility of our results.
>
> Below, we provide a high-level proof roadmap outlining how each of the main lemmas and theorems are logically connected.
>
> **Posterior Convergence Rate (Theorem 3.1, 3.2)**  We analyze the convergence rate of the posterior under both discrete ($\gamma < 1/3$) and continuous ($\gamma \geq 1/3$) regimes. The roadmap is as follows:
>
> $$
> \text{Lemma A.1, A.6} \to \text{Theorem 3.1}
> $$
>
> $$
> \text{Lemma A.1, A.7} \to \text{Theorem 3.2}
> $$
>
> - Lemma A.6, A.7: Provide covering number bounds in the discrete and continuous regimes, respectively.
> - Lemma A.1: Establish posterior consistency under mild assumptions.
> - Theorem 3.1, 3.2: Derive the posterior convergence rates.
> - Lemma C.1-C.4: Provide the decomposition of the Hellinger distance and the prior mass condition required for Lemma A.1, Theorem 3.1, and Theorem 3.2.
>
> **Regret Analysis (Theorem 5.2)** The regret analysis builds upon the posterior convergence results and decomposes regret across epochs:
>
> $$
> \text{Theorem 3.1, 3.2, Lemma C.5} \to \text{Lemma 5.1}
> $$
>
> $$
> \text{Lemma C.2, C.6, C.7} \to \text{Lemma B.1}
> $$
>
> $$
> \text{Lemma 5.1, Lemma B.1} \to \text{Lemma B.2} \to \text{Theorem 5.2}
> $$
>
> - Lemma 5.1: Bound the estimation error of the point estimator $\widehat{\theta}^{l-1}$ using posterior concentration.
> - Lemma B.1: Decompose the regret into estimation error and exploration terms.
> - Lemma B.2: Derive the regret during each epoch.
> - Theorem 5.2: Aggregate per-epoch regret to derive a total regret upper bound.
> - Lemma C.2, C.6, C.7: Technical lemmas required for Lemma 5.1 and Lemma B.2.
>
> We will incorporate this roadmap in the final version to improve the clarity and accessibility of our technical proofs.
>
> ---
>
> **W3** Request for discussion on more general censoring mechanism.
>
> **A3** We thank the reviewer for the detailed and thoughtful comment. We would like to clarify why the Cox PH model is both a principled and appropriate choice, particularly in the presence of censored observations.
>
> The key distinction between the Cox PH model and standard linear models lies in the use of the hazard function, which is a central concept in survival analysis. Unlike the linear model, which typically models the conditional mean of a random variable, the Cox PH model focuses on modeling the hazard rate, a quantity that fully characterizes the survival function. This is especially important in censored data settings, where direct observations of event times (in our case, customer valuations) are often unavailable or incomplete. In addition, the hazard function is particularly useful since it is only required to be non-negative and integrable. This flexibility makes hazard-based modeling especially well-suited to censored data.
>
> In our work, we adopt the Cox PH model, which takes the form: $\lambda(v \mid x) = \lambda_0(v) \exp(x^\top \beta)$. A key advantage of this model is that it permits separate analysis of $\lambda_0$ and $\beta$, enabling theoretical development under minimal assumptions (e.g., non-negativity) on the functional form of $\lambda_0$. This separability facilitates both flexibility in modeling and tractability in analysis.
>
> In contrast, linear models such as $v = x^\top \beta + \epsilon$, require explicit assumptions on the error distribution, with common choices including the Gaussian and extreme value distributions. However, these assumptions can be quite restrictive, particularly in censored data where estimating the tail behavior of $\epsilon$ is difficult. These limitations were a primary reason we use the Cox PH model in our framework.
>
> That said, we emphasize that our methodology is not inherently restricted to the Cox PH model. Other semi-parametric models commonly used in survival analysis, such as the Proportional Odds model, or the Accelerated Failure Time model, can also be considered within our framework. Extending our contextual dynamic pricing problem to these models under a discrete price setting is a promising direction for future work.
>
> In implementing the Cox PH model, we place a Gamma prior on the baseline hazard function mainly for computational convenience. We would like to clarify that this choice does not reduce the inherent complexity of the model, nor does it restrict the model’s generality.
>
> We again thank the reviewer for raising this important point and will incorporate a summary of the above discussion into the final version.

---

> > ### Comment · Reviewer_aTCm · 2025-08-05
> >
> > I appreciate the response, which provides helpful explanations and clarifications. I maintain my score for this submission.

---

### Official Review · Reviewer_Zntn · 2025-06-30

**Clarity:** 3
**Significance:** 3
**Originality:** 3
**Rating:** 5
**Confidence:** 4

**Summary:**

The authors consider the Cox proportional hazards model in the context of binary semi-parametric contextual dynamic pricing. Based on the non-contextual statistical estimation results under this Cox PH model from the statistics literature, they develop similar ones under the contextual modeling. Then an epoch-based algorithm design is utilized to incorporate those estimation results, along with imposed exploration probabilities similar to the epsilon-greedy algorithms.

Matching regret upper and lower bounds are derived. The finding of improved regrets for well-designed algorithms under discrete prices is interesting, insightful and valuable in practice.

**Questions:**

1. For Cox PH model, statistical estimation results are developed from survival analysis literature and contribute to the optimal dynamic pricing algorithms. Could the authors discuss whether extensions, still leveraging the case 1 interval-censored data structure, but towards other semi-parametric pricing models, are possible?

2. The nonuniform grids are discussed in Appendix D. The authors claim that "with some additional technical work, our results can be readily extended to more general discrete price sets", and "this generalization does not fundamentally change the regret behavior". Could the authors discuss in more details on 1) what additional technical complexities arise for nonuniform grids, and 2) why the generalization will not fundamentally change the regret behavior, is there any evidence from statistical estimation theory in survival analysis?

3. The convergence rates of the posterior distributions in Theorem 3.1, Theorem 3.2 generalize the results in [4] to incorporate contextual information $X_{t}$. This key technical development in estimation errors help build the contextual dynamic pricing algorithms. Could the authors discuss the difficulties and challenges in this generalization?

4. There is a phase transition of the convergence rates ($\gamma \geq 1/3$ and $\gamma < 1/3$) in both the existing non-contextual and newly-developed contextual scenarios, which lead to the same transition in the derived regret rates. Could the authors give more intuitive explanations and illustrations for this phenomenon?

5. In lines 309 - 311, the authors compare their regret rates with those in [6]. However, [6] considers continuous prices and thus enjoy a different benchmark. Intuitively, the continuous optimal price yields higher revenues than the discrete best price (the optimal price with the discrete set). Is it appropriate to conduct the regret comparison if the benchmarks are truly different? How would the regret rates of the authors' proposed algorithm transfer to ones under continuous prices?

6. More discussions on the choice of $\eta_{\ell}$, and satisfaction of Assumption (A4) may be added, as both of them are crucial elements of the proposed algorithm.

**Ethical Concerns:**

["NO or VERY MINOR ethics concerns only"]

**Final Justification:**

The authors have provided helpful explanations and clarifications. I maintain my score for this submission.

**Limitations:**

Yes.

**Paper Formatting Concerns:**

I did not notice any major formatting issues in this paper.

**Quality:**

4

**Strengths And Weaknesses:**

Strengths:
1. Although not first introduced in this paper, interpreting the purchase data from the binary semi-parametric contextual pricing model as the case 1 interval-censored data in survival analysis effectively captures the intrinsic nature of the data and helps bridge and revitalize the two fields.
2. The considered discrete pricing setting is both practical and valuable, and new theoretical insights for this setting are delivered through the contributions of this paper.
3. The essential statistical estimation analysis of this paper is based on and developed from solid research in statistic literature.
4. The technical material is delivered with high degree of clarity and rigor.

Weaknesses:
1. More discussions are needed: 1) possibility of extension beyond Cox PH to other models (See Question 1), 2) technical differences when dealing with nonuniform grids (See Question 2), 3) challenges in generalize the non-contextual convergence rates of posterior distributions to the contextual scenario, 4) intuitive explanation of the phase transition of estimation errors and regret rates, 5) whether comparison of regret rates with those for continuous prices makes sense, and whether the authors' derived regret rates for discrete prices can transfer to ones for continuous prices. 6) crucial elements of the algorithm, e.g., choice of $\eta_{\ell}$ and satisfaction of Assumption (A4).

---

> ### Author Rebuttal · Authors · 2025-07-31
>
> **Q1** Extensions to other semi-parametric models under interval-censored data
>
> **A1**  We thank the reviewer for highlighting the connection between statistical estimation results and pricing algorithms.
>
> Our key theoretical insight is that once the price is discretely supported, one can construct a better pricing algorithm by leveraging this support information, which leads to improved regret rates. While our work focuses on the Cox PH model, we agree that extensions to other semi-parametric models under the case 1 interval-censored data structure are both plausible and worth exploring. For example, models such as the Proportional Odds (PO) model or the Accelerated Failure Time (AFT) model can be considered within the same framework.
>
> If these models permit the incorporation of discrete support information, their performance could potentially be improved in a similar fashion. From a technical perspective, the Cox PH model permits separate analysis of the baseline function $F_0$ and the regression parameter $\beta$ (see Theorem 3.3 in Huang (1996)). Once such separability is established, the theoretical analysis of $F_0$ with discrete support information becomes more tractable. Notably, Theorem 3.3 in Huang (1995) also implies that such separability holds in the PO model, suggesting that discrete support information may be exploited for pricing algorithm design in that setting.
>
> A detailed extension to such directions across a broader class of models would be interesting for future work.
>
> ---
>
> **Q2** Technical challenges and justification for regret behavior under nonuniform grids
>
> **A2** In the main text, we focus on uniform grids where the grid spacing satisfies $|g_{k+1} - g_k| = \delta$ for all $k$. In contrast, Appendix D discusses nonuniform grids where the spacing satisfies $a\delta \leq |g_{k+1} - g_k| \leq b\delta$ for some constants $0 < a \leq b < \infty$. In both cases, the grid spacing satisfies the asymptotic equivalence $|g_{k+1} - g_k| \asymp \delta$, ensuring the grid resolution is still characterized by $\delta$.
>
> As a result, using nonuniform grids affects only the constant factors in the regret bound through their dependence on $a$ and $b$, while the key dependence on the sparsity level $\gamma$ remains unchanged. Hence, this generalization does not fundamentally change the regret behavior.
>
> ---
>
> **Q3** Challenges in generalizing posterior convergence rates to contextual settings
>
> **A3** Our first key technical contribution lies in extending the convergence rate analysis from the non-contextual case in Chae (2023) to the Cox PH model with covariates. This extension requires substantial non-trivial techniques and effort without strong assumptions. This challenge is particularly pronounced in the Bayesian setting, where many parts of the proof are highly prior-specific in infinite-dimensional models. In our case, we would like to emphasize that the prior for the baseline hazard function is different from that used in Chae (2023), mainly for computational convenience, and this difference necessitates additional prior-specific technical developments.
>
> A second technical difficulty lies in bounding the Hellinger distance when incorporating covariates. In our setting, we show that the Hellinger distance can be bounded by a separable and additive combination of the distances between the baseline functions and the regression parameters:
>
> $$
> D_H (p_{\theta}, p_{\theta^\*}) \leq C_1 \\| \mathbf{S}_0 - \mathbf{S}_0^\* \\|_2 + C_2 \\| \beta - \beta^\* \\|_2,
> $$
>
> as shown in Lemmas C.1 and C.2. This additive structure is crucial for controlling the Hellinger metric entropy in the posterior contraction analysis (see Lemma A.6 and A.7).
>
> To enable such analysis, it is essential that the posterior distribution concentrates around the true parameter, i.e., we require posterior consistency (Lemma A.1). Establishing this result under a semi-parametric model like the Cox PH model is technically demanding.
>
> A third technical point is related to Assumption (A4), which is required for proving Lemma A.1. As discussed in Remark A.2, one could impose a stronger condition than (A4) to simplify the proof and derive results as in (8) uniformly for all $\gamma$. However, such a restrictive condition would require larger values of the exploration parameter $\eta_l$, leading to higher regret due to excessive exploration (see lines 296–297). Hence, adopting the weaker assumption (A4) is essential in the contextual pricing problem, and proving posterior consistency under this condition is a delicate and non-trivial contribution. For further technical details, we refer the reviewer to Appendix A.1.
>
> ---
>
> **Q4** Phase transition in covergence and regret rates across $\gamma < 1/3$ and $\gamma \geq 1/3$
>
> **A4** We build upon a key insight from Tang et al. (2012), who first observed this phenomenon in non-contextual survival analysis. The underlying intuition is closely related to isotonic regression problems, where one aims to estimate a non-decreasing function such as the baseline function $F_0$.
>
> When $\gamma < 1/3$, the grid of prices is sufficiently sparse, meaning that grid points are far enough. In this case, the estimation problem at each grid point becomes nearly independent of the others. As a result, simple pointwise estimators are already well-ordered and directly serve as valid solutions to the isotonic regression problem. This leads to improved estimation efficiency and faster convergence rates.
>
> In contrast, when $\gamma \geq 1/3$, the grid becomes denser and neighboring points are more closely spaced. Consequently, the pointwise estimators can fluctuate and may violate the monotonicity. In such cases, local pooling over neighboring grid points becomes necessary to construct a valid non-decreasing estimator, which limits the convergence rate.
>
> This transition in estimation behavior naturally extends to the regret bounds in the dynamic pricing problem. Notably, regret in our setting is influenced not only by estimation error but also by the degree of exploration (see lines 296-297). When $\gamma < 1/3$, sparse grids allow accurate estimation at each grid point with minimal exploration. This enables the algorithm to prioritize exploitation, as reflected in the decreasing exploration parameter $\eta_l$ defined in Eq (6).
>
> On the other hand, for $\gamma \geq 1/3$, denser grids require more frequent exploration to maintain learning efficiency across the entire price space. This increased exploration effort leads to a higher regret bound. Thus, both estimation error and the degree of exploration jointly explain the phase transition observed in our regret analysis.
>
> ---
>
> **Q5** Validity of regret comparison with Choi et al. (2023) under different benchmarks
>
> **A5** We appreciate the reviewer's concern regarding the validity of comparing regret bounds under different benchmarks, and we would like to clarify that our discrete price setting naturally interploates between discrete and continuous regimes.
>
> In particular, this setting is parameterized by the grid size $K$, or equivalently the sparsity level $\gamma$. At one extreme, when $K = c$ (i.e., $\gamma = 0$) for some constant $c > 0$, the price set is fixed and highly sparse. At the other extreme, when $K=T$ (i.e., $\gamma = 1$), the grid becomes dense enough to cover the continuous price setup. Our algorithm is designed to adapt to any $\gamma \in (0,1]$, and thus the derived regret bounds naturally extend to the continuous regime, making them comparable to those in Choi et al. (2023). The main point in lines 309-311 is to emphasize that our algorithm can achieve faster regret rates by leveraging the discreteness when the grid is sparse.
>
> Moreover, we conducted the comparison between both algorithms under the same discrete pricing benchmark by varying the grid size. As shown in Figure 1, our algorithm significantly outperforms the method in Choi et al. (2023) when the grid is sparse, due to its ability to utilize the structure of the discrete support.
>
> ---
>
> **Q6** More discussions on the choice of $\eta_l$ and satisfaction of Assumption (A4)
>
> **A6** We appreciate the reviewer's attention to the role of $\eta_l$ and assumption (A4), both of which are indeed central to our algorithm.
>
> As noted in lines 296–297, the choice of $\eta_l$ directly influences the regret bound by balancing exploration and exploitation across epochs. Specifically, $\eta_l$ controls the degree of uniform exploration over the grid, which is reflected in $q(\cdot \mid x)$. Since assumption (A4) requires a lower bound on $q(\cdot \mid x)$, an appropriate choice of $\eta_l$ is essential to ensure this condition is satisfied. This connection is formally established in Lemma C.5.
>
> We will revise the manuscript to better highlight the role of $\eta_l$ and its relationship to (A4).
>
> ---
>
> **References**
> - Chae. Adaptive Bayesian inference for current status data on a grid. Bernoulli, 2023.
> - Choi et al. Semi-parametric contextual pricing algorithm using Cox proportional hazards model. ICML, 2023.
> - Huang. Maximum likelihood estimation for proportional odds regression model with current status data. Analysis of Censored Data, 1995.
> - Huang. Efficient estimation for the proportional hazards model with interval censoring. Annals of Statistics, 1996.
> - Tang et al. Likelihood based inference for current status data on a grid: A boundary phenomenon and an adaptive inference procedure. Annals of Statistics, 2012.

---

> > ### Comment · Reviewer_Zntn · 2025-08-05
> >
> > Thank you for the careful and detailed response. I maintain my score for this submission.

---

### Comment · Area_Chair_rsXW · 2025-08-04

Dear Reviewers,

Please kindly note that the author-reviewer discussion period has started. Please take a look at the rebuttals and acknowledge the receipt of it. Meanwhile, you are encouraged to initiate further discussions with the reviewers.

Best,

Your AC

---

### Decision · Program_Chairs · 2025-09-17

**Decision:**

Accept (poster)

**Comment:**

This proposes algorithm for contextual dynamic pricing under the Cox PH model and discrete price set. The model extends prior works which consider a non-contextual setting. The authors propose a Bayesian algorithm for this setting and establish both upper and lower bounds on the regret although the lower bound is for the non-contextual setting only.

The paper is in general well-written, and the setting considered is of practical interest as the price set is likely to be discrete in practice.  On the theory end, this work characterizes two regimes of the regret, which appears to be interesting. In the initial review, there is some concern regarding the incremental contribution made by this model, but the authors did a good job in clarifying them.